# POWER OF SIGN: HIGH PROBABILITY BOUNDS UNDER $(L_0, L_1)$-SMOOTHNESS AND HEAVY-TAILED NOISE

## ABSTRACT

In recent years, non-convex optimization problems are more often described by generalized $(L_0, L_1)$-smoothness assumption rather than standard one. Meanwhile, severely corrupted data used in these problems has increased the demand for methods capable of handling heavy-tailed noises, i.e., noises with bounded $\kappa$-th moment. Motivated by these real-world trends and challenges, we explore sign-based methods in this setup and demonstrate their effectiveness in comparison with other popular solutions like clipping or normalization. In theory, we prove the first-known high probability convergence bounds under $(L_0, L_1)$-smoothness and heavy-tailed noises with mild parameter dependencies. In the case of standard smoothness, these bounds are novel for sign-based methods as well. In particular, SignSGD with batching achieves sample complexity $\tilde{O}\left(\left(\frac{\Delta L_0}{\varepsilon^2} + \frac{\Delta L_1}{\varepsilon}\right)\left[1 + \left(\frac{\sigma}{\varepsilon}\right)^{\frac{\kappa}{\kappa-1}}\right]\right), \kappa \in (1, 2]$. Under the assumption of symmetric noises, SignSGD with Majority Voting can robustly work on the whole range of $\kappa \in (0, 2]$ with complexity $\tilde{O}\left(\left(\frac{\Delta L_0}{\varepsilon^2} + \frac{\Delta L_1}{\varepsilon}\right)\left[\frac{1}{\kappa^2} + \frac{\sigma^2}{\varepsilon^2}\right]\right)$. We also obtain results for parameter-free methods, Polyak-Lojasiewicz functions and momentum-based methods (in expectation). Our theoretical findings are supported by the superior performance of sign-based methods in training Large Language Models compared to clipping and normalization.

## 1 INTRODUCTION

### 1.1 PROBLEM STATEMENT.

Consider a stochastic optimization problem of a smooth non-convex function $f : \mathbb{R}^d \to \mathbb{R}$:

$$\min_{x \in \mathbb{R}^d} f(x) := \mathbb{E}_\xi[f(x, \xi)], \tag{1}$$

where the random variable $\xi$ can only be sampled from an unknown distribution. The main goal is to find a point with the smallest gradient norm. To achieve this, we are able to sample an unbiased estimate $\nabla f(x, \xi) \in \mathbb{R}^d$. For example, in machine learning, $f(x, \xi)$ can be interpreted as a loss function on a sample $\xi$ (Shalev-Shwartz & Ben-David, 2014). The backbone of all popular stochastic first-order methods for solving (1) is Stochastic Gradient Descent (SGD) (Robbins & Monro, 1951):

$$x^{k+1} = x^k - \gamma_k \cdot g^k, \quad g^k := \nabla f(x^k, \xi^k).$$

Huge success of these methods in the rapidly developing neural networks field (Bottou, 2012; Kingma & Ba, 2014) has sparked numerous works studying their convergence under various assumptions on noise corrupting true gradients. For SGD, the optimal sample complexity bound $O(\varepsilon^{-4})$ in expectation (Arjevani et al., 2023) is obtained for sub-Gaussian noise (Nemirovski et al., 2009) and for noise with bounded variance (BV) (Ghadimi & Lan, 2013). These results are derived under classic assumptions. However, motivated by real-world complex Machine Learning applications (Zhang et al., 2020b), modern theoretical papers focus on relaxed assumptions and settings. Below, we give three important stories that are relevant to this paper.

$(L_0, L_1)$-**smoothness.** Usually, for the objective function $f$, standard $L_0$-smoothness is assumed, i.e., $\|\nabla f(x) - \nabla f(y)\|_2 \leq L_0\|x - y\|_2, \forall x, y \in \mathbb{R}^d$. However, a new generalized $(L_0, L_1)$-smoothness assumption was recently proposed and motivated for Large Language Models (LLM) in (Zhang

et al., 2020b). This assumption describes objective functions with a linearly growing Hessian norm: $\|\nabla^2 f(x)\|_2 \leq L_0 + L_1\|\nabla f(x)\|_2, \forall x \in \mathbb{R}^d$. In ongoing research, other variants of this assumption were introduced: for only once differentiable functions (Chen et al., 2023), for symmetrically and asymmetrically growing powers of norms (Chen et al., 2023), and for sub-quadratic polynomially growing norms (Li et al., 2023a). Generalized smoothness applications can be found not only in LLM training (Zhang et al., 2020b; Liu et al., 2023a), but also in distributionally robust optimization (Levy et al., 2020; Jin et al., 2021), multitask learning (Zhang et al., 2024a), federated learning (Liu et al., 2022b), and bilevel optimization (Hao et al.; Gong et al.). The convergence of the most popular optimization algorithms Adam (Kingma & Ba, 2014) and SGD was explored under various noise and generalized smoothness assumptions in works (Li et al., 2023b; Zhang et al., 2024b; Wang et al., 2024b;a) and (Li et al., 2023a), respectively.

**High probability bounds.** Due to the expensive training of large deep learning models (Davis et al., 2021), *high probability (HP)* bounds have gained even more attention than bounds in expectation describing the behavior of stochastic methods over several runs. HP bounds provide convergence guarantees that hold true with probability at least $1 - \delta, \delta \in (0, 1)$. The bound in expectation can be reduced to the HP bound using Markov's inequality; however, it leads to a dominant $1/\delta$ factor. Meanwhile, much milder $\log 1/\delta$ factors can be achieved. For SGD, HP bound $O(\varepsilon^{-4}\log 1/\delta)$ under sub-Gaussian noise is obtained in (Li & Orabona, 2020). However, already under BV noise, SGD has $1/\sqrt{\delta}$ dependence under standard (Sadiev et al., 2023) and $(L_0, L_1)-$smoothness (Li et al., 2023a).

**Heavy-tailed noise.** Moreover, it is shown that the BV assumption cannot describe noises in loss functions in modern deep learning problems. In Transformer models, stochasticity tends to have a rather *heavy-tailed (HT)* distribution (Zhang et al., 2020c; Gurbuzbalaban et al., 2021). This means that the noise has bounded $\kappa$-th moment for some $\kappa \in (1, 2]$, that is, $\mathbb{E}_\xi[\|\nabla f(x, \xi) - \nabla f(x)\|_2^\kappa] \leq \sigma^\kappa$. The desire to obtain better $\delta$-dependence in HP bounds and to consider HT noise motivated the development of more robust modifications of SGD, e.g. SGD with clipping or normalization of the input gradient estimates. In this work, we show that applying a simple sign operator to the gradient estimates is an effective and comparable solution to cope with heavy-tailed noise as well.

## 1.2 RELATED WORKS

**Clipping.** The idea of clipping the norm of the gradient estimate to reduce heavy noise demonstrates significant empirical results (Pascanu et al., 2013; Goodfellow et al., 2016) and helps achieve $\log 1/\delta$ dependence under BV noise and standard smoothness (Nazin et al., 2019; Gorbunov et al., 2020). The clipping operator is defined as $\mathtt{clip}(g^k, \lambda_k) := \min\{1, \lambda_k/\|g^k\|_2\} \cdot g^k$ and SGD with clipping is called ClipSGD. Clipping can also be applied to convex optimization, variational inequalities (Sadiev et al., 2023), non-smooth optimization (Zhang et al., 2020c), zeroth-order optimization (Kornilov et al., 2024), robust aggregation (Karimireddy et al., 2021), distributed optimization (Liu et al., 2022b; Qin et al., 2025) and ensuring differential privacy (Andrew et al., 2021).

*For standard smoothness*, let us list the latest results on the HP convergence of ClipSGD under HT noise. First, for non-convex functions, the authors of (Zhang et al., 2020c) proved lower bounds $O(\varepsilon^{-(3\kappa-2)/(\kappa-1)})$ for sample complexity in expectation. As shown in (Nguyen et al., 2023), with fixed proper clipping levels and stepsizes, ClipSGD achieves the complexity $\tilde{O}_\delta(\varepsilon^{-(3\kappa-2)/(\kappa-1)})$. Remarkably, this complexity can be achieved via both small and large batchsizes. ClipSGD can also work with an infinite horizon resulting in extra $\log 1/\varepsilon$ factors in complexity (Nguyen et al., 2023). In (Sadiev et al., 2023), the authors apply ClipSGD to Polyak-Lojasiewicz functions and obtain faster convergence $\tilde{O}_{\delta,\varepsilon}(\varepsilon^{-\kappa/2(\kappa-1)})$. In a number of works (Chen et al., 2020; Puchkin et al., 2024), the authors work with symmetric HT noise to eliminate the dependence on $\kappa$, expand the range of feasible $\kappa$ for $\kappa \in (0, 1]$ and break the actual lower bounds from (Zhang et al., 2020c).

For $(L_0, L_1)-$*smoothness*, only BV noise is considered. Unlike the case of standard smoothness, the possibility of fast convergence of ClipSGD with small batchsizes remains unclear. In (Koloskova et al., 2023), the authors show that ClipSGD with constant clipping levels and stepsizes unavoidably requires either decreasing of the noise level via batching or significant increasing of the clipping levels. ClipSGD with the small batchsizes convergences slowly with rates $O(L_0\varepsilon^{-4} + L_1\varepsilon^{-5})$ in expectation. Meanwhile, large batchsizes help to achieve faster rates $O(L_0\varepsilon^{-4} + L_1\varepsilon^{-3})$. Another solution for small batchsizes is ClipSGD with momentum called M-ClipSGD (Zhang et al., 2020a) which achieves the fast rates $O(\varepsilon^{-4})$ when the noise in the gradient estimates is bounded.

Despite the effectiveness of clipping, it requires careful tuning, which depends on the iteration and the characteristics of the optimization problem (Sadiev et al., 2023, Theorem. 3.1).

**Normalization.** A natural simplification of clipping with a profound level schedule is the permanent normalization of the gradient estimate, i.e., $\text{norm}(g^k) := g^k/\|g^k\|_2$. SGD with normalization is called NSGD (Hazan et al., 2015; Zhao et al., 2021). Since normalization does not provide such subtle control over the noise like clipping, NSGD requires large batchsizes to soften it. For small batches, one can use NSGD with momentum called M-NSGD (Jin et al., 2021; Cutkosky & Mehta, 2020).

For *standard smoothness*, the HP convergence of vanilla NSGD under HT noise is proved in (Hübler et al., 2024a). The authors show that its complexity is $O(\varepsilon^{-(3\kappa-2)/(\kappa-1)} \log 1/\delta)$ for optimal parameters and $O(\varepsilon^{-2\kappa/(\kappa-1)} \log 1/\delta)$ for parameter-agnostic tuning. The same complexities hold for M-NSGD, but only in expectation. In (Liu et al., 2023b; Cutkosky & Mehta, 2021), normalization is combined with clipping which helps cope with HT noise and obtain suboptimal $\tilde{O}_{\delta,\varepsilon}(\varepsilon^{-(3\kappa-2)/(\kappa-1)})$.

For $(L_0, L_1)$-*smoothness*, in expectation convergence of M-NSGD with rates $O\left((L_0\varepsilon^{-(3\kappa-2)/(\kappa-1)} + L_1\varepsilon^{-(2\kappa-1)/(\kappa-1)})\right)$ is the first known convergence result established simultaneously under HT noise and generalized smoothness (Liu & Zhou, 2024). However, it is only in expectation bounds. In addition, parameter-free tuning over an infinite horizon for M-NSGD is derived only for BV noise achieving the rates $\tilde{O}_\varepsilon(\varepsilon^{-4})$.

**Sign operator.** There is one more promising modification of SGD which behavior under heavy-tailed noise has not yet been studied. Originally proposed in (Bernstein et al., 2018a) for distributed optimization, SignSGD takes only a sign of each coordinate of gradient estimate $\text{sign}(g^k)$. There is one peculiarity in bounds for sign-based methods: they are proved w.r.t. the $\ell_1$-norm instead of smaller $\ell_2$-norm. As a consequence, additional $d$ dependent factors appear.

For *standard smoothness*, SignSGD achieves sample complexity $O(d^2\varepsilon^{-4})$ in expectation under BV noise (Bernstein et al., 2018a). Similar to NSGD, SignSGD requires aggressive batching, which can be substituted by SignSGD with momentum (M-SignSGD) with bound $O(d^4\varepsilon^{-4})$(Sun et al., 2023). The alternative solution is to add error feedback mechanism that additionally fixes the biased nature of the sign operator and allows using convex functions (Seide et al., 2014; Karimireddy et al., 2019).

For $(L_0, L_1)-$*smoothness*, the authors of (Crawshaw et al., 2022) propose generalized SignSGD with Adam-like structure and, in particular, prove bound $O(\varepsilon^{-4} \log(d/\varepsilon))$ for M-SignSGD under almost surely bounded noise. For the same bounded noise, in (Crawshaw & Liu, 2025), the authors study the behavior of various adaptive gradient algorithms and derive lower bounds for them with explicit parameter dependencies.

The main motivation for the original SignSGD is communication effectiveness and empirical robustness in distributed optimization (Bernstein et al., 2018b), since sending sign vector costs $O(d)$ operations. In theory, the $O(d^2\varepsilon^{-4})$ effectiveness is proved only under additional assumptions on noise, e.g., symmetry and unimodality. Other expansions of SignSGD are the following: (Safaryan & Richtárik, 2021) proposes an updated theory for a wider class of noises in the distributed setup, (Liu et al., 2019) generalizes SignSGD to zeroth-order oracle, (Jin et al., 2020) studies federated learning with compression, (Chzhen & Schechtman, 2023; Jiang et al., 2024) explore variance reduction.

## 1.3 CONTRIBUTIONS

**Theory.** Using sign-based methods, **we prove the first-known high probability bounds for non-convex $(L_0, L_1)$-smooth optimization under heavy-tailed noise**. These bounds are valid for all possible problem parameters, have mild dependencies on them, and match the optimal bounds in the case of standard smoothness. Moreover, the **HP results for sign-based methods in case of standard $L_0$-smoothness are novel as well**. Among momentum and adaptive methods, our in expectation bounds are the first to consider together heavy-tailed noise and $(L_0, L_1)$-smoothness. In addition, we consider special cases of Polyak-Lojasiewicz functions, symmetric noises, and parameter-free tuning. The summarized results and comparisons with related works are presented in Table 1.

**Experiments.** To validate our findings in real-world scenarios with heavy-tailed noise and generalized smoothness, in Section 3 we evaluate the sign-based methods on Transformer models,

Table 1: Convergence guarantees for non-convex optimization. The metrics are Avr. $\ell_1$: $\frac{1}{T}\sum_{k=1}^{T}\|\nabla f(x^k)\|_1 \le \varepsilon$, Avr. $\ell_2$: $\frac{1}{T}\sum_{k=1}^{T}\|\nabla f(x^k)\|_2 \le \varepsilon$, Avr. $\ell_2^2$: $\frac{1}{T}\sum_{k=1}^{T}\|\nabla f(x^k)\|_2^2 \le \varepsilon^2$, Min $\ell_2$: $\min_k \|\nabla f(x^k)\|_2 \le \varepsilon$, Func. acc. : $f(x^T) - f(x^*) \le \varepsilon$. HP stands for bounds with probability at least $1-\delta$, $\mathbb{E}$ stands for in expectation bounds. The note 'small batches' means $O(1)$ batchsizes, and 'large batches' means polynomial growth of batchsizes in $1/\varepsilon$. Noise with parameter $\kappa$ satisfies $\mathbb{E}_\xi[|\nabla f(x,\xi)_i - \nabla f(x)_i|^\kappa] \le \sigma_i^\kappa$, and 'Bounded' noise is almost surely $|\nabla f(x,\xi)_i - \nabla f(x)_i| \le \sigma_i$.

| Method | Complexity Bound | Smoothness | Noise (As. 3) | Metric |
|---|---|---|---|---|
| **Basic methods** | | | | |
| minibatch-NSGD (Hübler et al., 2024a) large batches | $O\left(\frac{\Delta L_0}{\varepsilon^2}\left[1 + \left(\frac{\|\vec{\sigma}\|_2}{\varepsilon}\right)^{\frac{\kappa}{\kappa-1}}\right]\right)$ | $L_0$ | $\kappa \in (1,2]$ | HP Avr. $\ell_2$ |
| ClipSGD (Nguyen et al., 2023) small batches | $O\left(\left(\frac{\|\vec{\sigma}\|_2^\kappa \log\frac{1}{\delta}}{\sqrt{\Delta L_0}}\right)^{\frac{3\kappa-2}{\kappa-1}}\left(\frac{\sqrt{\Delta L_0}\log\frac{1}{\delta}}{\varepsilon^2}\right)^{\frac{3\kappa-2}{2\kappa-2}}\right)$ | $L_0$ | $\kappa \in (1,2]$ | HP Avr. $\ell_2^2$ |
| ClipSGD (Koloskova et al., 2023) small batches | $O\left(\frac{\Delta L_1\|\vec{\sigma}\|_2^4}{\varepsilon^5} + \frac{\Delta\|\vec{\sigma}\|_2^2(L_0+L_1\varepsilon)}{\varepsilon^4}\right)$ | $(L_0,L_1)$ | $\kappa = 2$ | $\mathbb{E}$ Min $\ell_2$ |
| ClipSGD (Koloskova et al., 2023) large batches | $O\left(\left(\frac{\Delta L_0}{\varepsilon^2} + \frac{\Delta L_1}{\varepsilon}\right)\left[1 + \left(\frac{\|\vec{\sigma}\|_2}{\varepsilon}\right)^2\right]\right).$ | $(L_0,L_1)$ | $\kappa = 2$ | $\mathbb{E}$ Min $\ell_2$ |
| minibatch-SignSGD (**Th. 1**), large batches | $O\left(\left(\frac{\Delta L_0 d}{\varepsilon^2} + \frac{\Delta L_1 d}{\varepsilon}\right)\left[1 + \left(\frac{\|\vec{\sigma}\|_1}{\varepsilon}\right)^{\frac{\kappa}{\kappa-1}}\right]\log\frac{1}{\delta}\right)$ | $(L_0,L_1)$ | $\kappa \in (1,2]$ | HP Avr. $\ell_1$ |
| **Momentum and adaptive methods, small batches** | | | | |
| M-NSGD (Liu & Zhou, 2024) | $O\left(\left(\frac{\Delta L_0}{\varepsilon^2} + \frac{\Delta L_1}{\varepsilon}\right)\left(1 + \left(\frac{\|\vec{\sigma}\|_2}{\varepsilon}\right)^{\frac{\kappa}{\kappa-1}}\right)\right)$ | $(L_0,L_1)$ | $\kappa \in (1,2]$ | $\mathbb{E}$ Avr. $\ell_2$ |
| M-ClipSGD (Zhang et al., 2020a) | $O\left(\frac{\Delta L_0\|\vec{\sigma}\|_2^2}{\varepsilon^4}\right)$ | $(L_0,L_1)$ | Bounded | $\mathbb{E}$ Avr. $\ell_2$ |
| D-AdaGrad (Crawshaw & Liu, 2025) | $\tilde{\Omega}\left(\frac{\Delta^2 L_0^2\|\vec{\sigma}\|_2^2}{\varepsilon^4} + \frac{\Delta^2 L_1^2\|\vec{\sigma}\|_2^2}{\varepsilon^2\log(1+\Delta L_1^2/L_0)}\right)$ | $(L_0,L_1)$ | Bounded | HP Avr. $\ell_2$ |
| AdaGrad-Norm (Wang et al., 2023) | $\tilde{O}\left(\frac{\Delta^2 L_1^2\|\vec{\sigma}\|_2^2}{\varepsilon^4} + \frac{\Delta L_0\|\vec{\sigma}\|_2^2}{\varepsilon^4} + \frac{\|\vec{\sigma}\|_2^6}{\delta^4\varepsilon^4}\right)$ | $(L_0,L_1)$ | $\kappa = 2$ affine | HP Avr. $\ell_2$ |
| M-SignSGD (**Th. 3**) | $O\left(\left(\frac{\Delta L_0 d}{\varepsilon^2} + \frac{\Delta L_1 d}{\varepsilon}\right)\left[1 + \left(\frac{\|\vec{\sigma}\|_1}{\varepsilon}\right)^{\frac{\kappa}{\kappa-1}}\right]\right)$ | $(L_0,L_1)$ | $\kappa \in (1,2]$ | $\mathbb{E}$ Avr. $\ell_1$ |
| **Polyak-Lojasiewicz functions (As. 4, $\mu > 0$)** | | | | |
| ClipSGD (Sadiev et al., 2023) large batches | $\tilde{O}\left(\frac{L_0}{\mu}\left[1 + \left(\frac{L_0\|\vec{\sigma}\|_2^2}{\mu^2\varepsilon}\right)^{\frac{\kappa}{2(\kappa-1)}}\right]\right)$ | $L_0$ | $\kappa \in (1,2]$ | HP Func. acc. |
| Restarted-SignSGD (**Th. 10**), large batches | $\tilde{O}\left(\left(\frac{L_0 d}{\mu} + \frac{L_1 d\sqrt{\Delta}}{\sqrt{\mu}}\right)\left[1 + \left(\frac{\|\vec{\sigma}\|_1^2}{\mu\varepsilon}\right)^{\frac{\kappa}{2(\kappa-1)}}\right]\right)$ | $(L_0,L_1)$ | $\kappa \in (1,2]$ | HP Func. acc. |
| **Symmetric and unimodal noise** | | | | |
| MajorityVote-SignSGD (Bernstein et al., 2018a;b), large batches | $O\left(\frac{\Delta L_0 d}{\varepsilon^2}\left[1 + \left(\frac{\|\vec{\sigma}\|_1}{\varepsilon}\right)^2\right]\right)$ | $L_0$ | $\kappa = 2$ symmetric | $\mathbb{E}$ Avr. $\ell_1$ |
| MajorityVote-SignSGD (**Th. 2**), large batches | $O\left(\left(\frac{\Delta L_0 d}{\varepsilon^2} + \frac{\Delta L_1 d}{\varepsilon}\right)\left[\frac{1}{\kappa^2} + \left(\frac{\|\vec{\sigma}\|_1}{\varepsilon}\right)^2\right]\log\frac{1}{\delta}\right)$ | $(L_0,L_1)$ | $\kappa \in (0,2]$ symmetric | HP Avr. $\ell_1$ |

specifically on pre-training LLaMA(Touvron et al., 2023) family models of sizes up to 1.3B on the C4 dataset (Raffel et al., 2020) and the Switch Transformer (Fedus et al., 2022) Mixture of Experts (MoE) model on the FineWeb dataset (Penedo et al.). Results demonstrate the effectiveness of sign-based methods compared to other commonly considered techniques to cope with heavy-tailed noise, namely, clipping and normalization. Surprisingly, our results also show that M-SignSGD demonstrates competitive performance and slight improvements compared to AdamW, which is the de facto optimizer for language model training.

## 2  HIGH PROBABILITY BOUNDS FOR SIGN-BASED METHODS UNDER HEAVY-TAILED NOISE AND $(L_0, L_1)$-SMOOTHNESS

In this section, we present our novel non-convex convergence guarantees with high probability for SignSGD with batching and majority voting. We prove them for $(L_0, L_1)$-smooth objective functions with heavy-tailed noise in gradient estimates. We provide the best convergence rates and optimal parameters or rates parameter-free tuning. All proofs are located in Appendix B.

### 2.1  ASSUMPTIONS AND NOTATIONS.

The notation $\overline{1,n}$ represents the set of natural numbers $\{1, 2, \ldots, n\}$. We define $\ell_p$-norm $p \in [1, +\infty]$ as $(\|x\|_p)^p := \sum_{i=1}^d |x_i|^p$, $x \in \mathbb{R}^d$. The notation $\langle x, y \rangle := \sum_{i=1}^d x_i y_i$ denotes the standard scalar product for $x, y \in \mathbb{R}^d$. The sign operator $\text{sign}(\cdot)$ returns the sign of a scalar input and can also be applied element-wise to a vector. The notation $\tilde{O}$ without index omits all logarithmic factors or, if it has an index, omits only the logarithmic factors on the variables from its index.

**Assumption 1** (Lower bound). *The objective function $f$ is lower bounded by $f^* > -\infty$.*

We use the following formulation of $(L_0, L_1)$-smoothness from (Gorbunov et al., 2024).

**Assumption 2** ($(L_0, L_1)$-smoothness). *The objective function $f$ is differentiable and $(L_0, L_1)$-smooth, i.e., for the non-negative constants $(L_0, L_1)$ and $x, y \in \mathbb{R}^d$, it holds*

$$\|\nabla f(x) - \nabla f(y)\|_2 \leq (L_0 + L_1 \sup_{u \in [x,y]} \|\nabla f(u)\|_2) \|x - y\|_2.$$

For examples of $(L_0, L_1)$-smooth functions and their properties, we refer the reader to Appendix B.1.

**Assumption 3** (Heavy-tailed noise in gradient estimates). *The unbiased estimate $\nabla f(x, \xi)$ has bounded $\kappa$-th moment $\kappa \in (1, 2]$ for each coordinate, i.e., $\forall x \in \mathbb{R}^d$:*

$$\mathbb{E}_\xi[\nabla f(x, \xi)] = \nabla f(x), \quad \mathbb{E}_\xi[|\nabla f(x, \xi)_i - \nabla f(x)_i|^\kappa] \leq \sigma_i^\kappa, i \in \overline{1, d}, \tag{2}$$

*where $\vec{\sigma} = [\sigma_1, \ldots, \sigma_d]$ are non-negative constants.*

## 2.2 HP CONVERGENCE PROPERTIES OF THE BACKBONE SignSGD METHOD

We begin our analysis with the simplest of sign-based methods, namely SignSGD (Alg. 1) and prove a general lemma on its convergence with high probability.

---
**Algorithm 1** SignSGD
---
**Input:** Starting point $x^1 \in \mathbb{R}^d$, number of iterations $T$, stepsizes $\{\gamma_k\}_{k=1}^T$.
 1: **for** $k = 1, \ldots, T$ **do**
 2:     Sample $\xi^k$ and compute estimate $x^{k+1} = x^k - \gamma_k \cdot \text{sign}(\nabla f(x^k, \xi^k))$;
 3: **end for**
**Output:** uniformly random point from $\{x^1, \ldots, x^T\}$.

---

**Lemma 1** (SignSGD **Convergence Lemma**). *Consider lower-bounded $(L_0, L_1)$-smooth function $f$ (As. 1, 2) and HT gradient estimates $\vec{\sigma}_k$ (As. 3). Then Alg. 1 after $T$ iterations with non-increasing stepsizes $\gamma_k \leq 1/(48 L_1 d^{\frac{3}{2}} \log \frac{1}{\delta})$ achieves with probability at least $1 - \delta$:*

$$\sum_{k=1}^T \frac{\gamma_k}{16} \|\nabla f(x^k)\|_1 \leq \Delta + L_0 d \sum_{k=1}^T \gamma_k^2 + 2 \sum_{k=1}^T \gamma_k \|\vec{\sigma}_k\|_1 + 6d(\gamma_1 \|\nabla f(x^1)\|_1 + 2 C_T L_0) \log \frac{1}{\delta}, \tag{3}$$

*where $C_T := \max_{k \in \overline{1,T}} \gamma_k \cdot \sum_{\tau=1}^{k-1} \gamma_\tau$ and $\Delta = f(x^1) - f^*$.*

The bound (3) resembles the convergence bound in expectation for SignSGD for $\kappa = 2$ (Bernstein et al., 2018a). The difference is the last term with linear dependence on $\log \frac{1}{\delta}$. Remarkably, $L_1$ constant affects only the upper bound for the largest possible stepsizes $\gamma_k \leq 1/(48 L_1 d^{\frac{3}{2}} \log \frac{1}{\delta})$. In the case of small $L_0$, this is the only condition that prevents us from increasing the stepsize too much. We provide synthetic experiments verifying dependencies in bound (3) in Appendix D.

## 2.3 NEW HP BOUNDS FOR $(L_0, L_1)$-SMOOTHNESS FOR SignSGD WITH AVERAGE BATCHING

In order to upper bound an average accuracy norm from (3) by $\varepsilon$, the noise $\|\vec{\sigma}\|_1$ has not to exceed $\varepsilon$. The first way to lower the noise is to use batch averaging.

---
**Algorithm 2** minibatch-SignSGD
---
**Input:** Starting point $x^1 \in \mathbb{R}^d$, number of iterations $T$, stepsizes $\{\gamma_k\}_{k=1}^T$, batchsizes $\{B_k\}_{k=1}^T$.
 1: **for** $k = 1, \ldots, T$ **do**
 2:     Sample $\{\xi_i^k\}_{i=1}^{B_k}$ and compute $x^{k+1} = x^k - \gamma_k \cdot \text{sign}(\sum_{i=1}^{B_k} \nabla f(x^k, \xi_i^k)/B_k)$;
 3: **end for**
**Output:** uniformly random point from $\{x^1, \ldots, x^T\}$.

---

**Theorem 1** (**HP complexity for** minibatch-SignSGD)**.** *Consider lower-bounded $(L_0, L_1)$-smooth function $f$ (As. 1, 2) and HT gradient estimates (As. 3). Then Alg. 2 requires the sample complexity $N$ to achieve $\frac{1}{T}\sum_{k=1}^{T}\|\nabla f(x^k)\|_1 \le \varepsilon$ with probability at least $1 - \delta$ for:*

*Optimal tuning:* $T = O\left(\frac{\Delta L_1^\delta d}{\varepsilon}\right), \gamma_k \equiv \frac{1}{48 L_1^\delta d}, B_k \equiv \left(\frac{16\|\vec{\sigma}\|_1}{\varepsilon}\right)^{\frac{\kappa}{\kappa-1}}$ *for* $\varepsilon \ge \frac{8L_0}{L_1}$ *and* $T = O\left(\frac{L_0^\delta d}{\varepsilon^2}\right), \gamma_k \equiv \sqrt{\frac{\Delta}{20 L_0^\delta dT}}, B_k \equiv \left(\frac{16\|\vec{\sigma}\|_1}{\varepsilon}\right)^{\frac{\kappa}{\kappa-1}}$ *for* $\varepsilon \le \frac{8L_0}{L_1}$:

$$N = O\left(\left(\frac{\Delta L_0 d}{\varepsilon^2} + \frac{\Delta L_1 d}{\varepsilon}\right)\left[1 + \left(\frac{\|\vec{\sigma}\|_1}{\varepsilon}\right)^{\frac{\kappa}{\kappa-1}}\right]\log 1/\delta\right), \tag{4}$$

*where* $\Delta = f(x^1) - f^*, L_0^\delta = L_0 \log(1/\delta), L_1^\delta = L_1 \log(1/\delta)$.

The proof of Theorem 1 and parameter-free method version are located in Appendices B.4 and A.

### 2.3.1 DISCUSSION

**Optimal tuning bounds.** From Theorem 1, we can clearly distinguish two phases of algorithm convergence: fast initial phase with rate $\tilde{O}_\delta\left(\varepsilon^{-2\kappa-1/\kappa-1}\right)$ before threshold $\varepsilon \ge 8L_0/L_1\sqrt{d}$ and slower one with rate $\tilde{O}_\delta\left(\varepsilon^{-3\kappa-2/\kappa-1}\right)$ after. We provide synthetic experiments verifying two stage convergence and batching effectiveness in Appendix D. In the case of $L_0 \approx 0$ (e.g. for logistic regression (Gorbunov et al., 2024) and deep neural networks (Zhang et al., 2020b)), minibatch-SignSGD runs in the fast regime the whole time and can work with large constant stepsizes. Otherwise, under standard smoothness $L_1 = 0$, the bound (4) matches the lower in expectation bound $\Omega\left(\Delta L_0/\varepsilon^2 + \Delta L_0/\varepsilon^2 \left(\|\vec{\sigma}\|_2/\varepsilon\right)^{\frac{\kappa}{\kappa-1}}\right)$ for first-order stochastic optimization (Zhang et al., 2020c). We also highlight the linear dependence of (4) on $\Delta, L_0, L_1, \log 1/\delta$ and the mild dependence on $\|\vec{\sigma}\|_1$.

**Extra $d$ factors.** In bound (4) for $\ell_1$-norm, there are extra $d$ factors that are missing in bounds for $\ell_2$-norm (Table 1). There are no existing lower bounds for the $\ell_1$-norm to the best of our knowledge. Hence, it is difficult to state whether the $d$ dependence is optimal. In practice (Bernstein et al., 2018a), gradients and noise vector during DL model training via minibatch-SignSGD actually keep high density, i.e., their norms are related by $\|\cdot\|_1 \approx \sqrt{d}\|\cdot\|_2$. Hence, we can substitute $\varepsilon \to \varepsilon'\sqrt{d}, \|\vec{\sigma}\|_1 \to \sqrt{d}\|\vec{\sigma}\|_2, dL_1 \to \sqrt{d}L_1$ (the last transition follows from the proofs, where we can use equality instead of $\|\cdot\|_2 \le \|\cdot\|_1$) in (4) and see that all extra factors are gone.

We would like to highlight that the $d$ linear factor in the bounds arises from the generalized smoothness (As. 2) defined in the $\ell_2$-norm. We can restate it in $\ell_1$-norm as

$$\|\nabla f(x) - \nabla f(y)\|_1 \le (\tilde{L}_0 + \tilde{L}_1 \cdot \sup_u\|\nabla f(u)\|_1)\|x - y\|_\infty.$$

With this assumption, all $d$ factors from the bounds will be gone. In the worst case, the relation between new and old constants $\tilde{L}_0 = dL_0, \tilde{L}_1 = dL_1$ makes factor $d$ unavoidable.

**Comparison under standard smoothness.** According to the HP analysis of ClipSGD from (Nguyen et al., 2023), it achieves the rates from Table 1. These rates have optimal $\tilde{O}_\delta\left(\varepsilon^{-3\kappa-2/\kappa-1}\right)$ dependence on $\varepsilon$, however, dependencies on $\Delta, L_0, \vec{\sigma}$ are much worse than ours. Moreover, we are not aware of any works proposing arbitrary tuning for clipping methods. ClipSGD requires careful clipping level scheduling which allows it to converge with small batchsizes. On the contrary, minibatch-SignSGD needs only large batchsizes. In (Hübler et al., 2024a), the authors analyze minibatch-NSGD and obtain the HP sample complexity w.r.t. to the $\ell_2$-norm (Table 1), the only difference from (4) is the absence of $d$ factors. As explained before, this difference comes from the different norms in the bounds. From a practical point of view, sign-based methods can be applied to distributed optimization (Appendix G) where normalization does not fit. Besides, one can use majority voting as a powerful alternative to batching.

**Comparison under generalized smoothness.** Under BV noise, the only existing results for CLipSGD in expectation with large batches (Koloskova et al., 2023) match our HP bound (4) (see Table 1). Unlike minibatch-SignSGD, CLipSGD can converge with small batches, but much with slower $O(\Delta L_1\|\vec{\sigma}\|_2^4\varepsilon^{-5})$ rates instead of our $\tilde{O}_\delta(\Delta L_1 d^{\frac{3}{2}}\|\vec{\sigma}\|_1^2\varepsilon^{-3})$.

### 2.3.2 POLYAK-LOJASIEWICZ FUNCTIONS.

The minibatch-SignSGD algorithm can be accelerated for the special class of generalized smooth functions that satisfy the Polyak-Lojasiewicz condition.

**Assumption 4** (Polyak-Lojasiewicz (PL))**.** *The objective function $f$ satisfies the PL condition, i.e., for the non-negative constant $\mu$ and $x \in \mathbb{R}^d$, it holds*

$$\|\nabla f(x)\|_2^2 \geq 2\mu(f(x) - f(x^*)).$$

For example, $\mu$-strongly convex functions satisfy the PL condition. A similar behavior has also been observed in over-parameterized models (Liu et al., 2022a). For these functions, we use restarts on minibatch-SignSGD to achieve the HP function accuracy $f(x^T) - f(x^*) \leq \varepsilon$. The explicit algorithm and parameters are presented in Theorem 10, Appendix C, and it achieves the bounds

$$N = \tilde{O}_{\delta,\varepsilon}\left(\left(\frac{L_0 d}{\mu} + \frac{L_1 d\sqrt{\Delta}}{\sqrt{\mu}}\right)\left[1 + \left(\frac{\|\vec{\sigma}\|_1^2}{\mu\varepsilon}\right)^{\frac{\kappa}{2(\kappa-1)}}\right]\right). \tag{5}$$

For the standard smoothness $L_1 = 0$, we compare our method with the most related ClipSGD with the complexity bound $\tilde{O}_{\delta,\varepsilon}\left(\frac{L_0}{\mu}\left[1 + \left(\frac{L_0\|\vec{\sigma}\|_2^2}{\mu^2\varepsilon}\right)^{\frac{\kappa}{2(\kappa-1)}}\right]\right)$ (Sadiev et al., 2023). In contrast, minibatch-SignSGD does not require adjusting the clipping schedule and has better $L_0/\mu$ dependence.

### 2.4 SignSGD WITH MAJORITY VOTING FOR SYMMETRIC HT NOISE

The second approach to noise reduction inherent to sign-based methods is majority voting.

**Majority voting and additional noise assumption.** As mentioned above, the original motivation of SignSGD is fast communication in distributed optimization (Bernstein et al., 2018b; Jin et al., 2020). In the literature, various types of communication were studied, but the most effective one turned out to be majority voting. For sign vectors $\mathrm{sign}(g_i^k), i \in \overline{1, M}$, the resulting update vector is the majority of the received signs $g^k = \mathrm{sign}(\sum_{i=1}^M \mathrm{sign}(g_i^k))$. To be effective, majority voting must decrease the probability of failure of the resulting vector with the growth of $M$. However, for very skewed or bimodal random variables, it might not be true. Choosing the most frequent value from the sign sequence $\{\mathrm{sign}(g_i^k)\}_{i=1}^M$ is actually $M$ Bernoulli trials. In these trials, the probability of choosing a correct answer grows only if the probability of failure of a single worker is less than $\frac{1}{2}$, i.e.: $\mathbb{P}\left[\mathrm{sign}(\nabla f(x^k)) \neq \mathrm{sign}(g_i^k)\right] < \frac{1}{2}, \forall i \in \overline{1, M}$. For example, this condition is satisfied if the noise of the gradient estimate for each coordinate is *unimodal and symmetric about its true value*. We use this assumption, but other assumptions (Safaryan & Richtárik, 2021) are valid as well.

---

**Algorithm 3** MajorityVote-SignSGD

---

**Input:** Starting point $x^0 \in \mathbb{R}^d$, number of iterations $T$, stepsizes $\{\gamma_k\}_{k=1}^T$, batchsizes $\{M_k\}_{k=1}^T$.
1: **for** $k = 1, \ldots, T$ **do**
2:   Sample $\{\xi_i^k\}_{i=1}^{M_k}$ and compute $x^{k+1} = x^k - \gamma_k \cdot \mathrm{sign}\left(\sum_{i=1}^{M_k} \mathrm{sign}(\nabla f(x^k, \xi_i^k))\right)$;
3: **end for**
**Output:** uniformly random point from $\{x^1, \ldots, x^T\}$.

---

**Theorem 2** (**HP complexity for** MajorityVote-SignSGD)**.** *Consider lower-bounded $(L_0, L_1)$-smooth function $f$ (As. 1, 2) and the gradient estimates corrupted by **unimodal and symmetric HT noise** with $\kappa > 0$ (As. 3). Then Alg. 3 requires the sample complexity $N$ to achieve $\frac{1}{T}\sum_{k=1}^T \|\nabla f(x^k)\|_1 \leq \varepsilon$ with probability at least $1 - \delta$ for:*

***Optimal tuning:*** $T = O\left(\frac{\Delta L_1^\delta d}{\varepsilon}\right), \gamma_k \equiv \frac{1}{48 L_1^\delta d}, M_k \equiv \max\left\{\frac{160}{\kappa^2}, \frac{2^{16}\|\vec{\sigma}\|_1^2}{\varepsilon^2}\right\}$ *for $\varepsilon \geq \frac{8L_0}{L_1}$ and* $T = O\left(\frac{\Delta L_0^\delta d}{\varepsilon^2}\right), \gamma_k \equiv \sqrt{\frac{\Delta}{80 L_0^\delta d T}}, M_k \equiv \max\left\{\frac{160}{\kappa^2}, \frac{2^{16}\|\vec{\sigma}\|_1^2}{\varepsilon^2}\right\}$ *for $\varepsilon \leq \frac{8L_0}{L_1}$:*

$$N = O\left(\left(\frac{\Delta L_0 d}{\varepsilon^2} + \frac{\Delta L_1 d}{\varepsilon}\right)\left[\frac{1}{\kappa^2} + \left(\frac{\|\vec{\sigma}\|_1}{\varepsilon}\right)^2\right]\log 1/\delta\right), \tag{6}$$

*where* $\Delta = f(x^1) - f^*, L_0^\delta = L_0 \log(1/\delta), L_1^\delta = L_1 \log(1/\delta).$

The proof of Theorem 2 and parameter-free method version are located in Appendices B.5 and A. For PL functions, we use the restart technique and achieve bounds similar to (5) as if $\kappa = 2$. The results are presented in Theorem 11 in Appendix C.

**Related works and discussion.** Works (Jakovetic et al., 2023; Armacki et al., 2023; 2024) analyze online non-linear SGD without batching for convex and non-convex functions under HT symmetric noise and standard smoothness. It includes a wide range of non-linear transformations of gradient estimates such as *clipping, normalization, and sign operator*. The authors of (Armacki et al., 2024) propose a unified theoretical framework and prove bounds which are arbitrarily close to $O(\varepsilon^{-4})$ for all $\kappa > 0$. In works (Compagnoni et al., 2024; 2025), the authors derive continuous SDE with Student's noise describing SignSGD dynamics and obtain the $O(\varepsilon^{-4})$ HP bound from it.

Similar to these works, the severity of the corrupting noise (the value of $\kappa$) has much milder effect on convergence of MajorityVote-SignSGD in comparison with minibatch-SignSGD and its bound (4). It was known for the standard smoothness, while we show it for the generalized one. Under standard smoothness, the bound (6) with $\log 1/\delta$ factor matches the optimal bound $\Omega\left(\Delta L_0 \|\vec{\sigma}\|_2^2 / \varepsilon^4\right)$ in expectation for first-order methods under *BV noise* (Arjevani et al., 2023).

## 2.5 SignSGD WITH MOMENTUM AND SMALL BATCHES

Both minibatch-SignSGD and MajorityVote-SignSGD methods require increasing batchsizes comparable to the number of iterations. In order to use small batches, one can utilize the momentum technique, resulting in the same total sample complexity. The proof is located in Appendix B.7.

---

**Algorithm 4** M-SignSGD

---

**Input:** Starting point $x^1 \in \mathbb{R}^d$, number of iterations $K$, stepsizes $\{\gamma_k\}_{k=1}^T$, momentums $\{\beta_k\}_{k=1}^T$.
 1: **for** $k = 1, \ldots, T$ **do**
 2:    Sample $\xi^k$ and compute $m^k = \beta_k m^{k-1} + (1 - \beta_k)\nabla f(x^k, \xi^k)$;
 3:    Set $x^{k+1} = x^k - \gamma_k \cdot \text{sign}(m^k)$;
 4: **end for**
**Output:** uniformly random point from $\{x^1, \ldots, x^T\}$.

---

**Theorem 3** (**Complexity for** M-SignSGD **in expectation**). *Consider a lower-bounded* $(L_0, L_1)$-*smooth function* $f$ *(As. 1, 2) and HT gradient estimates (As. 3). Then, Alg. 4 requires* $T$ *iterations to achieve* $\frac{1}{T}\sum_{k=1}^T \mathbb{E}\left[\|\nabla f(x^k)\|_1\right] \leq \varepsilon$ *starting with* $\Delta = f(x^1) - f^*$:

***Optimal tuning:*** $\beta_k \equiv 1 - \min\left\{1, \left(\frac{\Delta L_1 \sqrt{d}}{T\|\vec{\sigma}\|_\kappa}\right)^{\frac{\kappa}{2\kappa-1}}\right\}, \gamma_k \equiv \frac{1-\beta_k}{8}\frac{1}{L_1 d}$ *for* $\varepsilon \geq \frac{3L_0}{L_1}$ *and* $1 - \beta_k \equiv$
$1 - \min\left\{1, \left(\frac{\Delta L_0}{T\|\vec{\sigma}\|_\kappa^2}\right)^{\frac{\kappa}{3\kappa-2}}\right\}, \gamma_k \equiv \sqrt{\frac{\Delta(1-\beta_k)}{TL_0 d}}$ *for* $\varepsilon \leq \frac{3L_0}{L_1}$:

$$T = O\left(\left(\frac{\Delta L_0 d}{\varepsilon^2} + \frac{\Delta L_1 d}{\varepsilon}\right)\left(1 + \left(\frac{\|\vec{\sigma}\|_1}{\varepsilon}\right)^{\frac{\kappa}{\kappa-1}}\right)\right). \quad (7)$$

**Related works.** As we mentioned before, vanilla ClipSGD, NSGD and SignSGD cannot effectively handle small constant batchsizes under $(L_0, L_1)$-smoothness. Hence, here we compare with the works dedicated to theoretical analysis of more powerful modifications for noise control.

In (Liu & Zhou, 2024), the authors analyze M-NSGD under the same assumptions and derive the rates identical to ours in (7) up to $d$ factors which are inherent to sign-based methods. In comparison with adaptive methods (Wang et al., 2023; Crawshaw & Liu, 2025), our results remain valid for all possible values of parameters $L_0, L_1, \|\vec{\sigma}\|_2, \Delta \geq 0$ and accuracy $\varepsilon$. Moreover, our bounds demonstrate a milder dependence on $\varepsilon, \Delta, L_0, L_1$ and allow HT noise. Although the rates in (Wang et al., 2023; Crawshaw & Liu, 2025) are formally proved with high probability, they are derived either under restrictive a.s. bounded noise or under BV noise with suboptimal polynomial $\delta$ dependence.

**Parameter-free tuning.** In practice, the characteristics of the optimized function and corrupting noise are usually unavailable. To address this challenge, we propose a special parameter-agnostic tuning for our most applicable method, M-SignSGD, and analyze its convergence guarantees.

**Theorem 4** (**Complexity for Parameter-Free** M-SignSGD **in expectation**). *Consider a lower-bounded $(L_0, L_1)$-smooth function $f$ (As. 1, 2) and HT gradient estimates (As. 3). Then, Alg. 4 requires $T$ iterations to achieve $\min_{k \in \overline{1,T}} \mathbb{E}\|\nabla f(x^k)\|_1 \leq \varepsilon$ starting with $\Delta = f(x^1) - f^*$:*

*Paramer-free tuning:* $\beta_1 = 1, \beta_k = 1 - 1/\sqrt{k}$, $\gamma_k = \gamma_0 k^{-3/4}$, $\gamma_0 \leq \frac{1}{90 L_1 d}$:

$$T = \tilde{O}\left( \left( \frac{(\Delta/\gamma_0 + L_0 \gamma_0 d)}{\varepsilon} \right)^4 + \left( \frac{\|\vec{\sigma}\|_1}{\varepsilon} \right)^{\frac{2\kappa}{\kappa - 1}} \right). \tag{8}$$

Although parameter-free tuning has a worse polynomial dependence on parameters $L_0, L_1, \Delta$ and non-optimal rates in $\varepsilon$ for $\kappa \neq 2$, it does not require any knowledge about the optimization problem. The parameter-free tunings for other our methods and their discussion are located in Appendix A.

## 3 EXPERIMENTS

In this section, we present experimental results for sign-based methods described in Section 2. To demonstrate the effectiveness of sign-based methods, we focus on language model training tasks. This choice is motivated by two factors: first, these tasks are known to exhibit heavy-tailed noise Zhang et al. (2020c) and generalized smoothness Zhang et al. (2020b); Liu et al. (2023a) characteristics, and second, they represent an important real-world application domain.

To evaluate the performance of M-SignSGD (Algorithm 4) we adopt the

Table 2: Comparison of validation perplexity for various optimization methods across LLaMA model scales trained on C4

| Method | Perplexity ↓ | | |
|---|---|---|---|
| Model size | 130M | 350M | 1.3B |
| M-SignSGD | $\mathbf{18.37}_{\pm.01}$ | **13.73** | **11.56** |
| M-NSGD | $19.28_{\pm.03}$ | 14.60 | 12.62 |
| M-ClippedSGD | $18.95_{\pm.03}$ | 14.30 | 12.30 |
| AdamW | $18.67_{\pm.00}$ | 13.78 | 11.57 |
| Training tokens | 10B | 30B | 30B |
| Number of iterations | 100k | 300k | 300k |

established experimental setup from Lialin et al. (2023), training LLaMA-like models (Touvron et al., 2023) of various sizes — up to 1.3B parameters — on the Colossal Clean Crawled Corpus (C4) dataset (Raffel et al., 2020). The C4 dataset represents a colossal, cleaned version of Common Crawl's web corpus, specifically designed for pre-training language models and word representations.

For our comparison, we focus on two key techniques for handling heavy-tailed noise and small batches: gradient clipping with momentum and gradient normalization with momentum. As representative methods, we choose M-ClippedSGD Zhang et al. (2020a) and M-NSGD Cutkosky & Mehta (2020), respectively. We also compare to AdamW Loshchilov (2017), as a de-facto method for the first-order optimization algorithm for deep learning. To ensure a fair comparison, we conduct an extensive grid search over key hyperparameters, including learning rate, weight decay, and clipping level. Detailed information on the final hyperparameters and experimental setup is provided in Appendix F.1.

Table 2 presents final validation perplexity for each method. M-SignSGD demonstrates superior performance over other heavy-tail mitigating baselines, aligning with our theoretical results. Furthermore, to our surprise, we discovered that M-SignSGD outperforms the strong AdamW baseline, despite careful hyperparameter tuning of the latter. These findings highlight the significant potential of M-SignSGD for language model pretraining. Given these promising results on smaller LLaMA models, we invite the research community to further explore sign-based optimization methods for large-scale LLM training and other computationally demanding applications of practical importance.

To ensure the generalizability of our findings, we complemented our experiments with a new setup — new architecture and data. We have switched model to the Switch Transformer MoE architecture (Fedus et al., 2022), and data to the FineWeb dataset (Penedo et al.), a popular corpus for LLM pre-training. Further details on experimental setup and results can be found in Appendix E.

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

CONTENTS

## A  PARAMETER-FREE METHODS

In this section, we provide parameter-free tunings for all our methods and discuss them.

**Theorem 5** (**HP complexity for** minibatch-SignSGD, **infinite horizon**). *Consider lower-bounded* $(L_0, L_1)$*-smooth function* $f$ *(As. 1, 2) and HT gradient estimates (As. 3). Then Alg. 2 requires the sample complexity* $N$ *to achieve* $\min_{k \in \overline{1,T}} \|\nabla f(x^k)\|_1 \leq \varepsilon$ *with probability at least* $1 - \delta$ *for:*

*Parameter-free tuning: Until plateau* $\gamma_k = \gamma_0 \leq \frac{1}{48 L_1^\delta d}, B_k = B_0 k^2$, *after* $\gamma_k = \frac{\gamma_0}{\sqrt{k}}, B_k = B_0 k$:

$$\varepsilon \geq \frac{8L_0}{L_1} \quad \Rightarrow \quad N = \tilde{O}\left( B_0 \left(\frac{\Delta}{\gamma_0 \varepsilon}\right)^3 + \frac{1}{B_0^2} \left(\frac{\|\vec{\sigma}\|_1}{\varepsilon}\right)^{\frac{3\kappa}{2(\kappa-1)}} \right),$$

$$\varepsilon \ll \frac{8L_0}{L_1} \quad \Rightarrow \quad N = \tilde{O}\left( \frac{B_0(L_0^\delta \gamma_0 d + \Delta/\gamma_0)^4}{\varepsilon^4} + \frac{1}{B_0} \left(\frac{\|\vec{\sigma}\|_1}{\varepsilon}\right)^{\frac{2\kappa}{\kappa-1}} \right),$$

*where* $\Delta = f(x^1) - f^*, L_0^\delta = L_0 \log(1/\delta), L_1^\delta = L_1 \log(1/\delta)$.

The proof is located in the second part of Appendix B.4.

**Theorem 6** (**HP complexity for** MajorityVote-SignSGD, **infinite horizon**). *Consider lower-bounded* $(L_0, L_1)$*-smooth function* $f$ *(As. 1, 2) and HT gradient estimates corrupted by* **unimodal and symmetric HT noise with** $\kappa > 0$ *(As. 3). Then Alg. 3 requires the sample complexity* $N$ *to achieve* $\min_{k \in \overline{1,T}} \|\nabla f(x^k)\|_1 \leq \varepsilon$ *with probability at least* $1 - \delta$ *for:*

*Parameter-free tuning: Until plateau* $\gamma_k = \gamma_0 \leq \frac{1}{48 L_1^\delta d}, M_k = M_0 k^2/\kappa^2$, *after* $\gamma_k = \frac{\gamma_0}{\sqrt{k}}, M_k = M_0 k/\kappa^2$:

$$\varepsilon \geq \frac{8L_0}{L_1} \quad \Rightarrow \quad N = \tilde{O}\left( \frac{M_0(\Delta/\gamma_0)^3 + \|\vec{\sigma}\|_1^3/M_0^2}{\kappa^2 \varepsilon^3} \right),$$

$$\varepsilon \ll \frac{8L_0}{L_1} \quad \Rightarrow \quad N = \tilde{O}\left( \frac{M_0(L_0^\delta \gamma_0 d + \Delta/\gamma_0)^4 + \|\vec{\sigma}\|_1^4/M_0}{\kappa^2 \varepsilon^4} \right).$$

*where* $\Delta = f(x^1) - f^*, L_0^\delta = L_0 \log(1/\delta), L_1^\delta = L_1 \log(1/\delta)$.

The proof is similar to the proof of Theorem 5 with $\kappa = 2$ and additional condition $M_k \geq 160/\kappa^2$.

The parameter-free tuning for our M-SignSGD (Algorithm 4) is presented in Theorem 4.

**Parameter-free tuning bounds.**  For our methods, we use dynamic parameter-free tuning. We start with the largest stepsize $\gamma_0$ for which the method converges and continue until it reaches the plateau observed in the convergence bounds. After that, we begin to decrease the stepsizes. Parameter-free tuning does not require any knowledge about the optimization problem, but it has worse polynomial dependence on parameters $L_0, L_1, \Delta$ and slower (for asymmetric noise $\kappa \neq 2$) non-optimal rates.

The works (Hübler et al., 2024a;b) were the first to introduce this approach for building parameter-free tunings. They study it only for normalized methods and proposed tunings for NSGD and M-NSGD. However, these tunings were built either only under the assumption of standard smoothness and heavy-tailed noise, or under generalized smoothness and BV noise. That is, we are the first to build a tuning under both assumptions simultaneously. Under HT noise, the authors also observe that parameter-free tuning leads to the same slower non-optimal rates. In (Hübler et al., 2024b), they apply simple parameter-free tuning to M-NSGD and obtain in BV expectation bound $\tilde{O}_\varepsilon((\Delta/\gamma_0 + \gamma_0 L_0)e^{(\gamma_0 L_1)^2} + \|\vec{\sigma}\|_2)^4/\varepsilon^4)$. Unlike our methods, M-NSGD works with any stepsize $\gamma_0$ and , but has exponential dependence on $L_1$ due to it. Moreover, we explicitly divide convergence into two stages, what helps us to get accelerated rates during the initial training phase.

**Practical heuristics justification.** Using parameter-free tuning bounds, we can explain why popular practical heuristics for training neural networks such as grid search of hyperparmaters and decreasing stepsizes successfully work in real-world problems.

First, one can grid search hyperparameters (e.g., initial stepsize $\gamma_0$) and choose the best ones according to the achieved final accuracy. Parameter-free Tuning Theorems guarantee convergence to any accuracy for almost all $\gamma_0, B_0$, and the only difference is the convergence speed. Hence, trying various values reveals which values pair better with the unknown problem parameters $L_0, L_1, \vec{\sigma}$.

Second, one can start to decrease stepsizes at any moment before the oscillating plateau, even from the beginning of the training. In this case, the initial fast convergence phase is not fully utilized, and the slower speed comes earlier for all accuracies $\varepsilon > 0$.

## B PROOFS

### B.1 $(L_0, L_1)$-SMOOTHNESS

Standard $L$-smoothness assumes that the gradient of a function is globally Lipschitz continuous. However, this condition can be too restrictive in practice. Many functions arising in optimization, especially in Machine Learning and statistics, either do not satisfy $L$-smoothness or satisfy it with a very large constant $L_0$, leading to overly pessimistic theoretical guarantees. $(L_0, L_1)$-smoothness (Assumption 2) is weaker than $L$-smoothness and allows finer control over the smoothness behavior of functions with rapidly growing curvature in regions where the gradient is large.

Importantly, many functions satisfy $(L_0, L_1)$-smoothness with *significantly smaller constants* $L_0$ and $L_1$ compared to the $L$ required for global Lipschitz smoothness. As a result, optimization algorithms tailored for $(L_0, L_1)$-smooth functions can achieve better convergence guarantees, especially in settings involving large gradients or heavy-tailed noise. The examples of practically used $(L_0, L_1)$-smooth functions include:

**Example 1** (Power of Norm)**.** *Let $f(x) = \|x\|^{2n}$, where $n$ is a positive integer. Then, $f(x)$ is convex and $(2n, 2n-1)$-smooth. Moreover, $f(x)$ is not $L$-smooth for $n \geq 2$ and any $L \geq 0$.*

**Example 2** (Exponent of the Inner Product)**.** *Function $f(x) = \exp(a^\top x)$ for some $a \in \mathbb{R}^d$ is convex, $(0, \|a\|)$-smooth, but not $L$-smooth for $a \neq 0$ and any $L \geq 0$.*

**Example 3** (Logistic Function)**.** *Consider logistic function: $f(x) = \log\left(1 + \exp(-a^\top x)\right)$, where $a \in \mathbb{R}^d$ is some vector. It is known that this function is $L$-smooth and convex with $L = \|a\|^2$. However, one can show that $f$ is also $(L_0, L_1)$-smooth with $L_0 = 0$ and $L_1 = \|a\|$. For $\|a\| \gg 1$, both $L_0$ and $L_1$ are much smaller than $L$.*

**Example 4** (Quadratic Function with Linear Term.)**.** *Let $f(x) = \frac{1}{2} x^\top A x + b^\top x$, where $A \in \mathbb{R}^{d \times d}$ is symmetric positive semi-definite, and $b \in \mathbb{R}^d$. Then $f$ is convex and $(L_0, 0)$-smooth with $L_0 = \|A\|$. This function is also $L$-smooth with the same $L$, but here $(L_1 = 0)$ shows the gradient is Lipschitz regardless of gradient size.*

The condition of $(L_0, L_1)$-smoothness from Assumption 2 can be formulated in terms of inequalities without $\sup$ operator, similar to the case of standard smoothness.

**Lemma 2.** *($(L_0, L_1)$-smoothness properties (Gorbunov et al., 2024)) For $(L_0, L_1)$-smooth function $f$ (As. 2) and $x, y \in \mathbb{R}^d$, it holds*

$$\|\nabla f(x) - \nabla f(y)\|_2 \leq (L_0 + L_1 \|\nabla f(y)\|_2) \exp(L_1 \|x - y\|_2) \|x - y\|_2,$$

$$f(y) - f(x) - \langle \nabla f(x), y - x \rangle \leq \frac{L_0 + L_1 \|\nabla f(x)\|_2}{2} \exp(L_1 \|x - y\|_2) \|x - y\|_2^2. \tag{9}$$

### B.2 TECHNICAL LEMMAS AND PROPOSITIONS

We use the following facts from the linear algebra and convex analysis (Boyd, 2004):

**Proposition 1** (Norm Relation)**.** *For two norms $\ell_p$ and $\ell_q$ with $1 \leq p \leq q \leq 2$, the following relation holds true:*

$$\|x\|_q \leq \|x\|_p \leq d^{\frac{1}{p} - \frac{1}{q}} \|x\|_q, \quad \forall x \in \mathbb{R}^d. \tag{10}$$

**Proposition 2** (Jensen's Inequality). *For scalar random variable $\xi$ with bounded $\kappa$-th moment $\kappa \in (1, 2]$, the following inequality holds true:*

$$\mathbb{E}[|\xi|] \leq (\mathbb{E}[|\xi|^{\kappa}])^{\frac{1}{\kappa}}. \tag{11}$$

**Proposition 3** (Markov's Inequality). *For scalar random variable $\xi$ with bounded first moment, the following inequality holds true for any $a > 0$:*

$$\mathbb{P}(|\xi - \mathbb{E}[\xi]| \geq a) \leq \frac{\mathbb{E}[|\xi|]}{a}. \tag{12}$$

To prove the HP bounds with the logarithmic dependence, we use the following measure concentration result (see, for example, (Li & Orabona, 2020, Lemma 1).

**Lemma 3** (Measure Concentration Lemma). *Let $\{D_k\}_{k=1}^{T}$ be a martingale difference sequence (MDS), i.e., $\mathbb{E}[D_k | D_{k-1}, \ldots, D_1] = 0$ for all $k \in \overline{1, T}$. Furthermore, for each $k \in \overline{1, T}$, there exists positive $\sigma_k \in \mathbb{R}$, s.t. $\mathbb{E}\left[\exp\left(\frac{D_k^2}{\sigma_k^2}\right) | k\right] \leq e$. Then the following probability bound holds true:*

$$\forall \lambda > 0, \delta \in (0, 1) : \quad \mathbb{P}\left(\sum_{k=1}^{T} D_k \leq \frac{3}{4}\lambda \sum_{k=1}^{T} \sigma_k^2 + \frac{1}{\lambda} \log(1/\delta)\right) \geq 1 - \delta. \tag{13}$$

To control error reduction during batching, we use the following batching lemma for HT variables. Its modern proof for $d = 1$ was proposed in (Cherapanamjeri et al., 2022, Lemma 4.2) and then generalized for the multidimensional case in (Kornilov et al., 2024; Hübler et al., 2024a).

**Lemma 4** (HT Batching Lemma). *Let $\kappa \in (1, 2]$, and $X_1, \ldots, X_B \in \mathbb{R}^d$ be a martingale difference sequence (MDS), i.e., $\mathbb{E}[X_i | X_{i-1}, \ldots, X_1] = 0$ for all $i \in \overline{1, B}$. If all variables $X_i$ have bounded $\kappa$-th moment, i.e., $\mathbb{E}[\|X_i\|_2^{\kappa}] < +\infty$, then the following bound holds true*

$$\mathbb{E}\left[\left\|\frac{1}{B}\sum_{i=1}^{B} X_i\right\|_2^{\kappa}\right] \leq \frac{2}{B^{\kappa}} \sum_{i=1}^{B} \mathbb{E}[\|X_i\|_2^{\kappa}]. \tag{14}$$

We generalize the following lemma about changes after one update step of sign-based momentum methods from (Sun et al., 2023, Lemma 1).

**Lemma 5** (Sign Update Step Lemma). *Let $x, m \in \mathbb{R}^d$ be arbitrary vectors, $A = diag(a_1, \ldots, a_d)$ be diagonal matrix and $f$ be $(L_0, L_1)$-smooth function (As. 2). Then for the update step*

$$x' = x - \gamma \cdot A \cdot \text{sign}(m)$$

*with $\epsilon := m - \nabla f(x)$, the following inequality holds true*

$$f(x') - f(x) \leq -\gamma\|A\nabla f(x)\|_1 + 2\gamma\|A\|_F\|\epsilon\|_2 + \frac{L_0 + L_1\|A\nabla f(x^k)\|_2}{2} \exp\left(\gamma L_1\|A\|_F\right)\gamma^2\|A\|_F^2. \tag{15}$$

*and in particular, if $A = I$ for the $\ell_1$ and $\ell_2$ norms:*

$$f(x') - f(x) \leq -\gamma\|\nabla f(x)\|_1 + 2\gamma\|\epsilon\|_1 + \frac{L_0 + L_1\|\nabla f(x)\|_2}{2} \exp(\gamma L_1 \sqrt{d})\gamma^2 d. \tag{16}$$

*Proof.* Using $(L_0, L_1)$-smoothness of $f$ (Lemma 2) between points $x$ and $x'$, we have

$$f(x') \leq f(x) + \langle \nabla f(x), x' - x \rangle + \frac{L_0 + L_1\|x' - x\|}{2}\|x' - x\|^2 \exp(L_1\|x' - x\|).$$

Substitute $x' - x = -\gamma A\text{sign}(m)$ gives us:

$$\langle \nabla f(x), x' - x \rangle = -\gamma\langle \nabla f(x), A\text{sign}(m) \rangle.$$

Next, we decompose the inner product:

$$\langle \nabla f(x), A\text{sign}(m) \rangle = \langle \nabla f(x), A\text{sign}(\nabla f(x)) \rangle + \langle \nabla f(x), A(\text{sign}(m) - \text{sign}(\nabla f(x))) \rangle.$$

We use the identity:
$$\langle \nabla f(x), A\mathrm{sign}(\nabla f(x)) \rangle = \|A\nabla f(x)\|_1,$$
and define $[\nabla f(x)]_i =: g_i$, then the second term becomes
$$\sum_{i=1}^{d} a_i g_i \left( \mathrm{sign}(m_i) - \mathrm{sign}(g_i) \right).$$

Now we analyze two cases for each $i$:

- If $\mathrm{sign}(m_i) = \mathrm{sign}(g_i)$, then the term is equal to zero.

- Otherwise, $g_i \cdot m_i \le 0$, hence $|g_i - m_i| \ge |g_i|$, and we have the following with $\epsilon_i := m_i - g_i$:
$$a_i g_i \left( \mathrm{sign}(m_i) - \mathrm{sign}(g_i) \right) \le 2a_i|g_i| \le 2a_i|\epsilon_i|.$$

In total, we have:

$$\langle \nabla f(x), A\mathrm{sign}(\nabla f(x)) - A\mathrm{sign}(m) \rangle \le 2\sum_{i=1}^{d} a_i|\epsilon_i| \le 2\|A\|_F\|\epsilon\|_2,$$

$$\langle \nabla f(x), x' - x \rangle \le -\gamma\|A\nabla f(x)\|_1 + 2\gamma\|A\|_F\|\epsilon\|_2.$$

Finally, we observe that
$$\|x' - x\| = \gamma\|A\mathrm{sign}(m)\|_2 \le \gamma\|A\|_F,$$
and derive the upper bound:

$$f(x') - f(x) \le -\gamma\|A\nabla f(x)\|_1 + 2\gamma\|A\|_F\|\epsilon\|_2 + \frac{L_0 + L_1\|A\nabla f(x)\|_2}{2}\exp(\gamma L_1\|A\|_F)\gamma^2\|A\|_F^2.$$

For the $A = I$ part, substitute $x' - x = -\gamma\mathrm{sign}(m)$, so:
$$\langle \nabla f(x), x' - x \rangle = -\gamma\langle \nabla f(x), \mathrm{sign}(m) \rangle.$$
The inner product decomposition and analysis of $\langle \nabla f(x), \mathrm{sign}(m) \rangle$ follow the previous part, leading to:

$$\langle \nabla f(x), \mathrm{sign}(\nabla f(x)) - \mathrm{sign}(m) \rangle \le 2\sum_{i=1}^{d} |\epsilon_i| = 2\|\epsilon\|_1,$$

$$\langle \nabla f(x), x' - x \rangle \le -\gamma\|\nabla f(x)\|_1 + 2\gamma\|\epsilon\|_1.$$

Substituting into the smoothness bound:

$$f(x') - f(x) \le -\gamma\|\nabla f(x)\|_1 + 2\gamma\|\epsilon\|_1 + \frac{L_0 + L_1\|\nabla f(x)\|_2}{2}\exp(\gamma L_1\sqrt{d})\gamma^2 d.$$

$\square$

From (Hübler et al., 2024b) we use the following lemma:

**Lemma 6** (Technical Lemma). *Let $q \in (0,1), p \ge 0$ and $k > 0$. Further let $a, b \in \mathbb{N}_{\ge 2}$ with $a \le b$. Then the following statements are true.*

*i) We have*
$$\prod_{i=a}^{b}(1 - i^{-q}) \le \exp\left( \frac{1}{1-q}(a^{1-q} - b^{1-q}) \right).$$

*ii) If $p \ge q$, then*
$$\sum_{i=a}^{N} i^{-p} \prod_{\tau=a}^{i}(1 - \tau^{-q}) \le \frac{(a-1)^{q-p}\exp\left( \frac{a^{1-q} - (a-1)^{1-q}}{1-q} \right) - N^{q-p}\exp\left( \frac{a^{1-q} - N^{1-q}}{1-q} \right)}{1 + (p-q)N^{q-1}},$$
*and in particular,*
$$\sum_{i=a}^{N} i^{-p} \prod_{\tau=a}^{i}(1 - \tau^{-q}) \le (a-1)^{q-p}\exp\left( \frac{a^{1-q} - (a-1)^{1-q}}{1-q} \right) = O(a^{q-p}).$$

*iii) If $a \geq p^{\frac{1}{1-q}}$ and $a \geq \left(\frac{p-q}{2}\right)^{\frac{1}{1-q}}$, then*

$$\sum_{i=a}^{b} i^{-p} \prod_{\tau=i+1}^{b} (1 - \tau^{-q}) \leq 2 \exp\left(\frac{1}{1-q}\right)(b+1)^{q-p}.$$

*Note that these requirements are always fulfilled for $p \leq 1$.*

### B.3 PROOF OF SignSGD GENERAL CONVERGENCE LEMMA 1

For beginning, we prove general lemma about SignSGD convergence with HT gradient estimates $g^k$ with $\vec{\sigma}, \kappa \in (1, 2]$. This proof considerably relies on proof techniques for NSGD from (Hübler et al., 2024a).

*Proof.* Consider the $k$-th step of SignSGD. We use $(L_0, L_1)$ smoothness of function $f$ (Lemma 2) to estimate:

$$
\begin{aligned}
f(x^{k+1}) - f(x^k) &\leq \langle \nabla f(x^k), x^{k+1} - x^k \rangle + \frac{L_0 + L_1 \|\nabla f(x^k)\|_2}{2} \exp(L_1 \|x^{k+1} - x^k\|_2) \|x^{k+1} - x^k\|_2^2 \\
&= -\gamma_k \frac{\langle \nabla f(x^k), \text{sign}(g^k) \rangle}{\|\nabla f(x^k)\|_1} \cdot \|\nabla f(x^k)\|_1 + \frac{L_0 d \gamma_k^2}{2} \exp(L_1 \sqrt{d} \gamma_k) \\
&\quad + \frac{L_1 d \gamma_k \exp(L_1 \sqrt{d} \gamma_k)}{2} \cdot \gamma_k \|\nabla f(x^k)\|_2 \\
&\leq -\gamma_k \frac{\langle \nabla f(x^k), \text{sign}(g^k) \rangle}{\|\nabla f(x^k)\|_1} \cdot \|\nabla f(x^k)\|_1 + \frac{L_0 d \gamma_k^2}{2} \exp(L_1 \sqrt{d} \gamma_k) \\
&\quad + \frac{L_1 d \gamma_k \exp(L_1 \sqrt{d} \gamma_k)}{2} \cdot \gamma_k \|\nabla f(x^k)\|_1.
\end{aligned}
$$

Let us choose $\gamma_k \leq \frac{1}{4 L_1 d}$, then we have $L_1 d \gamma_k \exp(L_1 \sqrt{d} \gamma_k) \leq \frac{1}{4}$ and

$$
f(x^{k+1}) - f(x^k) \leq -\gamma_k \frac{\langle \nabla f(x^k), \text{sign}(g^k) \rangle}{\|\nabla f(x^k)\|_1} \cdot \|\nabla f(x^k)\|_1 + L_0 d \gamma_k^2 + \frac{\gamma_k}{4} \|\nabla f(x^k)\|_1.
$$

Consequently, after summing all $T$ steps, we obtain:

$$\sum_{k=1}^{T} \gamma_k \left[ \frac{\langle \nabla f(x^k), \text{sign}(g^k) \rangle}{\|\nabla f(x^k)\|_1} - \frac{1}{4} \right] \cdot \|\nabla f(x^k)\|_1 \leq \underbrace{f(x^1) - f(x^*)}_{=\Delta} + L_0 d \sum_{k=1}^{T} \gamma_k^2. \tag{17}$$

We introduce the following terms $\phi_k := \frac{\langle \nabla f(x^k), \text{sign}(g^k) \rangle}{\|\nabla f(x^k)\|_1} \in [-1, 1]$, $\psi_k := \mathbb{E}[\phi_k | x^k]$ and $D_k := -\gamma_k (\phi_k - \psi_k) \|\nabla f(x^k)\|_1$. We note that $D_k$ is a martingale difference sequence $(\mathbb{E}[D_k | D_{k-1}, \ldots, D_k] = 0)$ and satisfies

$$\exp\left(\frac{D_k^2}{4 \gamma_k^2 \|\nabla f(x^k)\|_1^2}\right) = \exp\left(\frac{(\phi_k - \psi_k)^2}{4}\right) \leq e.$$

Applying Measure Concentration Lemma 3 to MSD $D_k$ with $\sigma_k^2 = 4 \gamma_k^2 \|\nabla f(x^k)\|_1^2$, we derive the bound for all $\lambda > 0$ with probability at least $1 - \delta$:

$$\sum_{k=1}^{T} \gamma_k (\psi_k - 3\lambda \gamma_k \|\nabla f(x^k)\|_1 - 1/4) \|\nabla f(x^k)\|_1 \leq \Delta + L_0 d \sum_{k=0}^{T-1} \gamma_k^2 + \frac{1}{\lambda} \log(1/\delta).$$

We use norm relation (10) and $(L_0, L_1)$-smoothness to estimate maximum gradient norm for all $k \in \overline{2, T+1}$:

$$
\begin{aligned}
\|\nabla f(x^k)\|_1 / \sqrt{d} &\leq \|\nabla f(x^k)\|_2 = \|\nabla f(x^k) - \nabla f(x^{k-1}) + \nabla f(x^{k-1})\|_2 \\
&\leq \|\nabla f(x^k) - \nabla f(x^{k-1})\|_2 + \|\nabla f(x^{k-1})\|_2 \\
&\leq (L_0 + L_1 \|\nabla f(x^{k-1})\|_2) \exp(L_1 \|x^k - x^{k-1}\|_2) \|x^k - x^{k-1}\|_2 + \|\nabla f(x^{k-1})\|_2 \\
&\leq (L_0 + L_1 \|\nabla f(x^{k-1})\|_2) \exp(L_1 \sqrt{d} \gamma_k) \sqrt{d} \gamma_k + \|\nabla f(x^{k-1})\|_2.
\end{aligned}
$$

At this point, we take $\gamma_k \leq \frac{1}{48 L_1 d \log \frac{1}{\delta}}$ to obtain

$$
\begin{aligned}
\|\nabla f(x^k)\|_1/\sqrt{d} &\leq 2L_0\sqrt{d}\gamma_k + \frac{\|\nabla f(x^{k-1})\|_2}{48\sqrt{d}\log\frac{1}{\delta}} + \|\nabla f(x^{k-1})\|_2 \\
&\leq 2L_0\sqrt{d}\sum_{\tau=1}^{k-1}\gamma_\tau + \sum_{\tau=1}^{k-1}\frac{\|\nabla f(x^\tau)\|_2}{48\sqrt{d}\log\frac{1}{\delta}} + \|\nabla f(x^1)\|_2 \\
&\leq 2L_0\sqrt{d}\sum_{\tau=1}^{k-1}\gamma_\tau + \sum_{\tau=1}^{k-1}\frac{\|\nabla f(x^\tau)\|_1}{48\sqrt{d}\log\frac{1}{\delta}} + \|\nabla f(x^1)\|_1,
\end{aligned}
$$

$$
\gamma_k\|\nabla f(x^k)\|_1 \leq 2L_0 d\cdot\gamma_k\sum_{\tau=1}^{k-1}\gamma_\tau + \gamma_k\sum_{\tau=1}^{k-1}\frac{\|\nabla f(x^\tau)\|_1}{48\log\frac{1}{\delta}} + \sqrt{d}\gamma_k\|\nabla f(x^1)\|_1.
$$

Since stepsizes $\gamma_k$ are non-increasing, we have

$$
\gamma_k\sum_{\tau=1}^{k-1}\frac{\|\nabla f(x^\tau)\|_1}{48\log\frac{1}{\delta}} \leq \sum_{\tau=1}^{k-1}\frac{\gamma_\tau\|\nabla f(x^\tau)\|_1}{48\log\frac{1}{\delta}},
$$

$$
\gamma_k\|\nabla f(x^k)\|_1 \leq 2L_0 d\cdot\gamma_k\sum_{\tau=1}^{k-1}\gamma_\tau + \sum_{\tau=1}^{k-1}\frac{\gamma_\tau\|\nabla f(x^\tau)\|_1}{48\log\frac{1}{\delta}} + \sqrt{d}\gamma_k\|\nabla f(x^1)\|_1.
$$

Hence, the choice $\lambda := \frac{1}{6d(\gamma^{max}\|\nabla f(x^1)\|_1 + \sum_{k=1}^{T}\frac{\gamma_k\|\nabla f(x^k)\|_1}{48d\log\frac{1}{\delta}} + 2C_T L_0)}$ where $C_T := \max_{k\in\overline{1,T}}\gamma_k\cdot\sum_{\tau=1}^{k-1}\gamma_\tau$ and $\gamma^{max} := \max_{k\in\overline{1,T}}\gamma_k$ yields with probability at least $1-\delta$:

$$
\begin{aligned}
\sum_{k=1}^{T}\gamma_k\left(\psi_k - \frac{1}{2} - \frac{1}{4}\right)\|\nabla f(x^k)\|_1 &\leq \Delta + L_0 d\sum_{k=1}^{T}\gamma_k^2 + 6\sqrt{d}(\gamma^{max}\|\nabla f(x^1)\|_1 + 2C_T L_0)\log(1/\delta) \\
&\quad + \frac{6}{48}\sum_{k=1}^{T}\gamma_k\|\nabla f(x^k)\|_1,
\end{aligned}
$$

$$
\sum_{k=1}^{T}\gamma_k\left(\psi_k - \frac{1}{2} - \frac{1}{4} - \frac{1}{8}\right)\|\nabla f(x^k)\|_1 \leq \Delta + L_0 d\sum_{k=1}^{T}\gamma_k^2 + 6\sqrt{d}(\gamma^{max}\|\nabla f(x^1)\|_1 + 2C_T L_0)\log(1/\delta),
$$

Next, we estimate each term $\psi_k\|\nabla f(x^k)\|_1$ in the previous sum:

$$
\begin{aligned}
\psi_k\|\nabla f(x^k)\|_1 &= \mathbb{E}\left[\langle\nabla f(x^k), \mathrm{sign}(g^k)\rangle | x^k\right] \\
&= \|\nabla f(x^k)\|_1 - \sum_{i=1}^{d}2|\nabla f(x^k)|_i\cdot\mathbb{P}(\mathrm{sign}(\nabla f(x^k))_i \neq \mathrm{sign}(g^k)_i | x^k). \quad (18)
\end{aligned}
$$

For each coordinate, we have a bound derived from Markov's inequality (12) followed by Jensen's inequality (11):

$$
\begin{aligned}
\mathbb{P}(\mathrm{sign}(\nabla f(x^k))_i \neq \mathrm{sign}(g^k)_i | x^k) &\leq \mathbb{P}(|\nabla f(x^k)_i - g_i^k| \geq |\nabla f(x^k)_i| | x^k) \leq \frac{\mathbb{E}_{\xi^k}[|\nabla f(x^k)_i - g_i^k|]}{|\nabla f(x^k)_i|} \\
&\leq \frac{(\mathbb{E}_{\xi^k}[|\nabla f(x^k)_i - g_i^k|^\kappa])^{\frac{1}{\kappa}}}{|\nabla f(x^k)_i|} \leq \frac{\sigma_{k,i}}{|\nabla f(x^k)_i|}. \quad (19)
\end{aligned}
$$

Hence, the whole sum can be bounded as

$$
\sum_{i=1}^{d}2|\nabla f(x^k)|_i\cdot\mathbb{P}(\mathrm{sign}(\nabla f(x^k))_i \neq \mathrm{sign}(g^k)_i | x^k) \leq 2\|\vec{\sigma}_k\|_1.
$$

Finally, we put this bound in (18) and obtain:

$$
\begin{aligned}
\frac{1}{16}\sum_{k=1}^{T}\gamma_k\|\nabla f(x^k)\|_1 &\leq \Delta + L_0 d\sum_{k=1}^{T}\gamma_k^2 + 2\sum_{k=1}^{T}\gamma_k\|\vec{\sigma}_k\|_1 \\
&\quad + 6d(\gamma^{max}\|\nabla f(x^1)\|_1 + 2C_T L_0)\log(1/\delta). \quad (20)
\end{aligned}
$$

$\square$

The proof of Theorem 1 is divided into two parts: for finite horizon with optimal tuning (Theorem 7) and for infinite horizon with parameter-free tuning (Theorem 5).

**Theorem 7** (**HP complexity for** minibatch-SignSGD, **finite horizon, full version**)**.** *Consider lower-bounded* $(L_0, L_1)$*-smooth function* $f$ *(As. 1, 2) and HT gradient estimates (As. 3). Then Alg. 2 requires the sample complexity* $N$ *to achieve* $\frac{1}{T}\sum_{k=1}^{T}\|\nabla f(x^k)\|_1 \le \varepsilon$ *with probability at least* $1 - \delta$ *for:*

*Optimal tuning for* $\varepsilon \ge \frac{8L_0}{L_1}$: $T = O\left(\frac{\Delta L_1^\delta d}{\varepsilon}\right), \gamma_k \equiv \frac{1}{48 L_1^\delta d}, B_k \equiv \left(\frac{16\|\vec{\sigma}\|_1}{\varepsilon}\right)^{\frac{\kappa}{\kappa-1}}$ :

$$N = O\left(\frac{\Delta L_1^\delta d}{\varepsilon}\left[1 + \left(\frac{\|\vec{\sigma}\|_1}{\varepsilon}\right)^{\frac{\kappa}{\kappa-1}}\right]\right),$$

*Optimal tuning for* $\varepsilon \le \frac{8L_0}{L_1}$: $T = O\left(\frac{\Delta L_0^\delta d}{\varepsilon^2}\right), \gamma_k \equiv \sqrt{\frac{\Delta}{20 L_0^\delta dT}}, B_k \equiv \left(\frac{16\|\vec{\sigma}\|_1}{\varepsilon}\right)^{\frac{\kappa}{\kappa-1}}$ :

$$N = O\left(\frac{\Delta L_0^\delta d}{\varepsilon^2}\left[1 + \left(\frac{\|\vec{\sigma}\|_1}{\varepsilon}\right)^{\frac{\kappa}{\kappa-1}}\right]\right),$$

*Parameter-free tuning for* $\varepsilon \ge \frac{8L_0}{L_1}$: $T, \gamma_k \equiv \gamma_0 \le \frac{1}{48 L_1^\delta d}, B_k \equiv \max\{1, B_0 T^2\}$:

$$N = O\left(B_0 \left(\frac{\Delta}{\varepsilon\gamma_0}\right)^3 + \frac{1}{B_0^2}\left(\frac{\|\vec{\sigma}\|_1}{\varepsilon}\right)^{\frac{3\kappa}{2(\kappa-1)}}\right),$$

*Parameter-free tuning for* $\varepsilon \le \frac{8L_0}{L_1}$: $T, \gamma_k \equiv \frac{\gamma_0}{\sqrt{T}}, B_k \equiv \max\{1, B_0 T\}$:

$$N = O\left(\frac{B_0(\Delta/\gamma_0 + L_0^\delta d\gamma_0)^4}{\varepsilon^4} + \frac{1}{B_0}\left(\frac{\|\vec{\sigma}\|_1}{\varepsilon}\right)^{\frac{2\kappa}{\kappa-1}}\right),$$

*where* $\Delta = f(x^1) - f^*, L_0^\delta = L_0 \log(1/\delta), L_1^\delta = L_1 \log(1/\delta)$.

*Proof.* Plugging in constant stepsizes $\gamma_k \equiv \gamma \le \frac{1}{48 L_1 d \log\frac{1}{\delta}}$ in (20) implies $C_T = T\gamma^2, \gamma^{max} = \gamma$:

$$\frac{1}{T}\sum_{k=1}^{T}\|\nabla f(x^k)\|_1 \le \frac{4\Delta}{T\gamma} + 80 L_0 d\gamma \log(1/\delta) + 8\|\vec{\sigma}_k\|_1 + 24\frac{d\|\nabla f(x^1)\|_1}{T}\log(1/\delta).$$

Due to Batching Lemma 4, we can estimate the $\kappa$-th moment of the batched estimate for constant batchsizes $B_k \equiv B$ as $\|\vec{\sigma}_k\|_1 \le \frac{2\|\vec{\sigma}\|_1}{B^{\frac{\kappa-1}{\kappa}}}$ and derive:

$$\frac{1}{T}\sum_{k=1}^{T}\|\nabla f(x^k)\|_1 \le \frac{4\Delta}{T\gamma} + 80 L_0 d\gamma \log(1/\delta) + 8\frac{\|\vec{\sigma}\|_1}{B^{\frac{\kappa-1}{\kappa}}} + 24\frac{d\|\nabla f(x^1)\|_1}{T}\log(1/\delta).$$

We can omit the last term since its dependence on $T$ has the largest power.

**Case** $\varepsilon \ge \frac{8L_0}{L_1}$, **parameter-free tuning:** We use parameters $T, \gamma_k = \gamma_0, B_k = \max\{1, B_0 T^2\}$ to get:

$$\frac{1}{T}\sum_{k=1}^{T}\|\nabla f(x^k)\|_1 \le \frac{4\Delta}{T\gamma_0} + \varepsilon + 16\frac{\|\vec{\sigma}\|_1}{B_0^{\frac{\kappa-1}{\kappa}}T^{\frac{2(\kappa-1)}{\kappa}}} + 24\frac{d\|\nabla f(x^1)\|_1}{T}\log(1/\delta).$$

Setting such $T$ that the first two terms become less than $\varepsilon$, we obtain the final complexity $N = T \cdot B_0 T^2$.

**Case** $\varepsilon \ge \frac{8L_0}{L_1}$, **optimal tuning:** We use stepsize $\gamma = \frac{1}{48 L_1 d \log\frac{1}{\delta}} \Rightarrow 80 L_0 d\gamma \log(1/\delta) \le \varepsilon/2$ and batchsize $8\frac{\|\vec{\sigma}\|_1}{B^{\frac{\kappa-1}{\kappa}}} \le \varepsilon/2 \Rightarrow B_k \equiv \max\left\{1, \left(\frac{16\|\vec{\sigma}\|_1}{\varepsilon}\right)^{\frac{\kappa}{\kappa-1}}\right\}$. The number of iterations $T$ is chosen to bound the first term:

$$\frac{4\Delta}{T\gamma} = \frac{192\Delta L_1 \log\frac{1}{\delta}d}{T} \le \frac{\varepsilon}{2} \Rightarrow T = O\left(\frac{\Delta L_1 \log\frac{1}{\delta}d}{\varepsilon}\right).$$

The total number of oracle calls is:

$$\varepsilon \geq \frac{8L_0}{L_1} \quad \Rightarrow \quad N = O\left(\frac{\Delta L_1 \log(1/\delta)d}{\varepsilon}\left[1 + \left(\frac{\|\vec{\sigma}\|_1}{\varepsilon}\right)^{\frac{\kappa}{\kappa-1}}\right]\right).$$

**Case** $\varepsilon < \frac{8L_0}{L_1}$**, parameter-free tuning:** We use parameters $T, \gamma_k = \frac{\gamma_0}{\sqrt{T}}, B_k = \max\{1, B_0 T\}$ to get:

$$\frac{1}{T}\sum_{k=1}^{T}\|\nabla f(x^k)\|_1 \leq \frac{4\Delta}{\sqrt{T}\gamma_0} + 80\frac{L_0 d\gamma_0}{\sqrt{T}}\log(1/\delta) + 8\frac{\|\vec{\sigma}\|_1}{B_0^{\frac{\kappa-1}{\kappa}}T^{\frac{\kappa-1}{\kappa}}} + 24\frac{d\|\nabla f(x^1)\|_1}{T}\log(1/\delta).$$

Setting such $T$ that the first two terms become less than $\varepsilon$, we obtain the final complexity $N = T \cdot B_0 T$.

**Case** $\varepsilon < \frac{8L_0}{L_1}$**, optimal tuning:** We set the same batchsize $8\frac{\|\vec{\sigma}\|_1}{B^{\frac{\kappa-1}{\kappa}}} \leq \varepsilon/2 \Rightarrow B_k \equiv \max\left\{1, \left(\frac{16\|\vec{\sigma}\|_1}{\varepsilon}\right)^{\frac{\kappa}{\kappa-1}}\right\}$. The stepsize $\gamma$ is set to minimize the sum:

$$\min_{\gamma}\left[\frac{4\Delta}{T\gamma} + 80L_0 d\gamma\log(1/\delta)\right] = 2\sqrt{\frac{320\Delta L_0 d\log(1/\delta)}{T}},$$

it means that the stepsize $\gamma = \sqrt{\frac{4\Delta}{80TL_0\log(1/\delta)d}}$. The number of iterations $T$ is chosen to satisfy

$$2\sqrt{\frac{320\Delta L_0\log(1/\delta)d}{T}} \leq \frac{\varepsilon}{2} \Rightarrow T = O\left(\frac{\Delta L_0\log(1/\delta)d}{\varepsilon^2}\right).$$

We only need to check whether condition $\gamma \leq \frac{1}{48L_1 d\log\frac{1}{\delta}}$ holds:

$$\begin{aligned}
\gamma &= \sqrt{\frac{4\Delta}{80TL_0\log(1/\delta)d}} = \sqrt{\frac{4\Delta}{T}\frac{1}{80L_0\log(1/\delta)d}} \\
&\leq \frac{\varepsilon}{4}\frac{1}{80L_0\log(1/\delta)d} \leq \frac{8L_0}{4L_1}\frac{1}{80L_0\log(1/\delta)d} \\
&\leq \frac{1}{48L_1 d\log\frac{1}{\delta}}.
\end{aligned}$$

Hence, we have the following bound for sample complexity

$$\varepsilon < \frac{8L_0}{L_1} \quad \Rightarrow \quad N = O\left(\frac{\Delta L_0\log(1/\delta)d}{\varepsilon^2}\left[1 + \left(\frac{\|\vec{\sigma}\|_1}{\varepsilon}\right)^{\frac{\kappa}{\kappa-1}}\right]\right). \tag{21}$$

$\square$

*Proof.* Proof of Parameter-free Tuning Theorem 5 First, we derive upper bound for new $\min$ metric with non-constant parameters:

$$\begin{aligned}
\min_{k\in\overline{1,T}}\|\nabla f(x^k)\|_1 &\leq \frac{\sum_{k=1}^{T}\gamma_k\|\nabla f(x^k)\|_1}{\sum_{k=1}^{T}\gamma_k} = \frac{\Delta}{\sum_{k=1}^{T}\gamma_k} + L_0 d\frac{\sum_{k=1}^{T}\gamma_k^2}{\sum_{k=1}^{T}\gamma_k} + \frac{2\sum_{k=1}^{T}\gamma_k\|\vec{\sigma}\|_1/B_k^{\frac{\kappa-1}{\kappa}}}{\sum_{k=1}^{T}\gamma_k} \\
&+ 6d(\gamma^{max}\|\nabla f(x^1)\|_1 + 2C_T L_0)\frac{\log(1/\delta)}{\sum_{k=1}^{T}\gamma_k}.
\end{aligned}$$

**Case $\varepsilon \geq \frac{8L_0}{L_1}$, optimal tuning.** If we consider only first $T \leq \frac{64\Delta L_1^\delta L_1 d^2}{L_0}$ steps with constant stepsizes $\gamma_k = \frac{1}{48L_1^\delta d}$ and increasing batchsizes $B_k = (16k)^{\frac{\kappa}{\kappa-1}}$, we get

$$
\sum_{k=1}^T \gamma_k = \frac{T}{48L_1^\delta d}, \sum_{k=1}^T \gamma_k^2 = \frac{T}{(48L_1^\delta d)^2}, \gamma^{max} = \frac{1}{48L_1^\delta d}, C_T = \frac{T}{(48L_1^\delta d)^2},
$$

$$
\sum_{k=1}^T \frac{1}{B_k^{\frac{\kappa-1}{\kappa}}} = \sum_{k=1}^T \frac{1}{16k} \leq \frac{\ln T}{16},
$$

$$
\min_{k\in\overline{1,T}} \|\nabla f(x^k)\|_1 \leq \frac{48\Delta L_1^\delta d}{T} + \frac{24L_0^\delta d}{48L_1^\delta d} + \frac{2\|\vec{\sigma}\|_1}{T}\frac{\ln T}{16} \leq \varepsilon.
$$

The term $\frac{24L_0^\delta d}{48L_1^\delta d} \leq \frac{\varepsilon}{16}$ is bounded by condition, and the number of iterations $T = \tilde{O}\left(\frac{(\Delta L_1^\delta d + \|\vec{\sigma}\|_1)}{\varepsilon}\right)$ is enough to bound the other terms. The total sample complexity is

$$
\sum_{k=1}^T B_k = \sum_{k=1}^T (16k)^{\frac{\kappa}{\kappa-1}} \leq (16T)^{\frac{2\kappa-1}{\kappa-1}} = \tilde{O}\left(\left(\frac{(\Delta L_1^\delta d + \|\vec{\sigma}\|_1)}{\varepsilon}\right)^{\frac{2\kappa-1}{\kappa-1}}\right). \tag{22}
$$

**Case $\varepsilon \ll \frac{8L_0}{L_1}$, optimal tuning.** In this case, the first $\frac{64L_1 L_1^\delta d^2}{L_0}$ steps can be neglected, and we use decreasing stepsizes $\gamma_k = \sqrt{\frac{\Delta}{20dL_0^\delta k}}$ and increasing batchsizes $B_k = (16k)^{\frac{\kappa}{2(\kappa-1)}}$ to get

$$
\sum_{k=1}^T \gamma_k = 2\sqrt{\frac{\Delta T}{20L_0^\delta d}}, \sum_{k=1}^T \gamma_k^2 = \frac{\Delta \ln T}{20L_0^\delta d}, \gamma^{max} = \frac{1}{48L_1^\delta d}, C_T = \frac{\Delta}{20L_0^\delta d},
$$

$$
\sum_{k=1}^T \frac{\gamma_k}{B_k^{\frac{\kappa-1}{\kappa}}} = \sqrt{\frac{\Delta}{20L_0^\delta d}}\sum_{k=1}^T \frac{1}{\sqrt{k}}\frac{1}{4\sqrt{k}} \leq \sqrt{\frac{\Delta}{256L_0^\delta d}}\ln T,
$$

$$
\min_{k\in\overline{1,T}} \|\nabla f(x^k)\|_1 \leq \sqrt{\frac{80L_0^\delta d}{T}} + \sqrt{\frac{\Delta L_0^\delta d}{20T}}\ln T + 2\frac{\|\vec{\sigma}\|_1 \ln T}{\sqrt{T}} + \sqrt{\frac{\Delta L_0^\delta d}{20T}} \leq \varepsilon.
$$

Hence, the number of iterations $T = \tilde{O}\left(\frac{(\Delta L_0^\delta d + \|\vec{\sigma}\|_1^2)}{\varepsilon^2}\right)$ is enough to bound the sum. The total sample complexity is

$$
\sum_{k=1}^T B_k = \sum_{k=1}^T (16k)^{\frac{\kappa}{2(\kappa-1)}} \leq (16T)^{\frac{3\kappa-2}{2(\kappa-1)}} = \tilde{O}\left(\left(\frac{(\Delta L_0^\delta d + \|\vec{\sigma}\|_1^2)}{\varepsilon^2}\right)^{\frac{3\kappa-2}{2(\kappa-1)}}\right). \tag{23}
$$

**Case $\varepsilon \geq \frac{8L_0}{L_1}$, parameter-free tuning.** If we consider only first $T$ steps until plateau $\frac{8L_0}{L_1}$, we use constant stepsizes $\gamma_k = \gamma_0 \leq \frac{1}{48L_1^\delta d}$ and increasing batchsizes $B_k = B_0 k^2$ to get

$$
\sum_{k=1}^T \gamma_k = T\gamma_0, \sum_{k=1}^T \gamma_k^2 = T\gamma_0^2, \gamma^{max} = \gamma_0, C_T = T\gamma_0,
$$

$$
\sum_{k=1}^T \frac{1}{B_k^{\frac{\kappa-1}{\kappa}}} = \sum_{k=1}^T \frac{1}{(\sqrt{B_0}k)^{\frac{2(\kappa-1)}{\kappa}}} \leq \frac{T^{\frac{2-\kappa}{\kappa}}\ln T}{B_0^{\frac{\kappa-1}{\kappa}}},
$$

$$
\min_{k\in\overline{1,T}} \|\nabla f(x^k)\|_1 \leq \frac{\Delta}{\gamma_0 T} + 24L_0^\delta d\gamma_0 + \frac{2\|\vec{\sigma}\|_1}{(T\sqrt{B_0})^{\frac{2(\kappa-1)}{\kappa}}}\ln T \leq \varepsilon.
$$

The term $24L_0^\delta d\gamma_0 \leq \frac{\varepsilon}{16}$ is bounded by condition, and the number of iterations $T = \tilde{O}\left(\left(\frac{\Delta}{\gamma_0\varepsilon}\right) + \frac{1}{B_0}\left(\frac{\|\vec{\sigma}\|_1}{\varepsilon}\right)^{\frac{\kappa}{2(\kappa-1)}}\right)$ is enough to bound the other terms. The total sample complexity is

$$
\sum_{k=1}^T B_k = \sum_{k=1}^T B_0 k^2 \leq B_0 T^3 = \tilde{O}\left(B_0\left(\frac{\Delta}{\gamma_0\varepsilon}\right)^3 + \frac{1}{B_0^2}\left(\frac{\|\vec{\sigma}\|_1}{\varepsilon}\right)^{\frac{3\kappa}{2(\kappa-1)}}\right).
$$

**Case** $\varepsilon \ll \frac{8L_0}{L_1}$**, parameter-free tuning.** In this case, the first $\frac{64\Delta L_1 L_1^\delta d^2}{L_0}$ steps can be neglected, and we use decreasing stepsizes $\gamma_k = \frac{\gamma_0}{\sqrt{k}}$ and increasing batchsizes $B_k = B_0 k$ to get

$$
\sum_{k=1}^{T} \gamma_k = \gamma_0\sqrt{T}, \sum_{k=1}^{T} \gamma_k^2 = \gamma_0^2 \ln T, \gamma^{max} = \gamma_0, C_T = \gamma_0^2,
$$

$$
\sum_{k=1}^{T} \frac{\gamma_k}{B_k^{\frac{\kappa-1}{\kappa}}} = \frac{\gamma_0}{B_0^{\frac{\kappa-1}{\kappa}}} \sum_{k=1}^{T} \frac{1}{k^{\frac{3\kappa-2}{2\kappa}}} \le \frac{\gamma_0}{B_0^{\frac{\kappa-1}{\kappa}}} T^{\frac{2-\kappa}{2\kappa}} \ln T,
$$

$$
\min_{k \in \overline{1,T}} \|\nabla f(x^k)\|_1 \le \frac{\Delta}{\gamma_0\sqrt{T}} + L_0^\delta d\gamma_0 \frac{\ln T}{\sqrt{T}} + \frac{\|\vec{\sigma}\|_1 \ln T}{B_0^{\frac{\kappa-1}{\kappa}} T^{\frac{\kappa-1}{\kappa}}} \le \varepsilon.
$$

Hence, the number of iterations $T = \tilde{O}\left(\frac{(L_0^\delta \gamma_0 d + \Delta/\gamma_0)^2}{\varepsilon^2} + \frac{1}{B_0}\left(\frac{\|\vec{\sigma}\|_1}{\varepsilon}\right)^{\frac{\kappa}{\kappa-1}}\right)$ is enough to bound the sum. The total sample complexity is

$$
\sum_{k=1}^{T} B_k = \sum_{k=1}^{T} B_0 k \le B_0 T^2 = \tilde{O}\left(\frac{B_0(L_0^\delta \gamma_0 d + \Delta/\gamma_0)^4}{\varepsilon^4} + \frac{1}{B_0}\left(\frac{\|\vec{\sigma}\|_1}{\varepsilon}\right)^{\frac{2\kappa}{\kappa-1}}\right). \tag{24}
$$

$\square$

### B.5 Proof of MajorityVote-SignSGD Complexity Theorem 2

We start this section with a general lemma on convergence of MajorityVote-SignSGD. The proof of Theorem 2 is located after the lemma and divided into two parts: for finite horizon with optimal tuning (Theorem 8) and for infinite horizon with parameter-free tuning (Theorem 6).

**Lemma 7** (MajorityVote-SignSGD **Convergence Lemma**)**.** *Consider lower-bounded $(L_0, L_1)$-smooth function $f$ (As. 1, 2) and HT **unimodal and symmetric** gradient estimates $\kappa > 0$ (As. 3). Then Alg. 3 after $T$ iterations with non-increasing stepsizes $\gamma_k \le 1/(48L_1 d \log\frac{1}{\delta})$ and batchsizes $M_k \ge 160/\kappa^2$ achieves with probability at least $1 - \delta$:*

$$
\sum_{k=1}^{T} \frac{\gamma_k}{16} \|\nabla f(x^k)\|_1 \le \Delta + L_0 d \sum_{k=1}^{T} \gamma_k^2 + 2\sum_{k=1}^{T} \gamma_k \frac{\|\vec{\sigma}\|_1}{\sqrt{M_k}} + 6d(\gamma_1\|\nabla f(x^1)\|_1 + 2C_T L_0)\log\frac{1}{\delta}, \tag{25}
$$

*where $C_T := \max_{k \in \overline{1,T}} \gamma_k \cdot \sum_{\tau=1}^{k-1} \gamma_\tau$ and $\Delta = f(x^1) - f^*$.*

*Proof.* The beginning of this proof exactly copies the proof of SignSGD Convergence Lemma (Appendix B.3) until equality (18). We have to estimate the probability of failure of majority voting for each coordinate $j$ conditioned on $x^k$, namely,

$$
\mathbb{P}\left(\text{sign}(\nabla f(x^k))_j \neq \text{sign}\left[\sum_{i=1}^{M_k} \text{sign}(g_i^k)\right]_j\right), \quad g_i^k = \nabla f(x^k, \xi_i^k).
$$

We use the generalized Gauss's Inequality about distribution of unimodal symmetric random variables (Dharmadhikari & Joag-Dev, 1986, Theorem 1).

**Lemma 8** (Gauss's Inequality)**.** *Let a random variable $\xi$ be unimodal symmetric with mode $\nu$ and bounded $\kappa$-th moment, $\kappa > 0$. Then the following bounds hold:*

$$
\mathbb{P}[|\xi - \nu| \ge \tau] \le \begin{cases} \left(\frac{\kappa}{\kappa+1}\right)^\kappa \frac{\mathbb{E}[|\xi-\nu|]^\kappa}{\tau^\kappa}, & \tau^\kappa \ge \frac{\kappa^\kappa}{(\kappa+1)^{\kappa-1}} \cdot \mathbb{E}[|\xi - \nu|^\kappa], \\ 1 - \left[\frac{\tau^\kappa}{(\kappa+1)\mathbb{E}[|\xi-\nu|]^\kappa}\right]^{\frac{1}{\kappa}}, & \tau^\kappa \le \frac{\kappa^\kappa}{(\kappa+1)^{\kappa-1}} \cdot \mathbb{E}[|\xi - \nu|^\kappa]. \end{cases}
$$

We use Gauss's Inequality for each variable $g_{i,j}^k = \nabla f(x^k, \xi_i^k)_j$ satisfying the symmetry requirement from the theorem's statement. We denote $S_j := \frac{|\nabla f(x^k)_j|}{\sigma_j}$ and bound

$$
\begin{aligned}
\mathbb{P}\left[\text{sign}(\nabla f(x^k)_j) \neq \text{sign}(g_{i,j}^k)\right] &= \mathbb{P}\left[g_{i,j}^k - \nabla f(x^k)_j \geq |\nabla f(x^k)_j|\right] \\
&= \frac{1}{2}\mathbb{P}\left[|g_{i,j}^k - \nabla f(x^k)_j| \geq |\nabla f(x^k)_j|\right] \\
&\leq \begin{cases} \frac{1}{2}\left(\frac{\kappa}{\kappa+1}\right)^\kappa \frac{\sigma_j^\kappa}{|\nabla f(x^k)_j|^\kappa}, & |\nabla f(x^k)_j|^\kappa \geq \frac{\kappa^\kappa}{(\kappa+1)^{\kappa-1}} \cdot \sigma_j^\kappa, \\ \frac{1}{2} - \frac{1}{2}\left[\frac{|\nabla f(x^k)_j|^\kappa}{(\kappa+1)\sigma_j^\kappa}\right]^{\frac{1}{\kappa}}, & |\nabla f(x^k)_j|^\kappa \leq \frac{\kappa^\kappa}{(\kappa+1)^{\kappa-1}} \cdot \sigma_j^\kappa, \end{cases} \\
&\leq \begin{cases} \frac{1}{2}\left(\frac{\kappa}{\kappa+1}\right)^\kappa \frac{1}{S_j^\kappa}, & S_j^\kappa \geq \frac{\kappa^\kappa}{(\kappa+1)^{\kappa-1}}, \\ \frac{1}{2} - \frac{1}{2}\frac{S_j}{(\kappa+1)^{\frac{1}{\kappa}}}, & S_j^\kappa \leq \frac{\kappa^\kappa}{(\kappa+1)^{\kappa-1}}, \end{cases}
\end{aligned}
$$

We denote probability of failure of a single estimate by

$$
\begin{aligned}
q_j &:= \mathbb{P}\left[\text{sign}(\nabla f(x^k)_j) \neq \text{sign}(g_{i,j}^k)\right] \\
&\leq \begin{cases} \frac{1}{2}\left(\frac{\kappa}{\kappa+1}\right)^\kappa \frac{1}{S_j^\kappa}, & S_j^\kappa \geq \frac{\kappa^\kappa}{(\kappa+1)^{\kappa-1}}, \\ \frac{1}{2} - \frac{1}{2}\frac{S_j}{(\kappa+1)^{\frac{1}{\kappa}}}, & S_j^\kappa \leq \frac{\kappa^\kappa}{(\kappa+1)^{\kappa-1}}, \end{cases} \\
&=: \tilde{q}_j(S_j). \tag{26}
\end{aligned}
$$

Moreover, this probability $q_j \leq \tilde{q}_j(S_j) < \frac{1}{2}$, and the deviation of $q_j$ from $\frac{1}{2}$ can be bounded by

$$
\varepsilon_j := \frac{1}{2} - q_j \leq \frac{1}{2} - \tilde{q}_j(S_j) =: \tilde{\varepsilon}_j(S_j).
$$

The probability of getting the wrong sign can be restated as the probability of failing half out of $M_k$ Bernoulli trials with fail probability $q_j$:

$$
\mathbb{P}\left[\text{sign}(\nabla f(x^k)_j) \neq \text{sign}\left[\sum_{i=1}^{M_k} \text{sign}(g_{i,j}^k)\right]\right] \leq \frac{1}{1 + \frac{M_k}{\frac{1}{4\varepsilon_j^2} - 1}} < \frac{1}{1 + \frac{M_k}{\frac{1}{4\tilde{\varepsilon}_j^2(S_j)} - 1}}. \tag{27}
$$

- First, we consider the case $S_j \geq \frac{\kappa}{(\kappa+1)^{\frac{\kappa-1}{\kappa}}}$:

$$
\tilde{\varepsilon}_j^2(S_j) = \left(\frac{1}{2} - \frac{1}{2}\left(\frac{\kappa}{\kappa+1}\right)^\kappa \frac{1}{S_j^\kappa}\right)^2 \geq \frac{1}{4}\frac{\kappa^2}{(\kappa+1)^2},
$$

$$
\frac{1}{4\tilde{\varepsilon}_j^2(S_j)} - 1 \leq \frac{(\kappa+1)^2}{\kappa^2} - 1 \leq \frac{5}{\kappa^2}.
$$

If we set $M_k \geq \frac{160}{\kappa^2}$, then the fail probability is upper bounded by

$$
\mathbb{P}\left[\text{sign}(\nabla f(x^k)_j) \neq \text{sign}\left[\sum_{i=1}^{M_k} \text{sign}(g_{i,j}^k)\right]\right] < \frac{1}{1 + \frac{M_k}{\frac{1}{4\tilde{\varepsilon}_j^2(S_j)} - 1}} \leq \frac{1}{32}. \tag{28}
$$

- For the case $S_j < \frac{\kappa}{(\kappa+1)^{\frac{\kappa-1}{\kappa}}}$, we derive the bound:

$$
\frac{1}{4\tilde{\varepsilon}_j^2(S_j)} - 1 = \frac{(\kappa+1)^{\frac{2}{\kappa}}}{S_j^2} - 1 \leq \frac{4}{S_j^2}. \tag{29}
$$

And we use the inequality $\frac{1}{1+x^2} \leq \frac{1}{2x}, x > 0$ on (27):

$$
(27) \leq \frac{\sqrt{\frac{1}{4\tilde{\varepsilon}_j^2(S_j)} - 1}}{2\sqrt{M_k}} \leq \frac{1}{\sqrt{M_k}} \cdot \frac{1}{S_j}. \tag{30}
$$

Combining (28) and (30) together, we obtain the bound for each coordinate:

$$\mathbb{P}\left[\text{sign}(\nabla f(x^k)_j) \neq \text{sign}\left[\sum_{i=1}^{M_k}\text{sign}(g_{i,j}^k)\right]\right] \leq \frac{1}{32} + \frac{1}{\sqrt{M_k}}\cdot\frac{1}{S_j} = \frac{1}{32} + \frac{1}{\sqrt{M_k}}\frac{\sigma_j}{|\nabla f(x^k)_j|}. \quad (31)$$

The rest of this proof is copying the proof of SignSGD Convergence Lemma (Appendix B.3) until the equality (18). There we replace probability of single estimate with the majority voting and obtain:

$$\sum_{j=1}^{d}|\nabla f(x^k)|_j\cdot\mathbb{P}\left[\text{sign}(\nabla f(x^k)_j) \neq \text{sign}\left[\sum_{i=1}^{M_k}\text{sign}(g_{i,j}^k)\right]\right] \leq \frac{\|\nabla f(x^k)\|_1}{32} + \frac{\|\vec{\sigma}\|_1}{\sqrt{M_k}}$$

instead of

$$\sum_{j=1}^{d}|\nabla f(x^k)|_j\cdot\mathbb{P}(\text{sign}([\nabla f(x^k)])_j \neq [\text{sign}(g^k)]_j) \leq \frac{\|\vec{\sigma}\|_1}{B_k^{\frac{\kappa-1}{\kappa}}}.$$

Hence, the final bound on the sum of $\ell_1$-norm of gradients with probability at least $1 - \delta$ is

$$\frac{1}{16}\sum_{k=1}^{T}\gamma_k\|\nabla f(x^k)\|_1 \leq \Delta + L_0 d\sum_{k=1}^{T}\gamma_k^2 + 2\sum_{k=1}^{T}\gamma_k\frac{\|\vec{\sigma}\|_1}{\sqrt{M_k}}\frac{1}{32}\sum_{k=1}^{T}\gamma_k\|\nabla f(x^k)\|_1$$

$$+ 6d(\gamma^{max}\|\nabla f(x^1)\|_1 + 2C_T L_0)\log(1/\delta), \quad M_k \geq \frac{160}{\kappa^2}.$$

$\square$

**Theorem 8** (**HP complexity for** MajorityVote-SignSGD, **finite horizon**). *Consider lower-bounded $(L_0, L_1)$-smooth function $f$ (As. 1, 2) and the gradient estimates corrupted by **unimodal and symmetric HT noise** with $\kappa > 0$ (As. 3). Then Alg. 3 requires the sample complexity $N$ to achieve $\frac{1}{T}\sum_{k=1}^{T}\|\nabla f(x^k)\|_1 \leq \varepsilon$ with probability at least $1 - \delta$ for:*

***Optimal tuning for*** $\varepsilon > \frac{8L_0}{L_1}$**:** $T = O\left(\frac{\Delta L_1^{\delta} d}{\varepsilon}\right), \gamma_k \equiv \frac{1}{48 L_1^{\delta} d}, M_k \equiv \max\left\{\frac{160}{\kappa^2}, \frac{2^{16}\|\vec{\sigma}\|_1^2}{\varepsilon^2}\right\}$:

$$N = O\left(\frac{\Delta L_1^{\delta} d}{\varepsilon}\left[\frac{1}{\kappa^2} + \left(\frac{\|\vec{\sigma}\|_1}{\varepsilon}\right)^2\right]\right),$$

***Optimal tuning for*** $\varepsilon \leq \frac{8L_0}{L_1}$**:** $T = O\left(\frac{\Delta L_0^{\delta} d}{\varepsilon^2}\right), \gamma_k \equiv \sqrt{\frac{\Delta}{80 L_0^{\delta} dT}}, M_k \equiv \max\left\{\frac{160}{\kappa^2}, \frac{2^{16}\|\vec{\sigma}\|_1^2}{\varepsilon^2}\right\}$:

$$N = O\left(\frac{\Delta L_0^{\delta} d}{\varepsilon^2}\left[\frac{1}{\kappa^2} + \left(\frac{\|\vec{\sigma}\|_1}{\varepsilon}\right)^2\right]\right),$$

***Parameter-free tuning for*** $\varepsilon \leq \frac{8L_0}{L_1}$**:** $T, \gamma_k \equiv \frac{\gamma_0}{\sqrt{T}}, M_k \equiv \max\{160/\kappa^2, M_0 T\}$:

$$N = O\left(\frac{M_0(\Delta/\gamma_0 + L_0^{\delta} d\gamma_0)^4}{\varepsilon^4} + \frac{1}{M_0}\left(\frac{\|\vec{\sigma}\|_1}{\varepsilon}\right)^4\right),$$

***Parameter-free tuning for*** $\varepsilon \geq \frac{8L_0}{L_1}$**:** $T, \gamma_k \equiv \gamma_0 \leq \frac{1}{48 L_1^{\delta} d}, M_k \equiv \max\{160/\kappa^2, M_0 T^2\}$:

$$N = O\left(M_0\left(\frac{\Delta}{\varepsilon\gamma_0}\right)^3 + \frac{1}{M_0^2}\left(\frac{\|\vec{\sigma}\|_1}{\varepsilon}\right)^3\right),$$

*where $\Delta = f(x^1) - f^*, L_0^{\delta} = L_0\log(1/\delta), L_1^{\delta} = L_1\log(1/\delta)$.*

*Proof.* Plugging in constant stepsizes $\gamma_k \equiv \gamma$ implies $C_T = T\gamma^2, \gamma^{max} = \gamma$ into the bound (25) from Convergence Lemma 7, we have :

$$\frac{1}{2T}\sum_{k=1}^{T}\|\nabla f(x^k)\|_1 \leq \frac{16\Delta}{T\gamma} + 192 L_0 d\gamma\log(1/\delta) + \frac{32\|\vec{\sigma}\|_1}{\sqrt{M_k}} + 96\frac{d\|\nabla f(x^1)\|_1}{T}\log(1/\delta). \quad (32)$$

**Case** $\varepsilon > \frac{8L_0}{L_1}$**, parameter-free tuning:** We use parameters $T, \gamma_k = \gamma_0, M_k = \max\{160/\kappa^2, M_0 T^2\}$ to get:

$$\frac{1}{T}\sum_{k=1}^{T}\|\nabla f(x^k)\|_1 \leq \frac{16\Delta}{T\gamma_0} + 32\frac{\|\vec{\sigma}\|_1}{\sqrt{M_0 T}} + 96\frac{d\|\nabla f(x^1)\|_1}{T}\log(1/\delta).$$

Setting such $T$ that the first two terms become less than $\varepsilon$, we obtain the final complexity $N = T \cdot M_0 T^2$.

**Case** $\varepsilon \geq \frac{8L_0}{L_1}$**, optimal tuning:** We use stepsize $\gamma = \frac{1}{400 L_1 d \log\frac{1}{\delta}} \Rightarrow 192 L_0 d\gamma \log(1/\delta) \leq \varepsilon/2$ and batchsize $32\frac{\|\vec{\sigma}\|_1}{\sqrt{M_k}} \leq \varepsilon/4 \Rightarrow M_k \equiv \max\left\{\frac{160}{\kappa^2}, \left(\frac{128\|\vec{\sigma}\|_1}{\varepsilon}\right)^2\right\}$. The number of iterations $T$ is chosen to bound the first term:

$$\frac{16\Delta}{T\gamma} = \frac{2560\Delta L_1 \log\frac{1}{\delta} d}{T} \leq \frac{\varepsilon}{4} \Rightarrow T = O\left(\frac{\Delta L_1 \log\frac{1}{\delta} d}{\varepsilon}\right).$$

The total number of oracle calls is:

$$N = O\left(\frac{\Delta L_1 \log(1/\delta) d}{\varepsilon}\left[\frac{1}{\kappa^2} + \left(\frac{\|\vec{\sigma}\|_1}{\varepsilon}\right)^2\right]\right).$$

**Case** $\varepsilon < \frac{8L_0}{L_1}$**, parameter-free tuning:** We use parameters $T, \gamma_k = \frac{\gamma_0}{\sqrt{T}}, M_k = \max\{160/\kappa^2, M_0 T\}$ to get:

$$\frac{1}{T}\sum_{k=1}^{T}\|\nabla f(x^k)\|_1 \leq \frac{16\Delta}{\sqrt{T}\gamma_0} + 192\frac{L_0 d\gamma_0}{\sqrt{T}}\log(1/\delta) + 32\frac{\|\vec{\sigma}\|_1}{\sqrt{M_0 T}} + 96\frac{d\|\nabla f(x^1)\|_1}{T}\log(1/\delta).$$

Setting such $T$ that the first two terms become less than $\varepsilon$, we obtain the final complexity $N = T \cdot M_0 T$.

**Case** $\varepsilon < \frac{8L_0}{L_1}$**, optimal tuning:** We set the same batchsize $32\frac{\|\vec{\sigma}\|_1}{\sqrt{M_k}} \leq \varepsilon/4 \Rightarrow M_k \equiv \max\left\{\frac{160}{\kappa^2}, \left(\frac{128\|\vec{\sigma}\|_1}{\varepsilon}\right)^2\right\}$. The stepsize $\gamma$ is set to minimize the sum:

$$\min_{\gamma}\left[\frac{16\Delta}{T\gamma} + 192 L_0 d\gamma \log(1/\delta)\right] = 2\sqrt{\frac{3200\Delta L_0 d \log(1/\delta)}{T}},$$

it means that the stepsize $\gamma = \sqrt{\frac{4\Delta}{80 T L_0 \log(1/\delta) d}}$. The number of iterations $T$ is chosen to satisfy

$$2\sqrt{\frac{3200\Delta L_0 \log(1/\delta) d}{T}} \leq \frac{\varepsilon}{2} \Rightarrow T = O\left(\frac{\Delta L_0 \log(1/\delta) d}{\varepsilon^2}\right).$$

We only need to check whether condition $\gamma \leq \frac{1}{48 L_1 d \log\frac{1}{\delta}}$ holds:

$$\begin{aligned}
\gamma &= \sqrt{\frac{4\Delta}{80 T L_0 \log(1/\delta) d}} = \sqrt{\frac{4\Delta}{T}\frac{1}{80 L_0 \log(1/\delta) d}}\\
&\leq \frac{\varepsilon}{4}\frac{1}{80 L_0 \log(1/\delta) d} \leq \frac{8L_0}{4L_1}\frac{1}{80 L_0 \log(1/\delta) d}\\
&\leq \frac{1}{48 L_1 d \log\frac{1}{\delta}}.
\end{aligned}$$

Hence, we have the following bound for sample complexity

$$N = O\left(\frac{\Delta L_0 \log(1/\delta) d}{\varepsilon^2}\left[\frac{1}{\kappa^2} + \left(\frac{\|\vec{\sigma}\|_1}{\varepsilon}\right)^2\right]\right). \tag{33}$$

$\square$

## B.6  PROOF OF M-SignSGD COMPLEXITY THEOREM 3

**Theorem 9** (**Complexity for** M-SignSGD **in expectation, full version**). *Consider lower-bounded $(L_0, L_1)$-smooth function $f$ (As. 1, 2) and HT gradient estimates (As. 3). Then Alg. 4 requires $T$ iterations to achieve $\frac{1}{T} \sum_{k=1}^{T} \mathbb{E}\left[\|\nabla f(x^k)\|_1\right] \leq \varepsilon$ starting with $\Delta = f(x^1) - f^*$:*

***Optimal tuning for*** $\varepsilon \geq \frac{3L_0}{L_1}$**:** $\beta_k \equiv 1 - \min\left\{1, \left(\frac{\Delta L_1 \sqrt{d}}{T \|\vec{\sigma}\|_1}\right)^{\frac{\kappa}{2\kappa - 1}}\right\}, \gamma_k \equiv \frac{1 - \beta_k}{8} \frac{1}{L_1 d}$

$$T = O\left(\frac{\Delta L_1 d}{\varepsilon}\left(1 + \left(\frac{\|\vec{\sigma}\|_1}{\varepsilon}\right)^{\frac{\kappa}{\kappa - 1}}\right)\right),$$

***Optimal tuning for*** $\varepsilon < \frac{3L_0}{L_1}$**:** $1 - \beta_k \equiv 1 - \min\left\{1, \left(\frac{\Delta L_0}{T \|\vec{\sigma}\|_1^2}\right)^{\frac{\kappa}{3\kappa - 2}}\right\}, \gamma_k \equiv \sqrt{\frac{\Delta(1 - \beta_k)}{T L_0 d}}$

$$T = O\left(\frac{\Delta L_0 d}{\varepsilon^2}\left(1 + \left(\frac{\|\vec{\sigma}\|_1}{\varepsilon}\right)^{\frac{\kappa}{\kappa - 1}}\right)\right),$$

***Parameter-free tuning for*** $\varepsilon \geq \frac{3L_0}{L_1}$**:** $T, \beta_k \equiv 1 - 1/T^{\frac{2}{3}}, \gamma_k \equiv \gamma_0(1 - \beta_k), \gamma_0 \leq 1/8dL_1$**:**

$$T = O\left(\left(\frac{\Delta}{\gamma_0 \varepsilon}\right)^3 + \left(\frac{\|\vec{\sigma}\|_1}{\varepsilon}\right)^{\frac{3\kappa}{2(\kappa - 1)}}\right),$$

***Parameter-free tuning for*** $\varepsilon < \frac{3L_0}{L_1}$**:** $T, \beta_k \equiv 1 - 1/\sqrt{T}, \gamma_k \equiv \gamma_0 T^{-\frac{3}{4}}$**:**

$$T = O\left(\frac{(\Delta/\gamma_0 + L_0 d \gamma_0)^4}{\varepsilon^4} + \left(\frac{\|\vec{\sigma}\|_1}{\varepsilon}\right)^{\frac{2\kappa}{\kappa - 1}}\right).$$

In this proof, we generalize the proof of Theorem 1 from (Sun et al., 2023) for HT noise.

*Proof.* Consider the $k$-th step of M-SignSGD. We use $(L_0, L_1)$ step update Lemma 5 to estimate:

$$
\begin{aligned}
f(x^{k+1}) - f(x^k) &\leq \langle \nabla f(x^k), x^{k+1} - x^k \rangle + \frac{L_0 + L_1 \|\nabla f(x^k)\|_2}{2} \exp(L_1 \|x^{k+1} - x^k\|_2) \|x^{k+1} - x^k\|_2^2 \\
&\leq -\gamma_k \|\nabla f(x^k)\|_1 + 2\gamma_k \|\epsilon^k\|_2 + \frac{L_0 d \gamma_k^2}{2} \exp(L_1 \sqrt{d} \gamma_k) \\
&+ \frac{L_1 d \gamma_k \exp(L_1 \sqrt{d} \gamma_k)}{2} \cdot \gamma_k \|\nabla f(x^k)\|_1.
\end{aligned}
\tag{34}
$$

Since we set constant steps sizes and momentum, we denote them as $\gamma \equiv \gamma_k$ and $\beta \equiv \beta_k$, respectively. We use notations $\epsilon^k := m^k - \nabla f(x^k)$ and $\theta^k := g^k - \nabla f(x^k)$. Therefore, we have at $k$-th step values:

$$
\begin{aligned}
m^k &= \beta m^{k-1} + (1 - \beta)g^k = \beta(\epsilon^{k-1} + \nabla f(x^{k-1})) + (1 - \beta)(\theta^k + \nabla f(x^k)), \\
\epsilon^k &= m^k - \nabla f(x^k) = \beta \epsilon^{k-1} + \beta(\underbrace{\nabla f(x^{k-1}) - \nabla f(x^k)}_{=:s^k}) + (1 - \beta)\theta^k, \\
\epsilon^k &= m^k - \nabla f(x^k) = \beta \epsilon^{k-1} + \beta s^k + (1 - \beta)\theta^k.
\end{aligned}
$$

Unrolling the recursion, we obtain an explicit formula (upper index of $\beta$ is its power):

$$
\epsilon^k = \beta^{k-1} \epsilon^1 + \sum_{i=2}^{k} \beta^{k-i+1} s^i + (1 - \beta) \sum_{i=2}^{k} \beta^{k-i} \theta^i.
\tag{35}
$$

From $(L_0, L_1)$−smoothness of the function $f$ (Lemma 2) follows the bound:

$$\|s^k\|_2 \leq (L_0 + L_1 \|\nabla f(x^k)\|_2) \exp(L_1 \|x^k - x^{x+1}\|_2) \|x^k - x^{k+1}\|_2 = (L_0 + L_1 \|\nabla f(x^k)\|_2) \exp(L_1 \gamma_k \sqrt{d}) \gamma_k \sqrt{d}$$

Denote $\lambda := \exp(L_1\gamma_k\sqrt{d})\gamma_k\sqrt{d}$. Hence, the norm of (35) can be bounded as:

$$\|\epsilon^k\|_1 \le \beta^{k-1}\|\epsilon^1\|_1 + L_0\lambda\sum_{i=2}^{k}\beta^{k-i+1} + L_1\lambda\sum_{i=2}^{k}\beta^{k-i+1}\|\nabla f(x^k)\|_1 + (1-\beta)\|\sum_{i=2}^{k}\beta^{k-i}\theta^i\|_1.$$

We notice that variables $\{\theta_i\}$ are martingale difference sequence from Lemma 4 which we plan to use. Due to the formal definition of $\theta^i = g^i - \nabla f(x^i) = \nabla f(x^i, \xi_i) - \nabla f(x^i)$ and M-SingSGD step, the conditioning on $\theta^{i-1}, \ldots, \theta^1$ with randomness $\xi_1, \ldots, \xi_{i-1}$ is equivalent to the conditioning on point s $x^i, \ldots, x^2$. Hence, we show by definition of martingale difference sequence that

$$\mathbb{E}[\theta^i|\theta^{i-1}, \ldots, \theta^1] = \mathbb{E}[\theta^i|x^i, \ldots, x^2] = \mathbb{E}[\nabla f(x^i, \xi_i) - \nabla f(x^i)|x^i, \ldots, x^2] = 0.$$

To take math expectation from both sides, we first take it from the term

$$\mathbb{E}\left[\|\sum_{i=2}^{k}\beta^{k-i}\theta^i\|_2\right] \le \left(\mathbb{E}\left[\|\sum_{i=2}^{k}\beta^{k-i}\theta^i\|_1^\kappa\right]\right)^{\frac{1}{\kappa}} \overset{\text{Lem. } 4}{\le} \left(\sum_{i=2}^{k}2\mathbb{E}\left[\|\beta^{(k-i)}\theta^i\|_1^\kappa\right]\right)^{\frac{1}{\kappa}} \le \left(\sum_{i=2}^{k}2\beta^{\kappa(k-i)}\mathbb{E}\left[\|\theta^i\|_1^\kappa\right]\right)^{\frac{1}{\kappa}}.$$

For each $i \in \overline{2, T}$, we estimate $\mathbb{E}\left[\|\theta^i\|_2^\kappa\right]$ as

$$\mathbb{E}\left[\|\theta^i\|_1^\kappa\right] \overset{(10)}{\le} \mathbb{E}\left[\|\theta^i\|_\kappa^\kappa\right] = \mathbb{E}\left[\sum_{j=1}^{d}|g_j^k - \nabla f(x^k)_j|^\kappa\right] \overset{As.3}{\le} \sum_{j=1}^{d}\sigma_j^\kappa = \|\vec{\sigma}\|_\kappa^\kappa. \tag{36}$$

We continue bounding (36) with

$$(36) \le \left(\sum_{i=2}^{k}2\beta^{\kappa(k-i)}\|\vec{\sigma}\|_1^\kappa\right)^{\frac{1}{\kappa}} \le \frac{2\|\vec{\sigma}\|_1}{(1-\beta^\kappa)^{\frac{1}{\kappa}}}.$$

Therefore, the final math expectation can be calculated as:

$$\mathbb{E}\|\epsilon^k\|_2 \le \beta^{k-1}\mathbb{E}\|\epsilon^1\|_2 + \frac{L_0\lambda\beta}{1-\beta} + L_1\lambda\sum_{i=2}^{k}\beta^{k-i+1}\mathbb{E}\|\nabla f(x^k)\|_2 + \frac{2(1-\beta)\|\vec{\sigma}\|_1}{(1-\beta^\kappa)^{\frac{1}{\kappa}}}. \tag{37}$$

Then, we take math expectation from (34):

$$\begin{aligned}
\mathbb{E}[f(x^{k+1})] - \mathbb{E}[f(x^k)] &\le -\gamma\mathbb{E}[\|\nabla f(x^k)\|_1] + 2\gamma\beta^{k-1}\mathbb{E}\|\epsilon^1\|_2 \\
&+ L_0\lambda\frac{2\gamma\beta}{1-\beta} + L_1\lambda 2\gamma\sum_{i=2}^{k}\beta^{k-i+1}\mathbb{E}\|\nabla f(x^k)\|_1 + \frac{4\gamma(1-\beta)\|\vec{\sigma}\|_1}{(1-\beta^\kappa)^{\frac{1}{\kappa}}} \\
&+ \frac{L_0\sqrt{d}\gamma}{2}\lambda + \frac{L_1\sqrt{d}\gamma}{2}\lambda\mathbb{E}\|\nabla f(x^k)\|_1.
\end{aligned}$$

Summing it over $k$, we derive

$$\begin{aligned}
f^* - f(x^1) &\le -\gamma\sum_{k=1}^{T}\mathbb{E}\|\nabla f(x^k)\|_1 + 2\gamma\sum_{k=1}^{T}\beta^{k-1}\mathbb{E}\|\epsilon^1\|_2 + L_0 T\lambda\frac{2\gamma\beta}{1-\beta} \\
&+ L_1\lambda 2\gamma\sum_{k=1}^{T}\sum_{i=2}^{k}\beta^{k-i+1}\mathbb{E}\|\nabla f(x^i)\|_1 + \frac{4\gamma T(1-\beta)\|\vec{\sigma}\|_1}{(1-\beta^\kappa)^{\frac{1}{\kappa}}} \\
&+ \frac{L_0 T\sqrt{d}\gamma}{2}\lambda + \frac{L_1\sqrt{d}\gamma}{2}\lambda\sum_{k=1}^{T}\mathbb{E}\|\nabla f(x^k)\|_1. \tag{38}
\end{aligned}$$

Changing the order of summation in the right part of (38), we obtain:

$$
2\gamma L_1 \lambda \sum_{k=1}^{T} \left( \sum_{i=2}^{k} \beta^{k-i+1} \mathbb{E}\|\nabla f(x^i)\|_1 \right) = 2\gamma L_1 \lambda \sum_{i=2}^{T} \left( \sum_{k=i}^{T} \beta^{k-i+1} \mathbb{E}\|\nabla f(x^i)\|_1 \right)
$$

$$
= 2\gamma L_1 \lambda \sum_{i=2}^{T} \beta^{-i} \left( \sum_{k=i}^{T} \beta^{k+1} \right) \mathbb{E}\|\nabla f(x^i)\|_1
$$

$$
= 2\gamma L_1 \lambda \sum_{i=2}^{T} \beta^{-i+1} \beta^i \left( \frac{1-\beta^{T-i}}{1-\beta} \right) \mathbb{E}\|\nabla f(x^i)\|_1
$$

$$
\leq 2\gamma L_1 \lambda \sum_{i=2}^{T} \beta \left( \frac{1}{1-\beta} \right) \mathbb{E}\|\nabla f(x^i)\|_1.
$$

Finally, we have the bound

$$
\begin{aligned}
f^* - f(x^1) &\leq -\gamma \sum_{k=1}^{T} \mathbb{E}\|\nabla f(x^k)\|_1 + \frac{2\gamma \mathbb{E}\|\epsilon^1\|_2}{1-\beta} \\
&+ 2\gamma L_1 \lambda \cdot \frac{\beta}{1-\beta} \sum_{k=1}^{T} \mathbb{E}\|\nabla f(x^k)\|_1 + \frac{4\gamma T(1-\beta)\|\vec{\sigma}\|_1}{(1-\beta^\kappa)^{1/\kappa}} \\
&+ \frac{L_0 T \sqrt{d}\gamma}{2}\lambda + L_0 T \lambda \frac{2\gamma\beta}{1-\beta} + \frac{L_1\sqrt{d}\gamma}{2}\lambda \sum_{k=1}^{T} \mathbb{E}\|\nabla f(x^k)\|_1 \\
&\leq \left( -\gamma + \frac{2\gamma L_1 \lambda \beta}{1-\beta} + \frac{L_1\sqrt{d}\gamma}{2}\lambda \right) \sum_{k=1}^{T} \mathbb{E}\|\nabla f(x^k)\|_1 \\
&+ \frac{2\gamma \mathbb{E}\|\epsilon^1\|_2}{1-\beta} + \frac{4\gamma T(1-\beta)\|\vec{\sigma}\|_1}{(1-\beta^\kappa)^{1/\kappa}} + \frac{L_0 T\sqrt{d}\gamma(1+3\beta)}{2(1-\beta)}\lambda. \qquad (39)
\end{aligned}
$$

Evaluate $\frac{L_0 T\sqrt{d}\gamma(1+3\beta)}{2(1-\beta)}\lambda \leq \frac{2L_0 T\sqrt{d}\gamma}{(1-\beta)}\lambda$. Let us set stepsize $\gamma$ such that

$$
\frac{2\gamma L_1 \lambda \sqrt{d}\beta}{1-\beta} + \frac{L_1\sqrt{d}\gamma}{2}\lambda \leq \frac{3\gamma^2 L_1 d \exp(L_1 d\gamma)}{1-\beta} \leq \gamma/2 \Rightarrow \gamma \leq \frac{1-\beta}{8}\frac{1}{L_1 d}.
$$

Thus, we obtain

$$
f^* - f(x^1) \leq -\frac{\gamma}{2} \sum_{k=1}^{T} \mathbb{E}\|\nabla f(x^k)\|_1 + \frac{2\gamma \mathbb{E}\|\epsilon^1\|_1}{1-\beta} + 4\gamma T(1-\beta)^{\frac{\kappa-1}{\kappa}}\|\vec{\sigma}\|_1 + \frac{L_0 T d\gamma^2 2}{(1-\beta)},
$$

$$
\frac{1}{T} \sum_{k=1}^{T} \mathbb{E}\|\nabla f(x^k)\|_1 \leq \frac{2(f^* - f(x^1))}{\gamma T} + \frac{4\mathbb{E}\|\epsilon^1\|_1}{T(1-\beta)} + 8(1-\beta)^{\frac{\kappa-1}{\kappa}}\|\vec{\sigma}\|_1 + \frac{4L_0 d\gamma}{(1-\beta)}. \qquad (40)
$$

**Case $\varepsilon \geq \frac{3L_0}{L_1}$, parameter-free tuning:** We set $1-\beta = \frac{1}{T^{\frac{2}{3}}}, \gamma = \frac{\gamma_0(1-\beta)}{16d^{\frac{3}{2}}}$, omit the the smaller terms and obtain

$$
\frac{1}{T} \sum_{k=1}^{T} \mathbb{E}\|\nabla f(x^k)\|_1 \leq \frac{32\Delta d^{\frac{3}{2}}}{\gamma_0 T^{\frac{1}{3}}} + \frac{16\|\vec{\sigma}\|_1}{T^{\frac{2(\kappa-1)}{3\kappa}}} + \frac{\varepsilon}{4}.
$$

Next, we choose $T$ to limit $\frac{32\Delta d^{\frac{3}{2}}}{\gamma_0 T^{\frac{1}{3}}} \leq \frac{\varepsilon}{2}$ and $\frac{16\|\vec{\sigma}\|_1}{T^{\frac{2(\kappa-1)}{3\kappa}}} \leq \frac{\varepsilon}{4}$:

$$
T = O\left( \left( \frac{\Delta d^{\frac{3}{2}}}{\gamma_0 \varepsilon} \right)^3 + \left( \frac{\|\vec{\sigma}\|_1}{\varepsilon} \right)^{\frac{3\kappa}{2(\kappa-1)}} \right).
$$

**Case $\varepsilon \geq \frac{3L_0}{L_1}$, optimal tuning:** We choose the stepsize $\gamma = \frac{1-\beta}{8}\frac{1}{L_1 d} \leq \frac{1-\beta}{8}\frac{1}{L_1 d}$ and get:

$$
\begin{aligned}
\frac{1}{T}\sum_{k=1}^{T}\mathbb{E}\|\nabla f(x^k)\|_1 &\leq \frac{16\Delta L_1 d}{T(1-\beta)} + \frac{4\mathbb{E}\|\epsilon^1\|_1}{T(1-\beta)} + 8(1-\beta)^{\frac{\kappa-1}{\kappa}}\|\vec{\sigma}\|_1 + \frac{4L_0}{L_1} \\
&\leq \frac{16(\Delta L_1 + \mathbb{E}\|\epsilon^1\|_1)d}{T(1-\beta)} + 8(1-\beta)^{\frac{\kappa-1}{\kappa}}\|\vec{\sigma}\|_1 + \frac{4\varepsilon}{3}.
\end{aligned}
$$

Then, we choose $1-\beta = \min\left\{1, \left(\frac{\Delta L_1\sqrt{d}}{T\|\vec{\sigma}\|_1}\right)^{\frac{\kappa}{2\kappa-1}}\right\}$ to obtain

$$
\min_{\beta\in[0,1)}\left[\frac{16\Delta L_1 d}{T(1-\beta)} + 8\sqrt{d}(1-\beta)^{\frac{\kappa-1}{\kappa}}\|\vec{\sigma}\|_1\right] \leq 24\sqrt{d}\left(\frac{\Delta L_1\sqrt{d}}{T}\right)^{\frac{\kappa-1}{2\kappa-1}}\|\vec{\sigma}\|_1^{\frac{\kappa}{2\kappa-1}} + \frac{24\Delta L_1 d}{T}. \tag{41}
$$

Finally, we choose number of iterations $T$ to get:

$$
24\left(\frac{\Delta L_1\sqrt{d}}{T}\right)^{\frac{\kappa-1}{2\kappa-1}}\|\vec{\sigma}\|_1^{\frac{\kappa}{2\kappa-1}} + \frac{24\Delta L_1 d}{T} \leq \varepsilon \Rightarrow T = O\left(\frac{\Delta L_1 d}{\varepsilon}\left(1 + \left(\frac{\|\vec{\sigma}\|_1}{\varepsilon}\right)^{\frac{\kappa}{\kappa-1}}\right)\right).
$$

**Case $\varepsilon \leq \frac{3L_0}{L_1}$, parameter-free tuning:** We set $1-\beta = \frac{1}{\sqrt{T}}, \gamma = \gamma_0 T^{-\frac{3}{4}}$ and obtain

$$
\frac{1}{T}\sum_{k=1}^{T}\mathbb{E}\|\nabla f(x^k)\|_1 \leq \frac{2\Delta}{\gamma_0 T^{\frac{1}{4}}} + \frac{4Ld\gamma_0}{T^{\frac{1}{4}}} + \frac{8\|\vec{\sigma}\|_1}{T^{\frac{\kappa-1}{2\kappa}}} + \frac{4\|\epsilon^1\|_1}{T^{\frac{1}{2}}}.
$$

Next, we choose $T$ to limit $\frac{2\Delta/\gamma_0 + 4Ld\gamma_0}{T^{\frac{1}{4}}} \leq \frac{\varepsilon}{2}$ and $\frac{8\|\vec{\sigma}\|_1}{T^{\frac{\kappa-1}{2\kappa}}} \leq \frac{\varepsilon}{2}$.

**Case $\varepsilon \leq \frac{3L_0}{L_1}$, optimal tuning:** We choose stepsize $\gamma = \sqrt{\frac{\Delta(1-\beta)}{2TL_0 d}}$ to minimize the sum

$$
\min_{\gamma}\left[\frac{2(f^* - f(x^1))}{\gamma T} + \frac{4L_0 d\gamma}{(1-\beta)}\right] = 4\sqrt{\frac{2\Delta L_0 d}{T(1-\beta)}},
$$

$$
\frac{1}{T}\sum_{k=1}^{T}\mathbb{E}\|\nabla f(x^k)\|_1 \leq \frac{4\mathbb{E}\|\epsilon^1\|_1}{T(1-\beta)} + 4\sqrt{\frac{2\Delta L_0 d}{T(1-\beta)}} + 8(1-\beta)^{\frac{\kappa-1}{\kappa}}\|\vec{\sigma}\|_1. \tag{42}
$$

The first term is much smaller than the second one, hence we omit it. Next, we choose $1-\beta = \min\left\{1, \left(\frac{\Delta L_0}{T\|\vec{\sigma}\|_1^2}\right)^{\frac{\kappa}{3\kappa-2}}\right\}$ to minimize the last two terms:

$$
\min_{\beta\in[0,1)}\left[4\sqrt{\frac{2\Delta L_0 d}{T(1-\beta)}} + 8(1-\beta)^{\frac{\kappa-1}{\kappa}}\|\vec{\sigma}\|_1\right] \leq 12\left(\frac{\Delta L_0}{T}\right)^{\frac{\kappa-1}{3\kappa-2}}\|\vec{\sigma}\|_1^{\frac{\kappa}{3\kappa-2}} + 12\sqrt{\frac{\Delta L_0 d}{T}}.
$$

Finally, we choose number of iterations $T$ to satisfy:

$$
12\left(\frac{\Delta L_0}{T}\right)^{\frac{\kappa-1}{3\kappa-2}}\|\vec{\sigma}\|_1^{\frac{\kappa}{3\kappa-2}} + 12\sqrt{\frac{\Delta L_0 d}{T}} \leq \frac{\varepsilon}{2} \Rightarrow T = O\left(\frac{\Delta L_0 d}{\varepsilon^2}\left(1 + \left(\frac{\|\vec{\sigma}\|_1}{\varepsilon}\right)^{\frac{\kappa}{\kappa-1}}\right)\right). \tag{43}
$$

We only need to check that

$$
\gamma = \sqrt{\frac{\Delta(1-\beta)}{TL_0 d}} = \sqrt{\frac{\Delta L_0 d}{T(1-\beta)}}\frac{(1-\beta)}{L_0 d} \leq \frac{\varepsilon}{2\cdot12}\frac{(1-\beta)}{L_0 d} \overset{\varepsilon\leq\frac{3L_0}{L_1}}{\leq} \frac{(1-\beta)}{L_1 d}.
$$

$\square$

### B.7 PROOF OF PARAMETER-FREE M-SignSGD COMPLEXITY THEOREM 4

*Proof.* We denote the error between momentum and gradient as $\epsilon^k := m^k - \nabla f(x^k)$, and the stochastic gradient noise as $\theta^k := g^k - \nabla f(x^k)$.

By the momentum update rule:

$$m^k = \beta_k m^{k-1} + (1 - \beta_k)g^k,$$

we can write:

$$\epsilon^k = m^k - \nabla f(x^k) = \beta_k \epsilon^{k-1} + \beta_k(\nabla f(x^{k-1}) - \nabla f(x^k)) + (1 - \beta_k)\theta^k.$$

Define $s^k := \nabla f(x^{k-1}) - \nabla f(x^k)$. Then,

$$\epsilon^k = \beta_k \epsilon^{k-1} + \beta_k s^k + (1 - \beta_k)\theta^k.$$

Unfolding the recursion yields:

$$\epsilon^k = \prod_{j=1}^{k} \beta_j \cdot \epsilon^0 + \sum_{i=1}^{k}\left(\prod_{j=i+1}^{k}\beta_j\right)\beta_i s^i + \sum_{i=1}^{k}\left(\prod_{j=i+1}^{k}\beta_j\right)(1 - \beta_i)\theta^i.$$

Let us denote the coefficients:

$$w_i := \left(\prod_{j=i+1}^{k}\beta_j\right), \quad \text{so that} \quad \epsilon^k = w_0 \epsilon^0 + \sum_{i=1}^{k} w_i \beta_i s^i + \sum_{i=1}^{k} w_i(1 - \beta_i)\theta^i.$$

From $(L_0, L_1)$-smoothness, the gradient difference satisfies:

$$\|s^k\|_2 \le (L_0 + L_1\|\nabla f(x^k)\|_2) \cdot \exp(L_1\|x^k - x^{k-1}\|_2) \cdot \|x^k - x^{k-1}\|_2.$$

Let us define:
$$\lambda_k := \exp(L_1 \gamma_k \sqrt{d}) \cdot \gamma_k \sqrt{d},$$

which leads to:

$$\|s^k\|_2 \le (L_0 + L_1\|\nabla f(x^k)\|_2) \cdot \lambda_k.$$

Now, using triangle inequality:

$$\|\epsilon^k\|_1 \le w_0\|\epsilon^0\|_1 + \sum_{i=1}^{k} w_i \beta_i \|s^i\|_1 + \left\|\sum_{i=1}^{k} w_i(1 - \beta_i)\theta^i\right\|_1.$$

We aim to estimate the following sum involving martingale difference sequence $\theta^i$:

$$\mathbb{E}\left[\left\|\sum_{i=1}^{k} w_i(1 - \beta_i)\theta^i\right\|_2\right], \quad \text{where } w_i := \prod_{j=i+1}^{k}\beta_j.$$

We note that $\{\theta^i\}$ is a martingale difference sequence, i.e.,

$$\mathbb{E}[\theta^i \mid \theta^{i-1}, \dots, \theta^1] = 0.$$

Hence, by using the generalized version of the triangle inequality (Minkowski) and applying Lemma 4 for heavy-tailed noise, we get:

$$\mathbb{E}\left[\left\|\sum_{i=1}^{k} w_i(1 - \beta_i)\theta^i\right\|_2\right] \le \left(\sum_{i=1}^{k} w_i^\kappa(1 - \beta_i)^\kappa \mathbb{E}\|\theta^i\|_2^\kappa\right)^{1/\kappa} \le 2\|\vec{\sigma}\|_1 \cdot \left(\sum_{i=1}^{k} w_i^\kappa(1 - \beta_i)^\kappa\right)^{1/\kappa}.$$

Define the term:

$$S_k := \sum_{i=1}^{k} \left( \prod_{j=i+1}^{k} \beta_j \right)^{\kappa} (1 - \beta_i)^{\kappa}.$$

Assume $\beta_j = 1 - \frac{1}{j^q}$ for some $q \in (0, 1)$.

$$
\begin{aligned}
\mathbb{E}[f(x^{k+1})] - \mathbb{E}[f(x^k)] \leq \ & -\gamma_k \mathbb{E}[\|\nabla f(x^k)\|_1] + 2\gamma_k w_0 \mathbb{E}\|\epsilon^0\|_2 \\
& + \ 2\gamma_k \sum_{i=1}^{k} w_i \beta_i \lambda_i \left( L_0 + L_1 \mathbb{E}\|\nabla f(x^i)\|_2 \right) \\
& + \ 2\gamma_k \|\vec{\sigma}\|_1 \cdot S_k^{1/\kappa} \\
& + \ \frac{L_0 \sqrt{d} \gamma_k}{2} \lambda_k + \frac{L_1 \sqrt{d} \gamma_k}{2} \lambda_k \mathbb{E}[\|\nabla f(x^k)\|_2].
\end{aligned}
$$

$$
\begin{aligned}
\mathbb{E}[f^*] - \mathbb{E}[f(x^1)] \leq \ & -\sum_{k=1}^{T} \gamma_k \mathbb{E}[\|\nabla f(x^k)\|_1] + 2\sum_{k=1}^{T} \gamma_k w_0 \mathbb{E}\|\epsilon^0\|_1 \\
& + \ 2\sum_{k=1}^{T} \gamma_k \sum_{i=1}^{k} w_i \beta_i \lambda_i L_0 + 2\sum_{k=1}^{T} \gamma_k \sum_{i=1}^{k} w_i \beta_i \lambda_i L_1 \mathbb{E}\|\nabla f(x^i)\|_2 \\
& + \ 2\|\vec{\sigma}\|_1 \sum_{k=1}^{T} \gamma_k S_k^{1/\kappa} \\
& + \ \frac{L_0 \sqrt{d}}{2} \sum_{k=1}^{T} \gamma_k \lambda_k + \frac{L_1 \sqrt{d}}{2} \sum_{k=1}^{T} \gamma_k \lambda_k \mathbb{E}[\|\nabla f(x^k)\|_2].
\end{aligned}
$$

We now aim to estimate the term with the $S_k^{1/\kappa}$ part.

$$
\begin{aligned}
\sum_{k=1}^{T} \gamma_k S_k^{1/\kappa} &= \gamma_0 \sum_{k=1}^{T} k^{-r} \left( \sum_{i=1}^{k} \left( \prod_{j=i+1}^{k} \beta_j \right)^{\kappa} (1 - \beta_i)^{\kappa} \right)^{1/\kappa} \\
&= \gamma_0 \sum_{k=1}^{T} k^{-r} \left( \sum_{i=1}^{k} \left( \prod_{j=i+1}^{k} \left( 1 - \frac{1}{j^q} \right) \right)^{\kappa} i^{-q\kappa} \right)^{1/\kappa}
\end{aligned}
$$

Then by (ii) (Lemma 6) from (Hübler et al., 2024a) to estimate the term inside the sum, we have $\sum_{t=a}^{b} t^{-p} \prod_{\tau=t+1}^{b} (1 - \tau^{-q}) \leq 2 \exp\left( \frac{1}{1-q} \right) (b+1)^{q-p}$. Thus:

$$
\begin{aligned}
\left( \sum_{i=1}^{k} \left( \prod_{j=i+1}^{k} \left( 1 - \frac{1}{j^q} \right) \right)^{\kappa} i^{-q\kappa} \right)^{1/\kappa} &\leq \left( \sum_{i=1}^{k} \prod_{j=i+1}^{k} \left( 1 - \frac{1}{j^q} \right) i^{-q\kappa} \right)^{1/\kappa} \leq 2e^{2/\kappa} (k+1)^{(q-q\kappa)1/\kappa} \\
&\leq 2e^{2/\kappa} k^{(q-q\kappa)1/\kappa}
\end{aligned}
$$

Thus:

$$\sum_{k=1}^{T} \gamma_k S_k^{1/\kappa} = \gamma_0 \sum_{k=1}^{T} k^{-r} \left( \sum_{i=1}^{k} \left( \prod_{j=i+1}^{k} \beta_j \right)^{\kappa} (1 - \beta_i)^{\kappa} \right)^{1/\kappa}$$

$$\leqslant 2\gamma_0 e^{2/\kappa} \sum_{k=1}^{T} k^{(q-q\kappa)1/\kappa - r} \leqslant 2\gamma_0 e^{2/\kappa} \frac{T^{\frac{1}{2\kappa} - \frac{1}{4}}}{\frac{1}{2\kappa} - \frac{1}{4}} = 2\gamma_0 e^{2/\kappa} \frac{T^{\frac{2-\kappa}{4\kappa}}}{\frac{2-\kappa}{4\kappa}}$$

Hense, we get:

$$\mathbb{E}[f^*] - \mathbb{E}[f(x^1)] \leq \sum_{k=1}^{T} \mathbb{E}[\|\nabla f(x^k)\|_1] \left( -\gamma_k + \frac{L_1\sqrt{d}}{2} \gamma_k \lambda_k + 2\sqrt{d} L_1 \gamma_k \sum_{i=1}^{k} w_i \beta_i \lambda_i \right)$$

$$+ \quad 2 \sum_{k=1}^{T} \gamma_k w_1 \mathbb{E}\|\epsilon^1\|_1$$

$$+ \quad 2\sqrt{d} L_0 \sum_{k=1}^{T} \gamma_k \sum_{i=1}^{k} w_i \beta_i \lambda_i$$

$$+ \quad 4\|\vec{\sigma}\|_1 \gamma_0 e^{2/\kappa} \frac{T^{\frac{2-\kappa}{4\kappa}}}{\frac{2-\kappa}{4\kappa}}$$

$$+ \quad \frac{L_0\sqrt{d}}{2} \sum_{k=1}^{T} \gamma_k \lambda_k,$$

where $\gamma_k = \gamma_0 k^{-r}$, $\lambda_k = \exp(L_1 \gamma_k \sqrt{d}) \cdot \gamma_k \sqrt{d} \leqslant C k^{-r}$, $C := \gamma_0 \exp(L_1 \gamma_0 \sqrt{d}) \sqrt{d}$ and $w_i = \prod_{j=i+1}^{k} \beta_j$.

$$\sum_{i=1}^{k} w_i \beta_i \lambda_i \gamma_k \leq C \gamma_k \sum_{i=1}^{k} i^{-r} \prod_{j=i}^{k} \beta_j.$$

By part (iii) of (Lemma 6) with $p = -r$, $a = 2$, $b = k$ (noting that the requirements hold for $p \leq 1$),

$$\sum_{i=1}^{k} i^{-r} \prod_{j=i}^{k} \beta_j \leq 2 \exp \left( \frac{1}{1-q} \right) k^{-r+q}.$$

Thus,

$$\sum_{i=1}^{k} w_i \beta_i \lambda_i \gamma_k \leq C \gamma_k \cdot 2 \exp \left( \frac{1}{1-q} \right) k^{q-r} = C \gamma_k \cdot 2e^2 k^{q-r} := 2C' \gamma_k k^{q-r},$$

Let us set stepsize $\gamma_0$ such that

$$-\gamma_k + 2\sqrt{d} C' L_1 \gamma_k k^{q-r} + \frac{L_1\sqrt{d}}{2} \gamma_k \lambda_k \leq -\frac{\gamma_k}{2}.$$

$$-1 + 2\sqrt{d} C' L_1 k^{q-r} + \frac{L_1\sqrt{d}}{2} \lambda_k \leq -\frac{1}{2}.$$

Since $\lambda_k \leq C k^{-r} \leq C' k^{-r}$ and , we obtain:

$$2\sqrt{d} C' L_1 k^{q-r} + \frac{L_1\sqrt{d}}{2} C' k^{-r} \leq \frac{1}{2}.$$

This implies:

$$C'L_1\sqrt{d}\left(2k^{q-r}+\frac{1}{2}k^{-r}\right)\le\frac{1}{2}.$$

We finally obtain:

$$C'L_1\sqrt{d}\left(2+\frac{1}{2}\right)=\frac{5}{2}C'L_1\sqrt{d}\le\frac{1}{2}\quad\Rightarrow\quad C'\le\frac{1}{5L_1\sqrt{d}}.$$

Recall:

$$\exp(L_1\gamma_0\sqrt{d})\cdot\gamma_0\sqrt{d}\le\frac{1}{5L_1e^2\sqrt{d}}.$$

Letting $y:=L_1\gamma_0\sqrt{d}$, we get $\exp(y)\cdot y\le\frac{1}{5e^2\sqrt{d}}$, and hence:

$$\gamma_0\le\frac{y^*}{L_1\sqrt{d}},\quad\text{where }y^*\text{ solves }\exp(y^*)y^*=\frac{1}{5e^2\sqrt{d}}.$$

Since the right-hand side of the inequality is always $\le 1$, and noting that $xe^x\le 2x$ on that range, taking $y^*\le\frac{1}{5\cdot2\cdot9\sqrt{d}}$ we obtain the bound for

$$\gamma_0\le\frac{1}{90L_1d}.$$

Since the gradient term is now bounded above by $-\frac{\gamma_k}{2}$, we can move it to the left-hand side:

$$\sum_{k=1}^{T}\frac{\gamma_k}{2}\mathbb{E}\|\nabla f(x^k)\|_1\le\Delta\quad+\quad 2\sum_{k=1}^{T}\gamma_k w_0\mathbb{E}\|\epsilon^0\|_1$$

$$+\quad 4\sqrt{d}L_0\sum_{k=2}^{T}C'\gamma_k k^{q-r}$$

$$+\quad 4\|\vec{\sigma}\|_1\gamma_0 e^{2/\kappa}\frac{T^{\frac{2-\kappa}{4\kappa}}}{\frac{2-\kappa}{4\kappa}}$$

$$+\quad\frac{L_0\sqrt{d}}{2}\sum_{k=1}^{T}\gamma_k\lambda_k,\tag{44}$$

$$\frac{1}{\sum_{k=1}^{T}\gamma_k}\sum_{k=1}^{T}\gamma_k\mathbb{E}\|\nabla f(x^k)\|_1\quad\le\quad\frac{2\Delta}{\sum_{k=1}^{T}\gamma_k}+\frac{4\sqrt{d}}{\sum_{k=1}^{T}\gamma_k}\sum_{k=1}^{T}\gamma_k w_0\mathbb{E}\|\epsilon^0\|_2$$

$$+\quad\frac{8\sqrt{d}L_0}{\sum_{k=1}^{T}\gamma_k}\sum_{i=2}^{T}C'\gamma_k k^{q-r}$$

$$+\quad 8\sqrt{d}\|\vec{\sigma}\|_\kappa\gamma_0 e^{2/\kappa}\frac{T^{\frac{2-\kappa}{4\kappa}}}{\frac{2-\kappa}{4\kappa}\sum_{k=1}^{T}\gamma_k}$$

$$+\quad\frac{L_0\sqrt{d}}{\sum_{k=1}^{T}\gamma_k}\sum_{k=1}^{T}\gamma_k\lambda_k,\tag{45}$$

FIRST TERM: $\frac{2\Delta}{\sum_{k=1}^{T}\gamma_k}$

Approximate with an integral for large $T$:

$$\sum_{k=1}^{T}k^{-\frac{3}{4}}\approx\int_{1}^{T}x^{-\frac{3}{4}}\,dx=\left[\frac{x^{1-\frac{3}{4}}}{1-\frac{3}{4}}\right]_{1}^{T}=4(T^{\frac{1}{4}}-1)\approx 4T^{\frac{1}{4}}.$$

$$\sum_{k=1}^{T} \gamma_0 k^{-3/4} = 4\gamma_0 T^{1/4}$$

Thus we obtain the bound:

$$\frac{2\Delta}{\sum_{k=1}^{T} \gamma_k} = \frac{\Delta}{2\gamma_0 T^{1/4}}$$

SECOND TERM: $\frac{4}{\sum_{k=1}^{T} \gamma_k} \sum_{k=1}^{T} \gamma_k w_0 \mathbb{E}\|\epsilon^0\|_1$

Recalling the definition: $w_0 = \prod_{j=1}^{k} \left(1 - j^{-1/2}\right)$ Compute the sum using Lemma ? case ii:

$$\frac{4\mathbb{E}\|\epsilon^1\|_1}{\sum_{k=1}^{T} \gamma_k} \sum_{k=1}^{T} \gamma_0 k^{-3/4} \prod_{j=1}^{k} \left(1 - j^{-1/2}\right) = \frac{4\mathbb{E}\|\epsilon^1\|_1}{\gamma_0 \sum_{k=1}^{T} k^{-3/4}} O(1)$$

Thus, the term becomes:

$$\frac{4\mathbb{E}\|\epsilon^0\|_1}{\sum_{k=1}^{T} \gamma_k} \sum_{k=1}^{T} \gamma_0 k^{-3/4} \prod_{j=1}^{k} \left(1 - j^{-1/2}\right) = \frac{\mathbb{E}\|\epsilon^0\|_1}{T^{\frac{1}{4}}}.$$

THIRD TERM: $\frac{4\sqrt{d}C'L_0}{\sum_{k=1}^{T} \gamma_k} \sum_{k=1}^{T} \gamma_k k^{q-r}$

With $q - r = -\frac{1}{4}$:

$$\sum_{k=1}^{T} \gamma_k k^{q-r} = \sum_{k=1}^{T} \gamma_0 k^{-\frac{3}{4}} \cdot k^{-\frac{1}{4}} = \gamma_0 \sum_{k=1}^{T} k^{-1} \approx \gamma_0 \log T,$$

The term is:

$$\frac{4\sqrt{d}C'L_0}{\sum_{k=1}^{T} \gamma_k} \cdot \gamma_0 \log T = \frac{\sqrt{d}C'L_0 \log T}{T^{1/4}}.$$

FOURTH TERM: $8\|\vec{\sigma}\|_1 \gamma_0 e^{2/\kappa} \frac{T^{\frac{2-\kappa}{4\kappa}}}{\frac{2-\kappa}{4\kappa} \sum_{k=1}^{T} \gamma_k}$

The term is:

$$8\|\vec{\sigma}\|_1 \gamma_0 e^{2/\kappa} \frac{T^{\frac{2-\kappa}{4\kappa}}}{\frac{2-\kappa}{4\kappa} \sum_{k=1}^{T} \gamma_k} = 2\|\vec{\sigma}\|_1 e^{/\kappa} \frac{T^{\frac{2-\kappa}{4\kappa} - \frac{1}{4}}}{\frac{2-\kappa}{4\kappa}} = 2\|\vec{\sigma}\|_1 e^{2/\kappa} \frac{1}{\frac{2-\kappa}{4\kappa} T^{\frac{\kappa-1}{2\kappa}}}$$

FIFTH TERM: $\frac{L_0\sqrt{d}}{\sum_{k=1}^{T} \gamma_k} \sum_{k=1}^{T} \gamma_k \lambda_k$

Since $\lambda_k \le Ck^{-\frac{3}{4}}$:

$$\sum_{k=1}^{T} \gamma_k \lambda_k \le \sum_{k=1}^{T} \gamma_0 k^{-\frac{3}{4}} \cdot Ck^{-\frac{3}{4}} = \gamma_0 C \sum_{k=1}^{T} k^{-\frac{3}{2}} \approx \gamma_0 C \int_{1}^{T} x^{-\frac{3}{2}} dx = \gamma_0 C \cdot \left[\frac{x^{-\frac{1}{2}}}{-\frac{1}{2}}\right]_{1}^{T} \quad (46)$$

$$= \gamma_0 C \cdot 2(1 - T^{-\frac{1}{2}}) \approx 2\gamma_0 C. \quad (47)$$

The term is:

$$\frac{L_0\sqrt{d}}{\sum_{k=1}^{T} \gamma_k} \cdot 2\gamma_0 C = \frac{L_0\sqrt{d}C}{2T^{\frac{1}{4}}}.$$

The bound is:

$$\min_{k \in \overline{1,T}} \mathbb{E}\|\nabla f(x^k)\|_1 \leqslant \frac{1}{\sum_{k=1}^T \gamma_k} \sum_{k=1}^T \gamma_k \mathbb{E}\|\nabla f(x^k)\|_1 \leqslant \frac{\Delta}{2\gamma_0 T^{1/4}} + \frac{\mathbb{E}\|\epsilon^0\|_1}{T^{\frac{1}{4}}} + \frac{\sqrt{d}C'L_0 \log T}{T^{1/4}}$$

$$+ 2\|\vec{\sigma}\|_1 e^{2/\kappa} \frac{1}{\frac{2-\kappa}{4\kappa}T^{\frac{\kappa-1}{2\kappa}}} + \frac{L_0\sqrt{d}C}{2T^{\frac{1}{4}}}.$$

ARBITRARY TUNING

The second term is much smaller than the third one, hence we omit it. We now choose $T$ to limit:

$$\frac{(\Delta/\gamma_0 + L_0 C\sqrt{d})}{2T^{1/4}} + \frac{\sqrt{d}C'L_0 \log T}{T^{1/4}} \leq \frac{(\Delta/\gamma_0 + 3L_0 C'\sqrt{d}) \log T}{2T^{1/4}} \leq \frac{\varepsilon}{2},$$

and

$$2\|\vec{\sigma}\|_1 e^{2/\kappa} \frac{1}{\frac{2-\kappa}{4\kappa}T^{\frac{\kappa-1}{2\kappa}}} \leq \frac{\varepsilon}{2}.$$

Hense:

$$T = \tilde{O}\left(\left(\frac{(\Delta/\gamma_0 + L_0\gamma_0 d)}{\varepsilon}\right)^4 + \left(\frac{\|\vec{\sigma}\|_1}{\varepsilon}\right)^{\frac{2\kappa}{\kappa-1}}\right).$$

$\square$

## C   RESTARTED minibatch-SignSGD AND MajorityVote-SignSGD

For PL functions (As. 4), we can apply restart technique to minibatch-SignSGD and MajorityVote-SignSGD. At each round, we run a base algorithm with certain parameters and then aggregate the output point. This output point is used as an initial point for the next round.

---

**Algorithm 5** Restarted-$\mathcal{A}$

---

**Input:** Starting point $x^0 \in \mathbb{R}^d$, number of restarts $\tau$, base algorithm $\mathcal{A}$, parameters $\{\theta_n\}_{n=1}^\tau$.
1: **for** $n = 1, \ldots, \tau$ **do**
2:    Run Algorithm $\mathcal{A}$ with parameters $\theta_n$ and initial point $x^{n-1}$;
3:    Set $x^n$ as the aggregated output point from the previous round: the point with the minimal $\ell_2$ gradient norm;
4: **end for**
**Output:** $x^\tau$

---

**Theorem 10** (**HP complexity for Restarted** minibatch-SignSGD)**.** *Consider lower-bounded $(L_0, L_1)$-smooth, $\mu$-PL function $f$ (As. 1, 2, 4) and HT gradient estimates (As. 3). Then restarted* minibatch-SignSGD *requires the sample complexity $N$ to achieve $f(x^\tau) - f(x^*) \leq \varepsilon$ with probability at least $1 - \delta$ for:*

***Optimal tuning for*** $\varepsilon \geq (\frac{8L_0}{L_1})^2$***:*** $\tau = \log(\Delta/\varepsilon)$, *iterations* $T_n = O\left(\frac{L_1^\delta d\sqrt{\Delta}}{2^{n/2}\sqrt{\mu}}\right)$, *constant batchsizes* $B_n \equiv \max\left\{1, \left(\frac{1024\|\vec{\sigma}\|_1^2}{\mu\varepsilon}\right)^{\frac{\kappa}{2(\kappa-1)}}\right\}$, *constant stepsizes* $\gamma_n \equiv \frac{1}{48L_1^\delta d\sqrt{d}}$ :

$$N = O\left(\frac{L_1^\delta d\sqrt{\Delta}}{\sqrt{\mu}}\left[1 + \left(\frac{\|\vec{\sigma}\|_1^2}{\mu\varepsilon}\right)^{\frac{\kappa}{2(\kappa-1)}}\right]\right)$$

***Optimal tuning for*** $\varepsilon < (\frac{8L_0}{L_1})^2$***:*** $\tau = \log(\Delta/\varepsilon)$, *iterations* $T_n = O\left(\frac{L_0^\delta d}{\mu}\right)$, *constant batchsizes* $B_n \equiv \max\left\{1, \left(\frac{1024\|\vec{\sigma}\|_1^2}{\mu\varepsilon}\right)^{\frac{\kappa}{2(\kappa-1)}}\right\}$, *constant stepsizes* $\gamma_n \equiv \sqrt{\frac{\Delta}{2^{n+4}T_n L_0^\delta d}}$ :

$$N = O\left(\frac{L_0^\delta d \log \frac{\Delta}{\varepsilon}}{\mu}\left[1 + \left(\frac{\|\vec{\sigma}\|_1^2}{\mu\varepsilon}\right)^{\frac{\kappa}{2(\kappa-1)}}\right]\right),$$

*where* $\Delta = f(x^1) - f^*, L_0^\delta = L_0 \log(\frac{\log \frac{\Delta}{\varepsilon}}{\delta}), L_1^\delta = L_1 \log(\frac{\log \frac{\Delta}{\varepsilon}}{\delta})$.

**Theorem 11** (**HP complexity for Restarted** MajorityVote-SignSGD)**.** *Consider lower-bounded* $(L_0, L_1)$-*smooth,* $\mu$-*PL function* $f$ *(As.* 1, 2, 4*) and HT gradient estimates corrupted by **unimodal and symmetric HT noise with** $\kappa > 0$ *(As.* 3*). Then restarted* MajorityVote-SignSGD *requires the sample complexity* $N$ *to achieve* $f(x^\tau) - f(x^*) \leq \varepsilon$ *with probability at least* $1 - \delta$ *for:*

***Optimal tuning for*** $\varepsilon \geq (\frac{8L_0}{L_1})^2$**:** $\tau = \log(\Delta/\varepsilon)$, *iterations* $T_n = O\left(\frac{L_1^\delta d\sqrt{\Delta}}{2^{n/2}\sqrt{\mu}}\right)$, *constant batchsizes* $M_n \equiv \max\left\{\frac{160}{\kappa^2}, \frac{1024\|\vec{\sigma}\|_1^2}{\mu\varepsilon}\right\}$, *constant stepsizes* $\gamma_n \equiv \frac{1}{48L_1^\delta d}$ :

$$N = O\left(\frac{L_1^\delta d\sqrt{\Delta}}{\sqrt{\mu}}\left[\frac{1}{\kappa^2} + \frac{\|\vec{\sigma}\|_1^2}{\mu\varepsilon}\right]\right)$$

***Optimal tuning for*** $\varepsilon < (\frac{8L_0}{L_1})^2$**:** $\tau = \log(\Delta/\varepsilon)$, *iterations* $T_n = O\left(\frac{L_0^\delta d}{\mu}\right)$, *constant batchsizes* $M_n \equiv \max\left\{\frac{160}{\kappa^2}, \frac{1024\|\vec{\sigma}\|_1^2}{\mu\varepsilon}\right\}$, *constant stepsizes* $\gamma_n \equiv \sqrt{\frac{\Delta}{2^{n+4}T_n L_0^\delta d}}$ :

$$N = O\left(\frac{L_0^\delta d\log\frac{\Delta}{\varepsilon}}{\mu}\left[\frac{1}{\kappa^2} + \frac{\|\vec{\sigma}\|_1^2}{\mu\varepsilon}\right]\right),$$

*where* $\Delta = f(x^1) - f^*, L_0^\delta = L_0 \log(\frac{\log \frac{\Delta}{\varepsilon}}{\delta}), L_1^\delta = L_1 \log(\frac{\log \frac{\Delta}{\varepsilon}}{\delta})$.

*Proof.* Here we prove only Theorem 10. The proof of Theorem 11 is similar.

Consider one round of restarted algorithm with the initial condition $\Delta$ which will be transformed into $\Delta_2 \leq \Delta/2$. In total, we will have $\log(\Delta/\varepsilon)$ rounds. Instead of the initial failure probability $\delta$ we use decreased probability $\delta/\log\frac{\Delta}{\varepsilon}$, since the probability of holding bounds $\log(\Delta/\varepsilon)$ times for all restarts equals to $(1 - \frac{\delta}{\log(\Delta/\varepsilon)})^{\log(\Delta/\varepsilon)} \geq (1 - \frac{\delta}{\log(\Delta/\varepsilon)}\log(\Delta/\varepsilon)) = (1 - \delta)$.

Plugging in constant stepsizes $\gamma_k \equiv \gamma \leq \frac{1}{48L_1^\delta d}$ in (20) implies $C_T = T\gamma^2, \gamma^{max} = \gamma$:

$$\frac{1}{T}\sum_{k=1}^{T}\|\nabla f(x^k)\|_1 \leq \frac{16\Delta}{T\gamma} + 256L_0^\delta d\gamma + 32\|\vec{\sigma}_k\|_1.$$

Due to Batching Lemma 4, we can estimate the $\kappa-$th moment of the batched estimate for constant batchsizes $B_k \equiv B$ as $\|\vec{\sigma}_k\|_1 \leq \frac{2\|\vec{\sigma}\|_1}{B^{\frac{\kappa-1}{\kappa}}}$ and derive:

$$\min_{k\in\overline{1,T}}\|\nabla f(x^k)\|_2 \leq \frac{1}{T}\sum_{k=1}^{T}\|\nabla f(x^k)\|_2 \leq \frac{1}{T}\sum_{k=1}^{T}\|\nabla f(x^k)\|_1 \leq \frac{16\Delta}{T\gamma} + 256L_0^\delta d\gamma + 32\frac{\|\vec{\sigma}_k\|_1}{B^{\frac{\kappa-1}{\kappa}}}.$$

Next, we square the inequality and apply PL condition:

$$\|\nabla f(x_{min}^T)\|_2^2 \leq 8\left(\frac{16\Delta}{T\gamma}\right)^2 + 8(256L_0^\delta d\gamma)^2 + 8\left(32\frac{\|\vec{\sigma}_k\|_1}{B^{\frac{\kappa-1}{\kappa}}}\right)^2,$$

$$f(x_{min}^T) - f(x^*) \leq \|\nabla f(x_{min}^T)\|_2^2 \leq 8\left(\frac{16\Delta}{T\gamma}\right)^2 + 8(256L_0^\delta d\gamma)^2 + 8\left(32\frac{\|\vec{\sigma}_k\|_1}{B^{\frac{\kappa-1}{\kappa}}}\right)^2,$$

$$\Delta_2 = f(x_{min}^T) - f(x^*) \leq \frac{4}{\mu}\left[\left(\frac{16\Delta}{T\gamma}\right)^2 + (256L_0^\delta d\gamma)^2 + \left(32\frac{\|\vec{\sigma}_k\|_1}{B^{\frac{\kappa-1}{\kappa}}}\right)^2\right],$$

where $x_{min}^T = \arg\min_{k\in\overline{1,T}} f(x^k)$.

**Case** $\varepsilon \geq (\frac{8L_0}{L_1})^2$**, optimal tuning:** We use stepsizes $\gamma = \frac{1}{48L_1^\delta d} \Rightarrow (256L_0^\delta d\gamma)^2 \leq \varepsilon/2$ and batchsizes $32\frac{\|\vec{\sigma}\|_1}{B^{\frac{\kappa-1}{\kappa}}} \leq \sqrt{\mu\varepsilon/8} \Rightarrow B_k \equiv \max\left\{1, \left(\frac{1024\|\vec{\sigma}\|_1^2}{\mu\varepsilon}\right)^{\frac{\kappa}{2(\kappa-1)}}\right\}$. The number of iterations $T$

is chosen to decrease the term $\Delta$ by half:

$$\Delta_2 \leq \Delta^2 \left( \frac{1028 * 48^2 (L_1^\delta)^2 d^2}{\mu T^2} \right) \leq \frac{\Delta}{2} \Rightarrow T = O\left( \frac{L_1^\delta d \sqrt{\Delta}}{\sqrt{\mu}} \right).$$

At each restart, the initial condition $\Delta_n$ becomes $\Delta_{n+1} \leq \frac{\Delta_n}{2} \leq \frac{\Delta}{2^{n-1}}$, and the total number of iterations is

$$T_{total} = \sum_{n=1}^{\log(\Delta/\varepsilon)} O\left( \frac{L_1^\delta d \sqrt{\Delta}}{\sqrt{\mu} 2^{\frac{n-1}{2}}} \right) = O\left( \frac{L_1^\delta d \sqrt{\Delta}}{\sqrt{\mu}} \right), \tag{48}$$

with the total number of oracle calls:

$$N = T_{total} * B_k = O\left( \frac{L_1^\delta d \sqrt{\Delta}}{\sqrt{\mu}} \left[ 1 + \left( \frac{\|\vec{\sigma}\|_1^2}{\mu\varepsilon} \right)^{\frac{\kappa}{2(\kappa-1)}} \right] \right). \tag{49}$$

**Case $\varepsilon \leq \left(\frac{8L_0}{L_1}\right)^2$, optimal tuning:** We use batchsizes $32\frac{\|\vec{\sigma}\|_1}{B^{\frac{\kappa-1}{\kappa}}} \leq \sqrt{\mu\varepsilon/8} \Rightarrow B_k \equiv \max\left\{ 1, \left( \frac{1024\|\vec{\sigma}\|_1^2}{\mu\varepsilon} \right)^{\frac{\kappa}{2(\kappa-1)}} \right\}$ and stepsizes $\gamma = \sqrt{\frac{\Delta}{16TL_0^\delta d}}$ to have

$$\Delta_2 \leq \frac{8}{\mu} \left[ \frac{1024 L_0^\delta d}{T} \Delta + \varepsilon \right].$$

The number of iterations $T$ is chosen to decrease the term $\Delta$ by half:

$$\Delta_2 \leq \frac{8}{\mu} \frac{1024 L_0^\delta d}{T} \Delta \leq \frac{\Delta}{2} \Rightarrow T = O\left( \frac{L_0^\delta d}{\mu} \right).$$

At each restart, the initial condition $\Delta_n$ becomes $\Delta_{n+1} \leq \frac{\Delta_n}{2} \leq \frac{\Delta}{2^{n-1}}$, and the total number of iterations is

$$T_{total} = \sum_{n=1}^{\log(\Delta/\varepsilon)} O\left( \frac{L_0^\delta d}{\mu} \right) = \left( \frac{L_0^\delta d \log \frac{\Delta}{\varepsilon}}{\mu} \right),$$

with the total number of oracle calls:

$$N = T_{total} * B_k = O\left( \frac{L_0^\delta d \log \frac{\Delta}{\varepsilon}}{\mu} \left[ 1 + \left( \frac{\|\vec{\sigma}\|_1^2}{\mu\varepsilon} \right)^{\frac{\kappa}{2(\kappa-1)}} \right] \right). \tag{50}$$

$\square$

# D    EXPERIMENTAL VALIDATION OF THE THEORETICAL CONVERGENCE BOUNDS

In this section, we run experiments to verify the following convergence bound from Lemma 1 for the backbone SignSGD method:

$$\sum_{k=1}^{T} \frac{\gamma_k}{16} \|\nabla f(x^k)\|_1 \leq \Delta + L_0 d \sum_{k=1}^{T} \gamma_k^2 + 2 \sum_{k=1}^{T} \gamma_k \|\vec{\sigma}_k\|_1 + 6d(\gamma_1 \|\nabla f(x^1)\|_1 + 2C_T L_0) \log \frac{1}{\delta},$$

where $C_T := \max_{k \in \overline{1,T}} \gamma_k \cdot \sum_{\tau=1}^{k-1} \gamma_\tau, \gamma_k \leq 1/(48L_1 d^{\frac{3}{2}} \log \frac{1}{\delta})$ and $\Delta = f(x^1) - f^*$. In case of constant stepsizes $\gamma_k \equiv \gamma$, the bound transforms into

$$\frac{1}{T} \sum_{k=1}^{T} \|\nabla f(x^k)\|_1 \leq \frac{4\Delta}{T\gamma} + 80L_0 d\gamma \log(1/\delta) + 8\|\vec{\sigma}\|_1 + 24\frac{d\|\nabla f(x^1)\|_1}{T} \log(1/\delta), \quad \gamma \leq \frac{1}{(48L_1 d^{\frac{3}{2}} \log \frac{1}{\delta})}. \tag{51}$$

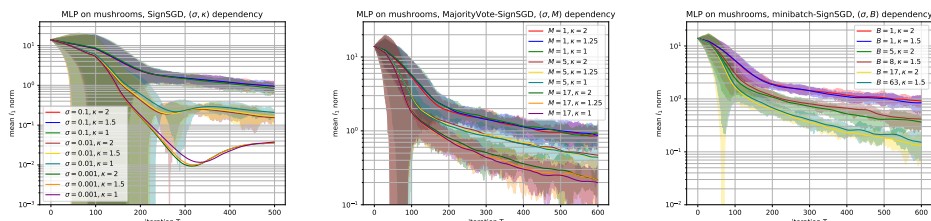

Figure 1: Experimental noise dependencies for $(L_0, L_1)$-smoooth problems.

**Objective function and noise.** We optimize a non-convex neural network for classification task with features $X$ and one-dimensional labels $y$. The network $NN_\theta$ with parameter vector $\theta$ consists of two fully connected layers, ReLU activation, batch normalization and dropout. The objective function is the following logistic regression with $L_0$ and $L_1$ regularizations with coefficients $\lambda_{L_0}$ and $\lambda_{L_1}$, respectively:

$$f(\theta) = \log(1 + \exp(-\langle y, NN_\theta(X) \rangle)) + \frac{\lambda_{L_0}}{2} \cdot \|\theta\|_2^2 + \exp(\lambda_{L_1} \cdot \langle \vec{1}, \theta \rangle).$$

The regularization coefficients $\lambda_{L_0}, \lambda_{L_1}$ are the smoothness constants of the corresponding regularization terms (see Appendix B.1). If their value are changed by some amount then the actual $L_0, L_1$ smoothness constants of the objective function $f$ are changed by the exactly same amount.

To model the noise, we compute the whole gradient $\nabla f(\theta)$ and artificially add noise vector with independent components sampled from $\alpha$-stable Levy distribution with scale $\sigma$ ($\alpha$ is the $\kappa$ parameter).

As training data, we consider the dataset mushrooms from LibSVM (Chang & Lin, 2011). The matrix $X$ has shape $(6499, 112)$, hence, we set the NN layers sizes $(112, 32, 1)$ and dropout rate $0.1$.

**Noise dependencies.** First, we verify the linear dependence of the achieved accuracy (51) on noise $\sigma$. We set small regularization coefficients $\lambda_{L_0} = 0.01, \lambda_{L_1} = 0.001$ and constant stepsize $\gamma = 3 \cdot 10^{-4}$ for all experiments in this paragraph.

Next, we vary $\sigma \in [0.1, 0.01, 0.001]$ and $\kappa \in [2, 1.5, 1]$. The results over 10 runs with standard deviation bars are depicted in the left graph of Figure 1.

In practice, the achieved accuracy does not depend on $\kappa$, only linearly on $\sigma$. We also wish to highlight the small size of error bars especially in the end of training which corresponds to mild $\log \frac{1}{\delta}$ dependence.

In addition, we check how well batching (Alg. 2) and majority voting (Alg. 3) reduce the noise, .i.e, whether reduction laws $\sigma/B^{\frac{\kappa-1}{\kappa}}$ and $\sigma/\sqrt{M}$ from Theorems 1 and 2 hold true. To reduce the noise by 2 and 4 times for majority voting, we use batchsizes $M = 1, 4, 16$ for all $\kappa \in [2, 1.25, 1]$. The results are shown in the middle graph of Figure 1. To reduce the noise by 2 and 4 times for batching, we use batchsizes $B = 1, 4, 16$ for $\kappa = 2$ and $B = 1, 8, 64$ for $\kappa = 1.5$. The results are shown in the right graph of Figure 1. In practice, both methods actually reduce the noise according to the theoretical laws.

**Two phase convergence.** Here, we demonstrate the convergence speed slowdown after reaching the accuracy $\frac{8L_0}{L_1\sqrt{d}}$ as it stated in Theorem 1. We also test the parameter-free tuning strategy proposed for reaching this behavior.

We slightly change the setup to better control constants $L_0, L_1$. We replace fully-connected neural network $NN_\theta(X)$ with simple linear transform $NN_\theta(X) = X \cdot \theta$, hence, the current objective function with only $L_0$ regularization is:

$$f(\theta) = \log(1 + \exp(-\langle y, X \cdot \theta \rangle)) + \frac{\lambda_{L_0}}{2} \cdot \|\theta\|_2^2.$$

In this case, we can directly compute (see Example 3 for *mushrooms* dataset) and control constants $L_0 = \lambda_{L_0}, L_1 \approx 5.58$. The noise parameters are $\kappa = 1.5$ and $\sigma = 0.1$.

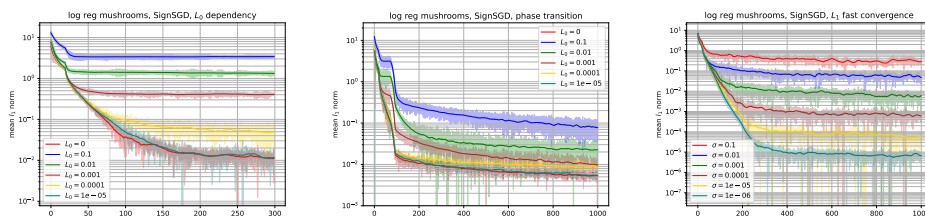

Figure 2: Experimental convergence speed transition for $(L_0, L_1)$-smooth problems.

First, we set constant stepsize $\gamma = 10^{-1}$ and vary $\lambda_{L_0} \in [0, 10^{-1}, 10^{-2}, 10^{-3}, 10^{-4}, 10^{-5}]$. The results over 10 runs are depicted in the left graph of Figure 2. One can see that the final accuracy drops linearly with $\lambda_{L_0}$ until it reaches the noise level. Before the plateau, we observe the fast $L_1$ convergence.

In the next experiment, we follow the parameter-free tuning strategy and start to decrease stepsizes as $1/\sqrt{k}$ after the plateau. The results are presented in the middle graph of Figure 2. One can see that now method can slowly reach the same noise level after the first plateau. The speed transition accuracy also drops linearly with $\lambda_{L_0}$.

Finally, we show that, for functions with $L_0 = 0$, our method with constant stepsize convergences to noise level $\sigma$ despite the value of the constant $L_1$. We set $\lambda_{L_0} = 0$ and vary the noise level $\sigma \in [10^{-1}, 10^{-2}, 10^{-3}, 10^{-4}, 10^{-5}, 10^{-6}]$. The results are shown in the right graph of Figure 2. These results clearly support the theory with alone linear dependence on $\sigma$.

# E    ADDITIONAL EXPERIMENTS

## E.1    MIXTURE OF EXPERTS PRE-TRAINING EXPERIMENTS

We complement our experiments with another setup – different architecture and data. In Section 3, we used a dense LLaMA model; now, we have switched to a Mixture of Experts (MoE) architecture based on the same LLaMA model, retaining RoPE and identical activation functions. Our MoE model follows the Switch Transformer (Fedus et al., 2022) MoE variant with classical top $k = 2$ gating and 8 experts, giving us approximately 520M parameters if we have the same configuration as 130M LLaMA. We conduct these experiments on the FineWeb dataset (Penedo et al.) a popular corpus for LLM pre-training.

We run AdamW, M-SignSGD, M-NSGD and M-ClippedSignSGD optimizers following the best practices from our earlier setup on dense models. We train with a batch size of 256 and sequence length 512 for 42k (5.5B tokens) and 336k steps (44B tokens). That is for the second training horizon we go far beyond the Chinchilla optimal tokens-per-parameters ration. The results are presented in Tables 3 and 4 respectively.

Table 3: Perplexity of LLaMa-base MoE 520M model pre-trained on FineWeb for 42k steps. Lower is better.

| Optimizer | Perplexity ↓ |
|---|---|
| AdamW | *22.85* |
| M-SignSGD | **23.19** |
| M-NSGD | 23.32 |
| M-ClippedSignSGD | 23.30 |

We would like to highlight that M-SignSGD scales remarkably well with increasing model size, outperforming M-NSGD and M-ClippedSignSGD. Additionally, we encountered difficulties running M-ClippedSGD in this setting. Consequently, we decided to include a clipped version of M-

Table 4: Perplexity of LLaMa-base MoE 520M model pre-trained on FineWeb for 336k steps. Lower is better.

| Optimizer | Perplexity ↓ |
|-----------|--------------|
| AdamW | *18.68* |
| M-SignSGD | 18.87 |

SignSGD, which aligns with our approach since we consider only an EMA of momentum in the update.

### E.2 ROBUSTNESS WITH RESPECT TO RANDOM SEED

To verify the robustness of our approach, we repeated the experiment from Table 2 with three different random seeds. As shown in Table 5, the performance remains highly consistent across all seeds, with a standard deviation $\leq 0.03$ for all the methods.

Table 5: Comparison of mean and standard deviation of the validation perplexity for various optimization methods for LLaMA 130M model trained on C4.

| Method | Perplexity ↓ |
|--------|--------------|
| Model size | 130M |
| M-SignSGD | $\mathbf{18.37}_{\pm.01}$ |
| M-NSGD | $19.28_{\pm.03}$ |
| M-ClippedSGD | $18.95_{\pm.03}$ |
| AdamW | $18.67_{\pm.00}$ |

## F EXPERIMENTAL DETAILS

### F.1 HYPERPARAMETERS SWEEP

We adopted a LLaMA-based architecture (Touvron et al., 2023) with RMSNorm (Zhang & Sennrich, 2019) and SwiGLU (Shazeer, 2020) activations on the C4 dataset (Raffel et al., 2020). Following (Lialin et al., 2023), we used a batch size of 512 sequences and a sequence length of 256. We used a T5 tokenizer, since it was also trained on C4 with dictionary size equal to 32k. We trained the model for 100k steps.

For all experiments, while the main model parameters use the respective optimization method, the LM head layer is optimized with AdamW (Loshchilov, 2017). This follows prior work (Zhao et al., 2024) which demonstrated that the LM head layer requires more fine-grained effective learning rate adaptation across different tokens for optimal performance. We used the Nesterov acceleration scheme with a momentum value of 0.9 for all methods except AdamW. For AdamW, we used standard hyperparameters: $\beta_1 = 0.9, \beta_2 = 0.999, \varepsilon =$1e-8.

We selected the learning rate through a grid search with multiplicative step of $10^{\frac{1}{4}}$ (LM head layer optimized with AdamW and learning rate equal to 1e-3). We used a cosine learning rate schedule with a warmup of 10% of the total number of steps and decay of the final learning rate down to 10% of the peak learning rate. In addition, we selected the best weight decay value between [0, 0.01, 0.1].

The final best hyperparameters are shown in Table 6.

### F.2 COMPUTATIONAL RESOURCES

We conducted all experiments described in Section 3 and appendix E using NVIDIA A100 GPUs. We utilized 8 GPUs (full node) with `torch.nn.parallel.DistributedDataParallel`

Table 6: LLaMA 130m pre-raining hyperparameters.

| Method | M-ClippedSGD | M-NSGD | M-SignSGD | AdamW |
|--------|--------------|--------|-----------|-------|
| **Learning rate** | $10^{1.5}$ | $10^0$ | $10^{-2.75}$ | $10^{-3}$ |
| **Gradient clipping** | 0.03125 | - | - | 1.0 |
| **Weight decay** | 0 | 0 | 0.01 | 0.01 |

for most of the runs. A complete run for the 130M model (100k steps) took 6 hours, whereas each run for 1.3B model (300k steps) lasted for approximately 2 days.

## G minibatch-SignSGD FOR DISTRIBUTED OPTIMIZATION

Consider distributed optimization with one server and $M$ workers, each of which calculates its own gradient estimate. The server receives all estimates, aggregates them, and sends back the updated solution to the workers. Sign-based methods are so effective in terms of communication (Bernstein et al., 2018b; Jin et al., 2020), as sending a sign vector costs only $O(d)$ operations. We use aggregation based on the majority voting.

---

**Algorithm 6** Distributed-MajorityVote-SignSGD

**Input:** Starting point $x^1 \in \mathbb{R}^d$, number of iterations $T$, stepsizes $\{\gamma_k\}_{k=1}^T$, batchsizes $\{B_k\}_{k=1}^T$.
1: **for** $k = 1, \ldots, T$ **do**
2:     Sample $\{\xi_i^{k,j}\}_{i=1}^{B_k}$ and compute gradient estimate $g^{k,j} = \sum_{i=1}^{B_k} \nabla f(x^k, \xi_i^{k,j})/B_k$ for each worker $j \in \overline{1, M}$;
3:     Send signs $\text{sign}(g^{k,j})$ to server for each worker $j \in \overline{1, M}$;
4:     Compute on server $g^k = \text{sign}\left(\sum_{j=1}^M \text{sign}(g^{k,j})\right)$;
5:     Send point $x^{k+1} = x^k - \gamma_k \cdot g^k$ to each worker;
6: **end for**
**Output:** uniformly random point from $\{x^1, \ldots, x^T\}$.

---

**Theorem 12** (**HP complexity for** Distributed-MajorityVote-SignSGD)**.** *Consider lower-bounded $(L_0, L_1)$-smooth function $f$ (As. 1, 2) and HT gradient estimates $\kappa \in (1, 2]$ (As. 3). Then Alg. 6 with $M$ workers requires the sample complexity $N_M$ per worker to achieve $\frac{1}{T}\sum_{k=1}^T \|\nabla f(x^k)\|_1 \leq \varepsilon$ with probability at least $1 - \delta$ for:*

***Optimal tuning:*** $T = O\left(\frac{\Delta L_1^\delta d}{\varepsilon}\right), \gamma_k \equiv \frac{1}{48 L_1^\delta d}, B_k \equiv \left(\frac{16\|\vec{\sigma}\|_1}{\sqrt{M}\varepsilon}\right)^{\frac{\kappa}{\kappa-1}}$ *for* $\varepsilon \geq \frac{8L_0}{L_1}$ *and* $T = O\left(\frac{L_0^\delta d}{\varepsilon^2}\right), \gamma_k \equiv \sqrt{\frac{\Delta}{20 L_0^\delta dT}}, B_k \equiv \left(\frac{16\|\vec{\sigma}\|_1}{\sqrt{M}\varepsilon}\right)^{\frac{\kappa}{\kappa-1}}$ *for* $\varepsilon \leq \frac{8L_0}{L_1}$:

$$N_M = O\left(\left(\frac{\Delta L_0 d}{\varepsilon^2} + \frac{\Delta L_1 d}{\varepsilon}\right)\left[1 + \left(\frac{\|\vec{\sigma}\|_1}{\sqrt{M}\varepsilon}\right)^{\frac{\kappa}{\kappa-1}}\right]\log 1/\delta\right), \tag{52}$$

***Parameter-free tuning:***[1] *Until plateau* $\gamma_k = \gamma_0 \leq \frac{1}{48 L_1^\delta d}, B_k = B_0 k^2$, *after* $\gamma_k = \frac{\gamma_0}{\sqrt{k}}, B_k = B_0 k$:

$$\varepsilon \geq \frac{8L_0}{L_1\sqrt{d}} \quad \Rightarrow \quad N_M = \tilde{O}\left(B_0\left(\frac{\Delta}{\gamma_0 \varepsilon}\right)^3 + \frac{1}{B_0^2}\left(\frac{\|\vec{\sigma}\|_1}{\sqrt{M}\varepsilon}\right)^{\frac{3\kappa}{2(\kappa-1)}}\right),$$

$$\varepsilon \ll \frac{8L_0}{L_1\sqrt{d}} \quad \Rightarrow \quad N_M = \tilde{O}\left(\frac{B_0(L_0^\delta \gamma_0 d + \Delta/\gamma_0)^4}{\varepsilon^4} + \frac{1}{B_0}\left(\frac{\|\vec{\sigma}\|_1}{\sqrt{M}\varepsilon}\right)^{\frac{2\kappa}{\kappa-1}}\right),$$

*where* $\Delta = f(x^1) - f^*, L_0^\delta = L_0 \log(1/\delta), L_1^\delta = L_1 \log(1/\delta)$.

---
[1]These bounds are proved for a metric $\min_{k \in \overline{1, T}} \|\nabla f(x_k)\|_1 \leq \varepsilon$.

*Proof of Theorem 12.* This proof completely copies the proof of minibatch-SignSGD Complexity Theorem 1 from Appendix B.4 with substitution of $\|\vec{\sigma}\|_1$ with $\frac{\|\vec{\sigma}\|_1}{\sqrt{M}}$. Such substitution is justified by MajorityVote-SignSGD Convergence Lemma 7 which tells that noise level drops as $\sqrt{M}$ with the growth of worker number $M$. The condition $M \geq 160/\kappa^2$ is satisfied for $\kappa > 1$ automatically after the fixed number of training steps in the beginning. $\qquad\square$

## H  THE USE OF LARGE LANGUAGE MODELS (LLMS)

Large Language Models were used only to check and correct grammar, as well as to rephrase short parts of the text for improved clarity.

