# OpenReview forum: "Power of Sign: High Probability Bounds Under $(L_0, L_1)$-smoothness and Heavy-Tailed Noise"
_ICLR.cc/2026/Conference — Submitted to ICLR 2026_

### Official Review · Reviewer_5J95 · 2025-10-16

**Soundness:** 1
**Presentation:** 2
**Contribution:** 1
**Rating:** 2
**Confidence:** 4

**Summary:**

This work studies sign-based algorithms for non-convex optimization under generalized smoothness and heavy-tailed noise. The authors studied three methods: minibatch-SignSGD, MajorityVote-SignSGD, and M-SignSGD (the first two, with a restarted scheme, can deal with functions satisfying the PL condition). For the first two algorithms, the authors provide high-probability convergence results. For the last one, the authors prove convergence in expectation. Lastly, numerical experiments are conducted to demonstrate the effectiveness of theories.

**Strengths:**

1. The writing is reader-friendly.
1. The motivation is well-explained.

**Weaknesses:**

1. Line 047, the work of (Nemirovski et al., 2009) didn't consider non-convex objectives. Moreover, for this sentence, I don't think it is necessary to distinguish between sub-Gaussian and bounded-variance noise, since the authors are discussing convergence in expectation.

1. Line 056, the first relaxation of $(L_0, L_1)$-smoothness to once differentiable functions is due to (Zhang et al., 2020a) cited in the current paper, but not (Chen et al., 2023).

1. Line 065, I cannot see how the work (Davis et al., 2021) reflects the expensive training of large deep learning models. This work is of course important in the literature of high-probability convergence, but the current position is clearly not a fit.

1. Line 111, when discussing the normalization technique, it is better to add the seminal work by Nestrov [1].

1. Line 194, missing a space after "...LLaMA".

1. Line 218, Assumption 2 is due to (Chen et al., 2023) cited in the current paper, but not  (Gorbunov et al., 2024). Moreover, though I understand the meaning of $u\in[x,y]$, it is better to provide a formal definition for readers not seen such a condition before.

1. Line 243, Lemma 1 is wrong due to wrong/inaccurate steps in the proof.

    1. Line 1053, the inequality "...$\leq \frac{1}{4}$" does not hold for $\frac{1}{4}$.

    1. Line 1062, the definition of $\psi_k$ is inaccurate. It should condition all randomness up to $x^k$.

    1. Line 1082, $2L_0\sqrt{d}\gamma_k$ is wrong, it should be $\exp(\frac{1}{48\sqrt{d}\log\frac{1}{\delta}})L_0\sqrt{d}\gamma_k$, which can be arbitrairly large as $\delta\to 1$. A similar issue holds for the term $\frac{\\\|\nabla f(x^{k-1})\\\|_2}{48\sqrt{d}\log\frac{1}{\delta}}$.

    1. Line 1085, this step is both right and wrong. It can be implied by bounding $\\\|\nabla f(x^k)\\\|_2$ directly, but not the way used in the current proof.

    1. Line 1100, the choice of $\lambda$ collapses the whole proof. Note that $\lambda$ in Lemma 3 can only be a real constant. However, the current choice is a random variable that depends on the randomness of the entire optimization process.

1. Due to the above point, any result related to Lemma 1 (i.e., all high-probability convergence theorems) does not hold anymore.

1. For the left two in-expectation results, Theorems 3 and 4 are not surprising. As far as I can check, all the proofs are standard and similar to prior works, without any new technical insights. Therefore, I cannot recognize them as very meaningful. In addition, their proofs also contain many errors. Here, I list some:

    1. Line 1568, everything should be in $2$-norm.

    1. Line 1587, missing a constant related to $d$ in the first step.

    1. Similar issues in the proof of Theorem 4.

1. Line 322, it should be "Clip".

1. For experiments, I also have two questions:

    1. Did the authors also run minibatch-SignSGD and MajorityVote-SignSGD? If yes, please report the results, as these two algorithms are two major methods studied in the paper. If not, I think it is reasonable to run new experiments.

    1. Please also report the confidence interval for 350M and 1.3B models.

1. Line 923, the statement of Markov's inequality (Proposition 3) seems not correct.

1. Line 932, $\mid k$ should be $\mid D_{k-1},\dots,D_1$.

1. Line 950, Lemma 5 only holds for $a_i\geq 0,\forall i\in[d]$. Moreover, inequality (15) is wrong since $\\\|A\nabla f(x^k)\\\|_2$
 should be $\\\|\nabla f(x)\\\|_2$. In addition, please either use $\\\|\cdot\\\|$ to denote $\\\|\cdot\\\|_2$ or stick to $\\\|\cdot\\\|_2$, but do not use both like in the proof.

**References**

[1] Nesterov, Yurii E. "Minimization methods for nonsmooth convex and quasiconvex functions." Matekon 29.3 (1984): 519-531.

**Questions:**

See **Weaknesses**.

**Details Of Ethics Concerns:**

No ethics concerns.

---

### Official Review · Reviewer_q5wA · 2025-10-28

**Soundness:** 3
**Presentation:** 2
**Contribution:** 2
**Rating:** 4
**Confidence:** 5

**Summary:**

The paper studies sign-based stochastic optimization under generalized $(L_0,L_1)$-smoothness and heavy-tailed noise. It provides new high-probability convergence bounds for minibatch SignSGD and Majority-Vote SignSGD. The paper also provides empirical validation through pre-training experiments on large language models, where sign-based methods other baselines in terms of perplexity and training stability.

**Strengths:**

1. This paper gives high-probability bounds for sign-based methods under both $(L_0, L_1)$-smoothness and heavy-tailed noise.

2. The proposed M-SignSGD achieves the convergence guarantee without growing batch sizes, which is attractive for resource-constrained training. The authors also include practical extensions like parameter-free tuning and momentum.

3. The paper is also supported by experiments on large-scale LLMs, showing superior performance of the proposed methods over baselines in perplexity and robustness.

**Weaknesses:**

1. Minibatch SignSGD and Majority-Vote SignSGD both need batch sizes (or worker counts) that grow with $1/\epsilon$, which can be costly in memory. Although M-SignSGD avoids large batches, it is no longer bound by the high-probability bound, which differs from the previous analysis. It is not clear why we can not avoid batches for the high-probability bound.

2. The Majority-Vote SignSGD requires symmetric and unimodal noise assumption, which is obviously a very strong assumption.

3. Theorem 4 provides the parameter-free tuning; however, it still requires $\gamma_0 \leq \frac{1}{90 L_1 d}$. In this sense, it seems not truly parameter-free.

4. The main contribution seems to lie in high probability bounds, but the experiments are only conducted on M-SignSGD, which is not a high probability bound. It is quite strange that the authors do not report results on Minibatch SignSGD and Majority-Vote SignSGD, which are the main contribution algorithms in this paper.

**Questions:**

See the Weakness part.

---

### Official Review · Reviewer_1kZD · 2025-10-29

**Soundness:** 1
**Presentation:** 2
**Contribution:** 1
**Rating:** 2
**Confidence:** 5

**Summary:**

This paper mainly delves into the high-probability convergence of signSGD under $(L_0,L_1)$-smoothness and heavy-tailed noise. Both centralized and distributed settings are studied. The authors also present the in-expectation convergence of signSGD with momentum. For almost all algorithms, their parameter-free versions are also analyzed, at the cost of polynomially weaker constants. Experiments are conducted to validate certain theoretical claims (noise and smoothness dependence). The authors also include pretraining LLMs to validate the effectiveness of sign-based methods.

**Strengths:**

1. The paper is relatively well-written and easy to follow.
2. The experiments are comprehensive, and I am happy to see Figures 1 and 2, which support theoretical claims to some degree.
3. As a common practice, using various optimizers to pretrain LLaMA models on C4 meets the standards of an optimization paper in ICLR.
4. The results of MOE training for high data-to-model ratio are reported, which is quite rare and valuable for optimization papers. Since in real-world scenarios, we often go far beyond the Chinchilla optimal ratio.

**Weaknesses:**

I am convinced that this version of the paper cannot be accepted for ICLR, with most theoretical results seemingly incorrect and potential problems in the empirical results. I will go over them in detail.

---

**1 Theoretical aspect**

**SignSGD GENERAL CONVERGENCE LEMMA 1 is not correct**. The authors have a major misunderstanding about the concentration inequality Lemma 3. In _Line 1101_: The authors erroneously set $\lambda$ to be a random variable depending on the algorithmic trajectory. This is clearly wrong, suggesting that all high-probability bounds in this paper are not correct. The authors may refer to Appendix A.1 in [1], and it can be easily seen that such $\lambda$ would invalidate the whole lemma.

**I am also skeptical about the in-expecation convergence of M-SignSGD**. In _Line 1579-1588_, the authors used Lemma 4 and Assumption 4 to bound the heavy-tailed noise. However, Lemma 4 is stated under $l_2$-norm, but the one used here is the $l_1$-norm version, which makes Lemma 4 not directly applicable. Extra clarifications are needed for this matter. Perhaps one may consider the coordinate-wise version of Lemma 4 and apply it here. This issue might be addressed, but the current derivations are problematic.

---

**2 Empirical aspects**

**Baselines (AdamW) seem to be undertuned**. In Table 6, the optimal lr for AdamW is smaller than M-SignSGD, which makes me very confused. It is well-known that M-SignSGD/Muon could outperform with AdamW if the RMS-norms of their updates are aligned. Generally speaking, this will result in $lr_{sign}\approx 5lr_{AdamW}$, since the RMS-norm of the AdamW update is roughly 0.2 [2]. I would strongly suggest that the authors evaluate other hyperparameter choices.

**Another serious problem is to turn off Nesterov momentum for AdamW**. This is fine if you keep it down for other optimizers, but the current setup is just unfair, since Nesterov acceleration is widely acknowledged and empirically validated to strongly boost the performance [1-4].

Lastly, the authors does not justify why the weight-decay of M-ClippedSGD and M-NSGD are set to zero. Also, the 0.01 weight decay for M-SignSGD and AdamW seems to be smaller than the commonly used 0.1.

**References**

[1] A High Probability Analysis of Adaptive SGD with Momentum.

[2] Muon is scalable for LLM training.

[3] MARS-M: When Variance Reduction Meets Matrices.

[4] MARS: Unleashing the Power of Variance Reduction for Training Large Models.

[5] Fantastic Pretraining Optimizers and Where to Find Them.

---

**3 Motivations**

This is the least significant point here. I don't think this paper is very well-motivated. First, there are no justifications for why high-probability convergence is important and valuable to the broader ML community. Although, as a theory guy myself, I admit that to establish theoretical convergence for a known algorithm is meaningful, in the main body of the paper, the authors still need to justify the reason to study it. To make things worse, given the current evidence indicating that most of the high-probability bounds are erroneous, this paper seems to become meaningless. Besides this part, it would also be better to empirically validate $(L_0,L_1)$-smoothness and heavy-tailed noise for signSGD, otherwise the paper will become a "math flexing" (not to mention that such flex is wrong!).

---

**4 Novelty**

I checked most of the proof in detail and did not find any novel technical insights. The tools are all relatively standard and quite well-known in signSGD/normalized momentum literature. For the signSGD part, the methods come mostly from Sun et al. (2023). For the generalized smoothness part, the analysis stems from Chen et al. (2023); Gorbunov et al. (2024).  For the heavy-tailed part, the HT batching tool comes from Kornilov et al. (2024), while the more advanced one in Liu & Zhou (2024) is not utilized. For the high-probability convergence, the type of Bernstein inequality/Freedman's inequality (or its variants) is well-known, but the paper did not invoke them correctly. For the parameter-free part, the type of inequalities to bound unknown parameters is due to Hübler et al. (2024b).

---

**5 Others**

1. I would strongly advise the authors to properly use the \citet{} and \citep{} commands.
2. Line 1145: $\epsilon^k$ should be in the $l_1$-norm.
3. Line 460: The citations of Zhang et al. (2020b) and Liu et al. (2023a) are inappropriate. The former does not include pretraining tasks on the Transformer architecture, which does not fit into the empirical settings in this paper. The latter is irrelevant and does not discuss optimization characteristics like generalized smoothness.
4. Line 2253: I would suggest reporting the exact value of the Chinchilla optimal ratio.
5. Line 2252: This is not necessarily a problem, but a sequence length of 512 seems to be smaller than common practice. See [1]. I understand the computational resources might be a burden, but many open-sourced implementations I encounter adopt the medium choice of 1024.

**Reference**

[1] The FineWeb Datasets: Decanting the Web for the Finest Text Data at Scale.

**Questions:**

Please see the __Weakness__ part.

**Details Of Ethics Concerns:**

None.

---

### Official Review · Reviewer_e67G · 2025-11-11

**Soundness:** 4
**Presentation:** 4
**Contribution:** 3
**Rating:** 4
**Confidence:** 3

**Summary:**

This paper studies stochastic optimization of non-convex and $(L_0, L_1)$ smooth functions under heavy-tailed gradient noise. Two sign based methods are analyzed. In addition, two special cases of PL functions, and symmetric noise are also studied.

**Strengths:**

1. This paper presents the first high-probability bound for the nonconvex generalized smooth functions.
2. In the case of symmetric and unimodal noise, high probability bound is also derived when generalized smoothness is assumed.
3. Large-scale experiments are conducted to validate the performance of M-SignSGD.

**Weaknesses:**

1. For Theorem 1, large-batch is required. Is this necessary or a proof artifact? Very large batch is typically not achievable due to hardware limit.
2. The technical challenges in dealing with additional generalized smoothess condition using sign based method needs more discussions.

**Questions:**

1. Are the same hyper-parameters in Table 6 used for all sized models?

---

### Meta-Review · Area_Chair_xsus · 2025-12-30

**Summary:**

An algorithm called SignSGD is proposed for optimizing generalized $(L_0,L_1)$-smooth functions using sign-based stochastic gradient with majority voting. Correctness of the theoretic claims is questioned. The requirement on the size of the batches are considered impractical.

**Reviewer Concerns:**

No rebuttal is provided.

**Reviewer Scores:**

No score will change.

---

### Decision · Program_Chairs · 2026-01-26

Reject